# Higgs decays to two leptons and a photon beyond leading order in the SMEFT

T. Corbett[1*], T. Rasmussen[2]

**1** Niels Bohr International Academy,
Niels Bohr Institute, University of Copenhagen,
Blegdamsvej 17, DK-2100, Copenhagen, Denmark
* corbett.t.s@gmail.com

May 3, 2022

## Abstract

We present the three-body decay of the Higgs boson into two leptons and a photon to dimension-eight in the Standard Model Effective Field Theory (SMEFT). In order to obtain this result we interfere the full one-loop Standard Model result with the tree-level result in the SMEFT. This is the first calculation of the partial width of the Higgs boson into two leptons and a photon in the SMEFT to incorporate the full one-loop dependence for the Standard Model as well as the full tree level dimension-eight dependence in the SMEFT. We find that this channel can aid in distinguishing strongly interacting and weakly interacting UV completions of the SMEFT under standard assumptions. We also find that this channel presents the opportunity to distinguish different operator Classes within the SMEFT, potentially including contact $H\bar{\ell}\ell\gamma$ operators which are first generated only at dimension-eight in the SMEFT.

# 1 Introduction

The coupling of the Higgs boson to two photons or a $Z$–boson and photon in the Standard Model (SM) were developed in [1–3]. The decay $H \to \bar{\ell}\ell\gamma$ in the SM has more recently been considered in [4–12]. In the case of the decay $H \to Z\gamma$ the narrow width approximation is typically employed for the $Z$ boson. Relaxing this assumption is expected to have a negligible effect on the overall prediction for the Higgs boson decay to two leptons and a photon.

The Standard Model Effective Field Theory (SMEFT) is a useful tool for studying the effects of heavy New Physics (NP) in a generally model independent fashion. The primary assumptions of this methodology are that the NP is too heavy to produce directly at the energies considered and that the Higgs boson belongs to an $SU(2)_L$ doublet. Under these assumptions the SMEFT is formed as a tower of operators suppressed by the mass scale of the heavy new physics $\Lambda$:

$$\mathcal{L}_{\text{SMEFT}} = \mathcal{L}_{\text{SM}} + \sum_{i}^{\infty} \sum_{j} \frac{1}{\Lambda^i} \mathcal{Q}_j^{(4+i)} \,. \tag{1}$$

Operators of dimension $(4 + i)$ are suppressed by the heavy scale $\Lambda^i$ and generally, therefore, higher dimension operators are treated as higher order corrections in the SMEFT approach. The sum over $j$ is over all operator forms at a given order $i$. For LHC studies odd-dimension operators are generally negligible as they are $B$ and/or $L$ violating and therefore expected to be strongly suppressed. The SMEFT at dimension-six has been a major area of investigation, particularly since the discovery of the Higgs boson. Recently much interest has been generated around how dimension-eight operators impact studies of the SMEFT [13–19] and how they change our understanding of truncation error in the SMEFT [20–23].

In the context of the SMEFT the present work is the first work to have studied the effect of interfering the non-resonant SM contribution for $H \to \bar{\ell}\ell\gamma$ with the SMEFT contributions. The SMEFT tree-level interference with the resonant SM contributions have been used across the field when performing global fits. One-loop studies with on-shell $\gamma$ and $Z$ were made in [24, 25] for $HZ\gamma$ and [26–29] for $H\gamma\gamma$.

In [30], the authors calculate $H \to 4\psi$ in the SMEFT and found the many novel couplings generated in the SMEFT could allow for larger corrections to the typically employed narrow

width calculation than anticipated. Indeed, couplings of the Higgs which do not occur in the SM at tree level, like $H\gamma\gamma$, $Hgg$, and $H\bar\psi\psi\gamma$ result in contributions to the partial width that are entirely neglected in the narrow width approximation. Taking from this, this work seeks to analyze the implications of allowing for off-shell $Z$ or $\gamma$ in the decay of the Higgs boson to two fermions and a photon.

This article is organized as follows: In Section 2 we lay out our notation and in Sec. 3 we discuss the SM tree-level and one-loop contributions to the process $H \to \bar\ell\ell\gamma$. In Sec. 4 we derive the contributions in the SMEFT up to and including $1/\Lambda^4$ effects, this includes the consistent calculation of dimension-six-squared contributions as well as contributions from dimension-eight operators. In this section we also look at different cuts in the invariant mass of the dilepton system and how they can be used to potentially distinguish operators generated by tree- and loop- processes in the UV, as well as distinguish between different classes of operators in the SMEFT. In the conclusions, Section 5, we summarize this work and briefly mention future perspectives.

Included are many appendices: the first, App. A lays out the Feynman rules for the terms in the one-loop effective action in the SMEFT that contribute to the partial width, App. B lays out the necessary Feynman rules in the SMEFT for deriving $H \to \bar\ell\ell\gamma$ up to $\frac{1}{\Lambda^4}$, App. C outlines and briefly discusses results of matching onto the one-loop effective action in the SM, and App. D contains results from the main text in the $M_W$ input parameter scheme. Appendix E discusses the approximately vanishing contribution from the tree-loop interference in the SM while App. F discusses the often neglect dipole operator contribution.

Also included in the ancillary files are `Mathematica` notebooks which derive the one-loop contributions to the SM process, including demonstrating how the one-loop effective action can be derived and employed in performing calculations of this sort.

## 2  Notation

In this article we follow the notation laid out in [31–34]. The geoSMEFT [32] methodology is used to obtain most of the relevant vertices in the SMEFT power counting to order $\frac{1}{\Lambda^4}$. Certain operators at dimension eight have not yet been written in the geoSMEFT, here we use those laid out in [35]. An alternative source of the dimension-eight operator basis can be found in [36]. We only lay out the conventions relevant to this work, further details can be found in the works cited above.

The couplings to the fermions coming from the covariant derivative are written:

$$\mathcal{L}_\psi = \bar\psi\gamma^\mu \left[ \partial_\mu + i\bar g_3 G_\mu^A T^A + i\frac{\bar g_2}{\sqrt{2}}(W_\mu^+ T^+ + W_\mu^- T^-) + i\bar g_Z(T_3 - s_z^2 Q_\psi)Z_\mu + iQ_\psi \bar e A^\mu \right] \psi \quad (2)$$

All fields appearing in the above equations are mass eigenstate fields. $T^\pm, T_3$ are the generators of $SU(2)_L$ written in the charged basis and $Q_\psi$ is the charge of the fermion. The barred quantities include corrections from the SMEFT and are defined to all orders in the SMEFT in [32]. Barred quantities such as $\bar g_2$ can be expanded in terms of Wilson coefficients and the unbarred gauge couplings as:

$$\bar g_2 = g_2 \left( 1 + 2c_{HW}v^2 + \frac{1}{2}c_{HW}^{(8)}v^4 + \frac{3}{2}c_{HW}^2 v^4 \right) \quad (3)$$

The expression for $\bar{g}_Z$ is too cumbersome to write to dimension eight in text. In this article $v$ is the vacuum expectation value that minimizes the Higgs potential in the SMEFT (frequently written as $\bar{v}_T$ in the literature). The ancillary files contain an updated version of the `Feynrules` model defined in [33] and can be used to derive the full expression. To dimension six $\bar{g}_Z$ is given by:

$$\bar{g}_Z = \frac{g_1^2 + g_2^2}{\sqrt{g_1^2 + g_2^2}} \left[ 1 + \frac{1}{g_1^2 + g_2^2} \left( g_1^2 c_{HB} v^2 + g_2^2 c_{HW} v^2 + g_1 g_2 c_{HWB} v^2 \right) \right] \tag{4}$$

The quantity $\bar{g}_3$ is not relevant to this work and so we do not define it. The new mixing angle, $s_z$, takes into account new mixing effects at dimension-eight. It is equivalent to the barred Weinberg angle at dimension six:

$$\bar{s}_W^2 = \frac{g_1^2}{g_1^2 + g_2^2} \left[ 1 + \frac{1}{g_1^2 + g_2^2} \left( 2g_2^2 c_{HB} v^2 - 2g_2^2 c_{HW} v^2 + \frac{g_2}{g_1}(g_2^2 - g_1^2) c_{HWB} v^2 \right) \right] . \tag{5}$$

The dimension-eight contributions to $s_z$ and $\bar{s}_W$ are also too cumbersome to fit neatly in print form, but can be derived using the Feynrules package [33] or can be looked up in the extensive Appendix of [13].

The coupling of the Higgs boson to fermions is given by:

$$\mathcal{L}_{\text{Yukawa}} = \left[ -H[Y_\psi]^\dagger + H \left( c_{\psi H}^{(6)} H^\dagger H + c_{\psi H}^{(8)} (H^\dagger H)^2 \right) \right] \Psi_L \psi_R \tag{6}$$

$\Psi_L$ represents a fermionic left-handed doublet and $\psi_R$ a fermionic right-handed $SU(2)_L$ singlet. Neglecting goldstone bosons for simplicity we can express the doublet in terms of the Higgs mass eigenstate $h$ as:

$$H = \begin{pmatrix} 0 \\ \frac{v + c_{H,\text{kin}} h}{\sqrt{2}} \end{pmatrix} \tag{7}$$

$$c_{H,\text{kin}} = 1 + \frac{1}{4}(c_{HD} - 4c_{H\square})v^2 - \frac{1}{32}(4c_{HD}^{(8)} + 4c_{HD,2}^{(8)} - 3[c_{HD} - 4c_{H\square}]^2)v^4 \tag{8}$$

Expanding Eq. 6 defines the tree-level masses of the fermions and their relation to the SM masses:

$$\hat{m}_\psi = \frac{v}{\sqrt{2}} \left[ Y_\psi - \frac{v^2}{2} c_{\psi H} - \frac{v^4}{4} c_{\psi H}^{(8)} \right] \tag{9}$$

In the next section we perform calculations to one-loop in the SM. As such we only include the SM dependence. Again we neglect goldstone bosons in this discussion. The one-loop calculations below are in general $R_\xi$ gauge and therefore do include goldstone boson dependence. The coupling of the Higgs boson to gauge bosons and the relevant triple gauge couplings come from the Higgs gauge kinetic term and the gauge boson kinetic terms of the SM:

$$\mathcal{L}_{HVV} = (D^\mu H)^\dagger (D_\mu H) - \frac{1}{4} W^{A,\mu\nu} W_{\mu\nu}^A - \frac{1}{4} B^{\mu\nu} B_{\mu\nu} \tag{10}$$

Where $W^{A,\mu\nu}$ and $B^{\mu\nu}$ are the field strength tensors of the $SU(2)_L$ and hypercharge gauge fields respectively.

# 3   Higgs decays to two leptons and a photon in the SM

## 3.1   Tree level decay

In the Standard Model the Higgs boson decays to two leptons via the Yukawa coupling, relative to the mass scale of the Higgs this is typically a very small coupling, but can be relevant for the $\mu$ and $\tau$. Taking the lepton mass to be zero, but keeping the Yukawa coupling nonzero, the decay width of a Higgs boson to two leptons and a photon at tree level is given by:

$$\Gamma^{(0)}_{h\to\bar{\ell}\ell\gamma} = \frac{8\pi^2}{64\bar{m}_H^3(2\pi)^5}\int ds_1 ds_2 \Theta(G[s_1,s_2,\bar{m}_H^2,\hat{m}_\ell^2,0,\hat{m}_\ell^2]\leq 0)\frac{4\bar{e}^2\hat{m}_\ell^2}{v^2}\frac{(s_2^2+\bar{m}_H^4)}{s_1(s_1+s_2-\bar{m}_H^2)}\,,$$
(11)

where $\Theta$ denotes the unit step function, the superscript "(0)" indicates this is the tree level result, and $G=0$ defines the boundary of integration and is given by [37]:

$$G[x,y,z,u,v,w] = -\frac{1}{2}\begin{vmatrix} 0 & 1 & 1 & 1 & 1 \\ 1 & 0 & v & x & z \\ 1 & v & 0 & u & y \\ 1 & x & u & 0 & w \\ 1 & z & y & w & 0 \end{vmatrix}$$
(12)

The kinematic invariants are defined as $s_1 = (k_\psi + k_\gamma)^2$ and $s_2 = (k_\psi + k_{\bar\psi})^2$. This integral contains an IR divergence which we regulate by requiring the photon energy be above 5 GeV following the approach of [5]. Adopting the input parameters given in Table 1 we find the tree level results:

$$\begin{aligned}
\Gamma^{(0)}_{h\to e^+e^-\gamma} &= 4.13\cdot 10^{-12}\,\mathrm{GeV} & (4.09\cdot 10^{-12}\,\mathrm{GeV})\,, \\
\Gamma^{(0)}_{h\to\mu^+\mu^-\gamma} &= 1.23\cdot 10^{-7}\,\mathrm{GeV} & (1.18\cdot 10^{-7}\,\mathrm{GeV})\,, \\
\Gamma^{(0)}_{h\to\tau^+\tau^-\gamma} &= 2.10\cdot 10^{-5}\,\mathrm{GeV} & (2.05\cdot 10^{-5}\,\mathrm{GeV})\,,
\end{aligned}$$
(13)

where the first result denotes the use of the $\hat{\alpha}$ input parameter scheme and the result in parenthesis corresponds to the $\hat{m}_W$ scheme [38, 39]. Hatted quantities correspond to input parameters while barred quantities are the Lagrangian parameters. Therefore, for example, $\bar{m}_W$ is determined from $\hat{\alpha}$ in the $\hat{\alpha}$ scheme, while it corresponds to $\hat{m}_W$ in the $\hat{m}_W$ scheme. These results include the full lepton mass dependence. Comparing with the leading order results[1] this results in an approximately $+10\%$ correction for taus, a $+4\%$ correction for muons, and a $-14\%$ correction for electrons. These corrections do not scale as $\hat{m}_\ell/\bar{m}_H$, this is due to the photon mass cut off. This behavior can be understood from $E_\gamma$ dependence shown in Figure 4 of [11]. The error in the phase-space integrations is approximately 5 per mil. Phase space integrations were performed using the Vegas routine from the Cuba library [40].

## 3.2   One loop decay

As a result of the smallness of the lepton masses compared with the Higgs boson mass scale we expect the chirally suppressed tree level terms to be of the same order or smaller than the

---

[1]Here in the mass expansion we *retain* the fermion mass dependence in the denominators as this leads to quicker numerical convergence of the integrals.

| Input parameters | Value | Ref. |
|---|---|---|
| $\hat{m}_Z$ [GeV] | 91.1876(21) | [41] |
| $\hat{m}_W$ [GeV] | 80.387(16) | [42] |
| $\hat{m}_h$ [GeV] | 125.10(14) | [41] |
| $\hat{m}_t$ [GeV] | 172.4(7) | [41] |
| $\hat{m}_e$ [MeV] | 0.51099895000(15) | [41] |
| $\hat{m}_\mu$ [MeV] | 105.6583745(24) | [41] |
| $\hat{m}_\tau$ [GeV] | 1.77686(12) | [43] |
| $\hat{G}_F$ [GeV$^{-2}$] | $1.1663787 \cdot 10^{-5}$ | [43, 44] |
| $\hat{\alpha}_{EW}$ | 1/137.035999084(21) | [41] |
| $\Delta\alpha$ | $0.0590 \pm 0.0005$ | [45] |
| $m_W^{\hat{\alpha}}$ [GeV] | $80.36 \pm 0.01$ | – |
| $\Delta\alpha^{\hat{m}_W}$ | $0.0576 \pm 0.0008$ | – |
| $\Gamma_Z^{\hat{\alpha}}$ | $2494.4 \pm 0.7$ MeV | – |
| $\Gamma_Z^{\hat{m}_W}$ | $2495.7 \pm 1.0$ MeV | – |

Table 1: Input parameter values used in this work, the last two lines are $m_W$ as predicted in the $\alpha$ scheme and $\Delta\alpha$ as predicted in the $m_W$ scheme. The $Z$ widths are for the SM using the values given above. Table taken from [20], where further details can be found. Note that the fermion mass is *always* used as the input parameter.

one-loop contributions. The leading term in the interference of the tree amplitude with the loop amplitude is proportional to $m_\ell^2/(16\pi^2)$, because of the smallness of the lepton mass we assume:

$$\frac{m_\ell^2}{16\pi^2} \quad \to \quad 0 \,. \tag{14}$$

This is further justified in App. E.

In order to perform the relevant SM loop calculations we employ the background field method (BFM) [46], and use the conventions and Feynman rules presented in [33]. This choice is convenient as it reduces the number of diagrams contributing to each process as well as preserving the naive Ward identities [46, 47], thereby greatly simplifying many expressions. In the BFM the bosonic fields are doubled as $\phi \to \phi + \hat{\phi}$, but only the quantum gauge fields (corresponding to the unhatted fields) are gauge fixed. As fermionic fields are not a part of the gauge fixing procedure their fields are not doubled, as such the Feynman rules involving the fermions are the same in the BFM and in traditional gauge fixing. In what follows the hatted fields, $\hat{\phi}$, correspond to background fields. One-loop contributions to the effective action are formed of loops with external background fields and internal quantum fields. It can be shown that the effective action in the BFM is equivalent to that in the traditional gauge fixing [48]. The effective action is the generating functional of one-particle irreducible diagrams, once derived the background fields can be gauge fixed independent of the choice of gauge fixing for the quantum fields and the S-matrix elements can be derived. Using unitary gauge for the background fields has the added benefit of removing goldstone boson contributions from one-particle reducible diagrams formed from the effective action.

In adopting the BFM we first construct the relevant contributions to the one-loop effective action which contribute to the one-loop couplings $\hat{h}\hat{\gamma}\hat{\gamma}$ and $\hat{h}\hat{Z}\hat{\gamma}$. We then form the one-particle reducible diagrams contributing to the process $\hat{h} \to \bar{\ell}\ell\hat{\gamma}$. For simplicity we do not match the

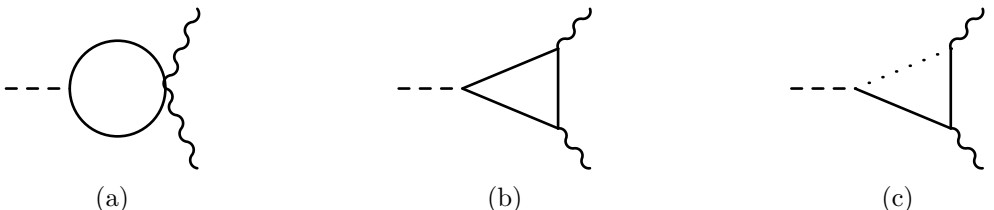

(a)                                    (b)                                    (c)

Figure 1: Standard Model diagrams contributing to Higgs boson decays to two photons and a $Z$ boson and photon. In (a) and (b) the solid line represents the contribution of either $W-$ bosons, the charged goldstone bosons, charged ghosts, or fermions. In the case of the $HZ\gamma$ coupling loops can be composed of one goldstone boson and two $W-$bosons or two goldstone bosons and one $W$. This is illustrated in Figure (c) where the solid line may represent $W-$ bosons while the dotted line represents goldstone bosons, or vice versa.

one-loop diagrams involving external fermions to the effective action, this is possible as the Feynman rules involving fermions are the same in the background field method as in standard $R_\xi$ gauge fixing. The relevant diagrams contributing to the effective action at one loop are shown in Fig. 1. Figure 2 shows diagrams which do not contribute in the background field method – both diagrams vanish identically as a result of the Ward identities [46, 47][2].

These diagrams can be matched onto the one-loop effective action given by:

$$
\begin{aligned}
\mathcal{L}_{1-\mathrm{loop}} &= f_{H\gamma\gamma}^{(1)}(k_1, k_2)\hat{h}\hat{F}_{\mu\nu}\hat{F}^{\mu\nu} + f_{H\gamma\gamma}^{(2)}(k_1, k_2)\hat{h}(\partial^\mu \hat{F}_{\mu\nu})(\partial_\sigma \hat{F}^{\sigma\nu}) \\
&\quad + f_{HZ\gamma}^{(1)}(k_1, k_2)\hat{h}\hat{F}_{\mu\nu}\hat{Z}^{\mu\nu} + f_{HZ\gamma}^{(2)}(k_1, k_2)\hat{h}(\partial^\mu \hat{F}_{\mu\nu})\partial_\sigma\partial^\nu \hat{Z}^\sigma \quad\quad (15) \\
&\quad + f_{HZ\gamma}^{(3)}(k_1, k_2)\hat{h}\hat{F}_{\mu\nu}\square \hat{Z}^\nu \, ,
\end{aligned}
$$

where $\hat{F}_{\mu\nu}$ and $\hat{Z}_{\mu\nu}$ are the field strength of the photon and $Z$-boson respectively. We note that, as we are matching below electroweak symmetry breaking (EWSB), the appearance of the massive gauge boson $Z$ is not restricted by gauge symmetry. As such the operator form corresponding to $f_{H\gamma\gamma}^{(2)}$ is split between two $\hat{h}\hat{Z}\hat{\gamma}$ operators. The $f$ are form factors coming from matching the one-loop calculations in the background field method onto the effective action. The $f(k_1, k_2)$ indicates that these form factors depend on the momenta of the external bosons, which by conservation of momentum can be chosen to be the momenta of the photons or photon and $Z$-boson.

The Feynman rules for these effective operators are given in Appendix A. We note that in both cases the operators corresponding to the form factors $f^{(2)}$ and $f^{(3)}$ give vanishing contributions when one-particle reducible diagrams are formed and the lepton masses are taken to be zero.

Employing the Feynman rules of [33] and using Package-X [49], we have derived the form factors of Eq. 15. We have confirmed that the $f^{(1)}$ are gauge invariant and the $f^{(2)}$ vanish identically for on-shell external particles. The analytic results for the form factors are reproduced in simplifying limits in Appendix C: The $H\gamma\gamma$ operators are reproduced for on-shell Higgs momenta and in the limit of small photon off-shell momenta and arbitrary gauge parameter. This serves two purposes: the first is to demonstrate that the gauge parameter dependence vanishes identically in the limit the photon momenta vanishes as well as to show

---

[2]In the case of the $\hat{Z}\hat{\gamma}$ mixing, this diagram vanishes only for on-shell photon.

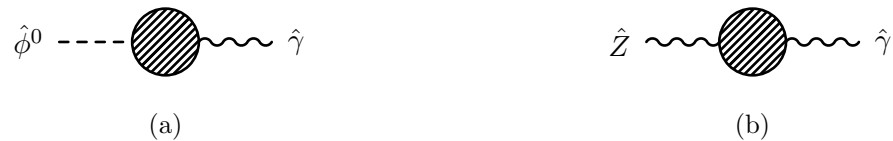

(a)                                             (b)

Figure 2: Standard Model diagrams contributing to Higgs boson decays to two photons which vanish identically. These diagrams, if nonzero, would contribute to one-particle reducible diagrams where the $\hat{\phi}^0$ or $\hat{Z}$ are contracted with a $\hat{H}\hat{\phi}^0\hat{Z}$ or $\hat{H}\hat{Z}\hat{Z}$ vertex.

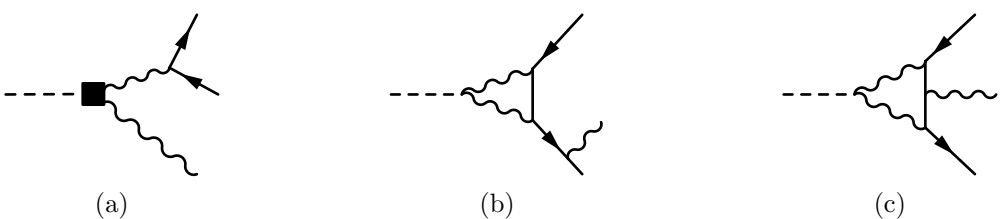

(a)                            (b)                            (c)

Figure 3: Diagrams contributing to the process $H \to \bar{\ell}\ell\gamma$. In (a) the square vertex represents an insertion of the one-loop effective action coupling to two photons or a $Z$ and photon. In (b) and (c) the Higgs boson decays to two fermions via a loop containing two $W$s or two $Z$s, a photon is then either radiated by an external or the internal fermion line.

the leading off-shell dependence of the form factors. It should be noted that for the calculation of the process $H \to \bar{\ell}\ell\gamma$ the off-shell photon is allowed to have a momentum-squared up to the Higgs mass, and the limit in the Appendix is not relevant. In the case of the coupling $HZ\gamma$ we have set $\xi = 1$ in order to have more compact expressions. In this case we again find that the results are gauge parameter invariant for on-shell external particles. In what follows we use the full momentum dependence in order to allow intermediate particles to go arbitrarily off shell.

In order to form the amplitude for $H \to \bar{\ell}\ell$ we must decay the off-shell photon and $Z$ to two leptons. Gauge invariance also demands we add the box diagrams and one-loop decays of the Higgs boson to two fermions where one fermion radiates a photon. These diagrams can be found in Figure 3. We assume that lepton Yukawa couplings as well as masses are zero in the loop diagrams, this removes the contribution of goldstone bosons in the loops as well as the loops where the lepton couples directly to the Higgs boson.

The one-loop amplitudes can be parameterized as:

$$
\begin{aligned}
i\mathcal{M} &= \left(c_{1L}\eta^{\mu\nu} + c_{2L}k_2^\mu k_1^\nu + c'_{2L}k_3^\mu k_1^\nu\right)\bar{u}_{k_2}\gamma_\nu P_L v_{k_3}\epsilon_\mu^*(k_1)\\
&\quad + \left(c_{1R}\eta^{\mu\nu} + c_{2R}k_2^\mu k_1^\nu + c'_{2R}k_3^\mu k_1^\nu\right)\bar{u}_{k_2}\gamma_\nu P_R v_{k_3}\epsilon_\mu^*(k_1),
\end{aligned}
\tag{16}
$$

where the $c_i$ parameterize the loop contributions, $\bar{u}_{k_2}$ is the barred on-shell spinor for the lepton with momentum $k_2$, $v_{k_3}$ is the on-shell spinor for the anti-lepton with momentum $k_3$, and $\epsilon_\mu^*(k_1)$ is the on-shell polarization vector for the photon with momentum $k_1$. The diagrams of Fig. 3 generate all $c_i$ when an intermediate photon or $Z$ boson is involved. Diagrams (b) and (c) for intermediate $W$s only generate the left handed $c_i$. We find that the sum of diagrams (b) and (c) for intermediate $Z$ bosons are gauge invariant on their own. As the $W$s don't generate right-handed couplings this also implies the right-handed components of the sum of the diagrams of (a), when including both the intermediate $Z$ and photon, are gauge invariant.

The left handed components of the triangle diagrams require the addition of diagrams (b) and (c) with intermediate $W$s to be gauge invariant.

After obtaining the squared amplitudes corresponding to the one-loop results we find the following partial widths (although the one-loop process does not contain an IR divergence[3] we maintain the requirement that the photon energy be greater than 5 GeV):

$$
\begin{aligned}
\Gamma^{(1)}_{h \to e^+ e^- \gamma} &= 5.70 \cdot 10^{-7} \, \text{GeV} \quad (5.40 \cdot 10^{-7} \, \text{GeV}) , \\
\Gamma^{(1)}_{h \to \mu^+ \mu^- \gamma} &= 3.73 \cdot 10^{-7} \, \text{GeV} \quad (3.88 \cdot 10^{-7} \, \text{GeV}) , \\
\Gamma^{(1)}_{h \to \tau^+ \tau^- \gamma} &= 2.84 \cdot 10^{-7} \, \text{GeV} \quad (2.92 \cdot 10^{-7} \, \text{GeV}) ,
\end{aligned}
\tag{17}
$$

where the superscript "(1)" indicates these are one-loop results. In order to efficiently integrate the phase space for these processes we employed the `CollierLink` library for `Package X` [49] which links the Passarino-Veltman functions of Package X with the `Collier` library [50–53]. Phase space integrations were performed using gauge parameter $\xi = 1$ as well as $\xi = 2$ to confirm gauge invariance of the results. The differences between different flavors of leptons is due to the reduced phase space for increasing fermion mass: in integrating the phase space we have neglected $\hat{m}_\ell$ in the matrix element, but the mass is still used in the integration region as defined by $G$ in Eq. 12.

Dalitz plots of the tree and one-loop partial widths are shown in Figure 4 for the muon case. This is the most interesting comparison as the two contributions differ by less than an order of magnitude. The diagrams are qualitatively the same for the cases of electrons and taus. Qualitatively we can identify that the large $s_2$ region is favorable for isolating the tree-level contribution, $s_2 \sim \bar{m}_Z^2$ isolates the pole region corresponding to $H \to \gamma Z (\to \bar{\ell}\ell)$, while the low $s_2$ region favors the conversion of an off-shell photon to the two leptons. In [8], the authors studied cuts isolating various regions of the phase space for the SM. In the next section we study how the presence of the SMEFT impacts this decay process and seek to identify how to isolate regions of phase space that can emphasize the SMEFT effects.

# 4 SMEFT contributions

We have so far discussed the tree-level and one-loop contributions to the process $H \to \bar{\ell}\ell\gamma$ within the SM. In this context, the assumption of Eq. 14 removed the possibility of interference between the tree and loop processes. As such the observable corresponding to the one-loop amplitude corresponds to a two-loop observable, or symbolically comes with a $1/(16\pi^2)^2$ suppression. In the presence of the SMEFT, however, there are new opportunities to generate tree-level amplitudes contributing to the process $H \to \bar{\ell}\ell\gamma$ which, when interfered with the SM loops, will not be chirally suppressed.

Following recent interest in the dimension-eight contributions in the SMEFT we include the relevant dimension-eight operators. The inclusion of dimension-eight terms is particularly interesting as the $H \to \bar{\ell}\ell\gamma$ process can be generated by a single contact interaction – at dimension-six there is no four-vertex which couples two leptons of the same chirality, a Higgs boson, and a photon. The relevant diagrams are shown in Fig. 5 where a circle represents an effective vertex.

---

[3]See the discussion in [11] below Eq. 2.8

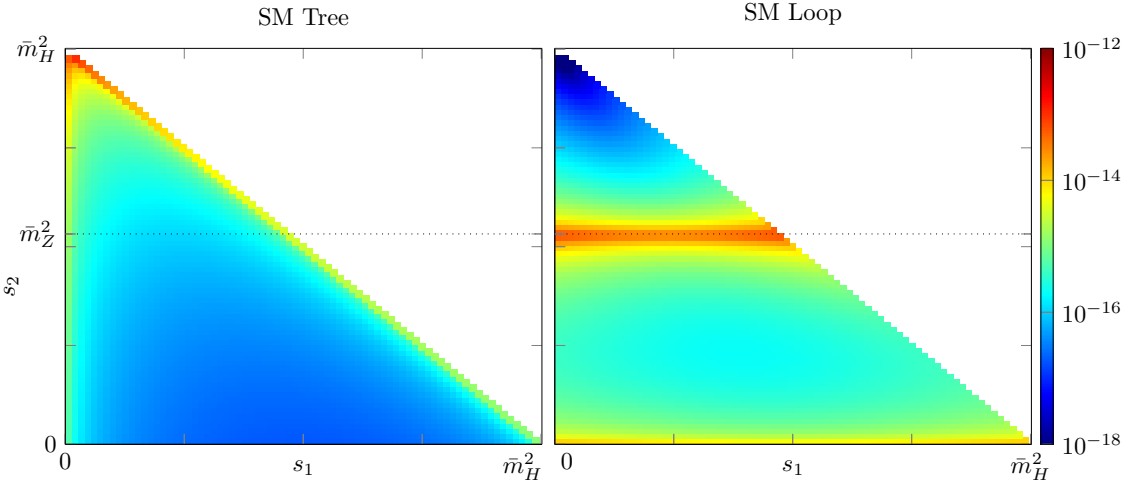

Figure 4: Dalitz plots comparing the tree (Left) and one-loop (Right) contributions to the process $H \to \bar{\mu}\mu\gamma$ in the SM. The dotted line shows where $s_2$, the invariant mass of the di-muon system, crosses the $Z$-pole. Units are $[\text{GeV}]^{-3}$.

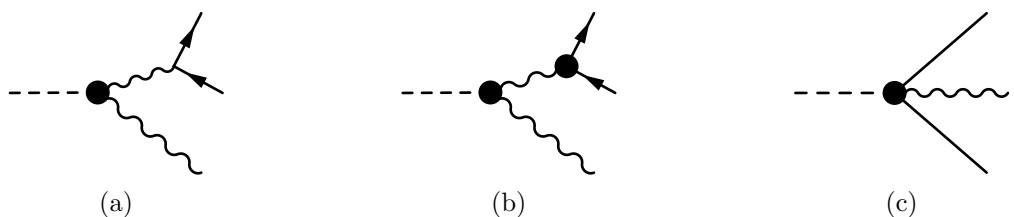

Figure 5: Diagrams contributing to the process $H \to \bar{\ell}\ell$ in the SMEFT. The solid circle represents an insertion of an effective vertex. There is also the possibility that both the $HVV$ and $V\bar{\ell}\ell$ vertex correspond to insertions of dimension-six operators giving an overall dimension-eight contribution. Diagram (a) is generated at orders $1/\Lambda^2$ and $1/\Lambda^4$, (b) requires the insertion of two dimension-six operators and is therefore $\mathcal{O}(1/\Lambda^4)$, and the four-point vertex of diagram (c) occurs first at dimension eight in the SMEFT. We do not consider diagrams with a single operator insertion at the $V\bar{\ell}\ell$ vertex as this occurs only at one-loop.

The operators which shift the Higgs Yukawa couplings are given by the "Class 5" operators:

$$\mathcal{L}_{\text{cl5}} = c_{eH}^{(6)}(H^\dagger H)(\bar{l}e_R H) + c_{eH}^{(8)}(H^\dagger H)^2(\bar{l}e_R H)\,. \tag{18}$$

Those which shift gauge-lepton couplings are the Class 7 operators:

$$
\begin{aligned}
\mathcal{L}_{\text{cl7}} &= c_{Hl}^{(6),1}(H^\dagger i\overleftrightarrow{D}_\mu H)(\bar{l}\gamma^\mu l) + c_{Hl}^{(8),1}(H^\dagger H)(H^\dagger i\overleftrightarrow{D}_\mu H)(\bar{l}\gamma^\mu l)\\
&+ c_{Hl}^{(6),3}(H^\dagger i\overleftrightarrow{D}_\mu^I H)(\bar{l}\sigma^I\gamma^\mu l) + c_{Hl}^{(8),3}(H^\dagger H)(H^\dagger i\overleftrightarrow{D}_\mu^I H)(\bar{l}\sigma^I\gamma^\mu l)\\
&+ c_{He}^{(6),1}(H^\dagger i\overleftrightarrow{D}_\mu H)(\bar{e}_R\gamma^\mu e_R) + c_{He}^{(8),1}(H^\dagger H)(H^\dagger i\overleftrightarrow{D}_\mu H)(\bar{e}_R\gamma^\mu e_R)\\
&+ c_{Hl}^{(8),2}(H^\dagger \sigma_a H)(H^\dagger i\overleftrightarrow{D}^\mu H)(\bar{l}\gamma_\mu \sigma_a l)\,,
\end{aligned}
\tag{19}
$$

where we have used:

$$(H^\dagger \overleftrightarrow{D}_\mu H) = H^\dagger iD_\mu H - (iD_\mu H^\dagger)H \tag{20}$$

$$(H^\dagger \overleftrightarrow{D}_\mu^I H) = H^\dagger i\tau^I D_\mu H - (iD_\mu \tau^I H^\dagger)H \tag{21}$$

While the dimension-eight contributions proportional to $c_{Hl}^{(8),1}$ and $c_{Hl}^{(8),3}$ represent re-scalings of their dimension-six operators by $(H^\dagger H)$ that corresponding to $c_{Hl}^{(8),2}$ represents a new form of operator in Class 7 which does not exist at dimension-six. The last remaining operator corresponding to $c_{Hl}^{(8),\epsilon}$ does not contribute (see [32] for the full operator form). As can be seen in App. B where the Feynman Rules for the SMEFT are collected, these operators only result in a shift in the $\bar{\ell}\ell Z$ coupling and do not shift the coupling of leptons to the photon. This can be understood from the operator form as well, where the covariant derivative acts on the Higgs doublet. As the Higgs, $h$, is chargeless there is no coupling to the photon induced by these operators.

Class 4 operators result in anomalous $H\gamma\gamma$ and $HZ\gamma$ couplings, they are:

$$
\begin{aligned}
\mathcal{L}_{\text{cl4}} &= c_{HB}^{(6)}(H^\dagger H)B^{\mu\nu}B_{\mu\nu} + c_{HB}^{(8)}(H^\dagger H)^2 B^{\mu\nu}B_{\mu\nu}\\
&+ c_{HW}^{(6)}(H^\dagger H)W_a^{\mu\nu}W_{a,\mu\nu} + c_{HW}^{(8)}(H^\dagger H)^2 W_a^{\mu\nu}W_{a,\mu\nu}\\
&+ c_{HWB}^{(6)}(H^\dagger \sigma^a H)W_a^{\mu\nu}B_{\mu\nu} + c_{HWB}^{(8)}(H^\dagger H)(H^\dagger \sigma^a H)W_a^{\mu\nu}B_{\mu\nu}\\
&+ c_{HW,2}^{(8)}(H^\dagger \sigma^a H)(H^\dagger \sigma^b H)W_a^{\mu\nu}W_{b,\mu\nu}
\end{aligned}
\tag{22}
$$

As in the case of the Class 7 operators there is one entirely new operator form which occurs first at dimension-eight. These operators generate a new momentum-dependent coupling of the Higgs boson to two vectors which is not present at tree level in the SM.

In addition to these operators, there exist operators which shift the $H\gamma Z$ coupling without affecting the $H\gamma\gamma$[4]:

$$\mathcal{L}_{\text{cl8}} = c_{WH^4D^2}^{(8),1}(H^\dagger H)(D^\mu H^\dagger \tau^I D^\nu H)W_{\mu\nu}^I + c_{BH^4D^2}^{(8),1}(H^\dagger H)(D^\mu H^\dagger D^\nu H)B_{\mu\nu} \tag{23}$$

Finally, we can generate shifts in the coupling of the leptons to the $Z$ boson as well as a contact four-point interaction via the operators in Tab. 2. These operators as well as the Class 7 operators are implemented in an updated version of the Feynman Rules presented

---

[4]These are referred to in [35] and this work as "Class 8" operators. They are not to be confused with the four-fermion operators called Class 8 in the usual dimension-six literature, which are not relevant to this work.

Class 11: $\psi^2 H^2 D^3$

| $Q_{l^2H^2D^3}^{(8),1}$ | $i(\bar{l}\gamma^\mu D^\nu l)(D_{(\mu}D_{\nu)}H^\dagger H)$ |
|---|---|
| $Q_{l^2H^2D^3}^{(8),2}$ | $i(\bar{l}\gamma^\mu D^\nu l)(H^\dagger D_{(\mu}D_{\nu)}H)$ |
| $Q_{l^2H^2D^3}^{(8),3}$ | $i(\bar{l}\gamma^\mu \tau^I D^\nu l)(D_{(\mu}D_{\nu)}H^\dagger \tau^I H)$ |
| $Q_{l^2H^2D^3}^{(8),4}$ | $i(\bar{l}\gamma^\mu \tau^I D^\nu l)(H^\dagger \tau^I D_{(\mu}D_{\nu)}H)$ |
| $Q_{e^2H^2D^3}^{(8),1}$ | $i(\bar{e}_R\gamma^\mu D^\nu e_R)(D_{(\mu}D_{\nu)}H^\dagger H)$ |
| $Q_{e^2H^2D^3}^{(8),2}$ | $i(\bar{e}_R\gamma^\mu D^\nu e_R)(H^\dagger D_{(\mu}D_{\nu)}H)$ |

Class 15: $\psi^2 X H^2 D$

| $Q_{e^2WH^2D}^{(8),1}$ | $(\bar{e}_R\gamma^\nu e_R)D^\mu(H^\dagger \tau^I H)W_{\mu\nu}^I$ |
|---|---|
| $Q_{e^2BH^2D}^{(8),1}$ | $(\bar{e}_R\gamma^\nu e_R)D^\mu(H^\dagger H)B_{\mu\nu}$ |
| $Q_{l^2WH^2D}^{(8),1}$ | $(\bar{l}\gamma^\nu l)D^\mu(H^\dagger \tau^I H)W_{\mu\nu}^I$ |
| $Q_{l^2WH^2D}^{(8),5}$ | $(\bar{l}\gamma^\nu \tau^I l)D^\mu(H^\dagger H)W_{\mu\nu}^I$ |
| $Q_{l^2WH^2D}^{(8),9}$ | $\epsilon^{IJK}(\bar{l}\gamma^\nu \tau^I l)D^\mu(H^\dagger \tau^J H)W_{\mu\nu}^K$ |
| $Q_{l^2BH^2D}^{(8),1}$ | $(\bar{l}\gamma^\nu \tau^I l)D^\mu(H^\dagger \tau^I H)B_{\mu\nu}$ |
| $Q_{l^2BH^2D}^{(8),5}$ | $(\bar{l}\gamma^\nu l)D^\mu(H^\dagger H)B_{\mu\nu}$ |

Table 2: Novel operator forms occuring first at dimension-eight which contribute to $H \to \bar{\ell}\ell\gamma$. The parenthesis in $D_{(\mu}D_{\nu)}$ indicate symmetrization in the Lorentz indices. $l$ is the left handed lepton $SU(2)_L$ doublet, while $e_R$ is the right handed leptonic singlet. Operators such as $(\bar{l}\gamma^\nu l)(H^\dagger \overleftrightarrow{D}^\mu H)B_{\mu\nu}$ do not contribute as the relative sign in the derivative removes the terms with a single Higgs. These operator forms are not yet implemented into the geoSMEFT framework.

in [33], the updated package is available in the Feynrules model database as well as in the ancillary files to this publication. The Class 11 operators are not included in this update as the additional derivatives of the Higgs doublet take an excessive amount of time to compute, they are available upon request.

All SM and dimension-six contributions receive corrections from the Class 3 operators which shift couplings related to the Higgs boson. The Class 4 operators mentioned above also shift all SM and dimension-six couplings of the vector bosons. These shifts occur due to finite field and mass renormalizations. These operators are:

$$
\begin{aligned}
\mathcal{L}_{\text{cl3}} &= c_{H\square}^{(6)}(H^\dagger H)\square(H^\dagger H) + c_{HD}^{(6)}(H^\dagger D^\mu H)^*(H^\dagger D_\mu H) \\
&+ c_{HD}^{(8)}(H^\dagger H)^2(D_\mu H^\dagger)(D^\mu H) + c_{HD,2}^{(8)}(H^\dagger H)(H^\dagger \sigma^a H)(D_\mu H^\dagger)\sigma^a(D^\mu H)
\end{aligned}
\tag{24}
$$

With the above, we have identified all operator forms contributing the process $H \to \bar{\ell}\ell\gamma$, without inducing chiral flips, at tree level in the SMEFT up to and including $1/\Lambda^4$ effects.

Below we discuss the potential of the decay $H \to \tau\tau\gamma$ may allow distinguishing operators which shift the SM tree coupling from the other operators effects. This discussion must be understood as having the potentially strong caveat that operators inducing chiral flips, have been neglected. As an estimate of the importance of such effects, we have included the Yukawa-like operators and the dimension-six dipole operators in the main text. In App F we discuss the size of the dipole contributions. We find that the interference of the dipole operators with other contributions is small compared with the dimension-six squared dipole contributions. We use this result as an argument to neglect the dimension-eight operators which induce chiral flips from our analysis. A more careful study of the viability of measurement in the tau decay-channel should be performed before such contributions are explored. The dipole operators we consider are given by:

$$
\mathcal{L}_{\text{cl6}} = c_{eW}^{(6)}(\bar{l}\sigma^{\mu\nu}e)\tau^I H W_{\mu\nu}^I + c_{eB}^{(6)}(\bar{l}\sigma^{\mu\nu}e)H B_{\mu\nu} + h.c.
\tag{25}
$$

We assume real Wilson coefficients for the dipole operators as their CP-violating parts are strongly constrained by low energy measurements [54].

In all cases of operators involving fermions we have assumed the flavor dependence is diagonal without invoking symmetry arguments or spurions. Since each decay channel has a separate flavor we choose to be agnostic about the universality of such couplings.

## 4.1  SMEFT amplitudes and Dalitz plots

With the full set of SMEFT Feynman rules in Appendix B we can identify all amplitudes with novel kinematics from those in the SM. We make the assumption that *at amplitude level*:

$$\frac{1}{16\pi^2}c_i^{(6)} \to 0 \tag{26}$$

This assumption "allows" us to neglect to include SMEFT $\frac{1}{\Lambda^2}$ contributions in the loop. Ultimately these contributions should be included, but are beyond the scope of this work. We emphasize that these terms are neglected at the amplitude level as they are still allowed to interfere with the SM loops. This neglects potential contributions from the Class 3 and 4 operators at one loop which could interfere with the tree level SMEFT amplitudes as in Figure 5(a). Such contributions are beyond the scope of this work and are left for future studies. While this affects our final numerical results, the contributions should be of the same magnitude as those included and therefore are not expected to greatly affect the qualitative discussions of Section 4.2.

The novel SMEFT kinematic forms can be categorized into the following cases:

1. $g_{H\gamma\gamma}$: Direct couplings of the Higgs boson to two photons as defined in Eq. B.3 due to Class 4 operators. These include the SM and shifts to the SM-like coupling of the photon to two leptons as defined in Eq. A.8 (implicit in the definition of $\bar{e}$). We note that $g_{H\gamma\gamma}$ mimics the one-loop coupling of $f_{H\gamma\gamma}^{(1)}$ without the full momentum dependence.

2. $g_{HZ\gamma} \times [(g_L + g_L')P_L + (g_R + g_R')P_R]$: Direct couplings of the Higgs boson to a $Z$ and $\gamma$ as defined in Eqs. B.3 and B.4 due to Class 4 operators. These include the SM and shifts to the SM-like coupling of the $Z$ to two leptons as defined in Eqs. A.8 and A.9. As above, $g_{HZ\gamma}$ mimics the one-loop coupling of $f_{HZ\gamma}^{(1)}$.

3. Contact $(A_{11}'P_L + B_{11}'P_R)$: the direct coupling $H\bar{\ell}\ell\gamma$ generated by Class 11 operators as in Eq. B.21.

4. Contact $(A_{15}P_L + B_{15}P_R)$: the direct coupling $H\bar{\ell}\ell\gamma$ generated by Class 15 operators as in Eq. B.21.

5. Shifted Yukawa couplings: The Yukawa couplings of the SM are shifted by the Class 5 operators.

6. Contact $c_{DP}$: direct coupling of $H\bar{\ell}\ell\gamma$ generated by the dipole operators

In what follows we refer to these cases as, for example, "Case 1" and "C1" in context.

The above cases generate novel kinematic forms for the amplitudes contributing to the decay of the Higgs boson to two leptons and a photon. Defining,

$$\Pi^{\mu\nu} = [k_1^\nu(k_2 + k_3)^\mu - k_1 \cdot (k_2 + k_3)\eta^{\mu\nu}] , \tag{27}$$

we can write the corresponding amplitudes as follows[5]:

$$i\mathcal{M}_{C1} = -i\frac{\bar{e}Q_\ell g_{HAA}v}{s_2}\Pi^{\mu\nu}(\bar{u}_{k_2}\gamma_\nu v_{k_3})\epsilon_\mu^*(k_1)\,, \tag{28}$$

$$\begin{aligned}
i\mathcal{M}_{C2} &\equiv \mathcal{M}_{C2}^L + \mathcal{M}_{C2}^R\,, \\
i\mathcal{M}_{C2}^L &= -i\frac{g_{HAZ}\bar{g}_Z(g_L+g_L')v}{4(s_2-\bar{m}_Z^2+i\bar{m}_Z\Gamma_Z)}\Pi^{\mu\nu}(\bar{u}_{k_2}\gamma_\nu P_L v_{k_3})\epsilon_\mu^*(k_1)\,, \\
i\mathcal{M}_{C2}^R &= -i\frac{g_{HAZ}\bar{g}_Z(g_R+g_R')v}{4(s_2-\bar{m}_Z^2+i\bar{m}_Z\Gamma_Z)}\Pi^{\mu\nu}(\bar{u}_{k_2}\gamma_\nu P_R v_{k_3})\epsilon_\mu^*(k_1)\,,
\end{aligned} \tag{29}$$

$$\begin{aligned}
i\mathcal{M}_{C3} &\equiv \mathcal{M}_{C3}^L + \mathcal{M}_{C3}^R\,, \\
i\mathcal{M}_{C3}^L &= \frac{iA_{11}'\bar{e}Q_\ell v}{2}k_1^\nu(k_2+k_3)^\mu(\bar{u}_{k_2}\gamma_\nu P_L v_{k_3})\epsilon_\mu^*(k_1)\,, \\
i\mathcal{M}_{C3}^R &= \frac{iB_{11}'\bar{e}Q_\ell v}{2}k_1^\nu(k_2+k_3)^\mu(\bar{u}_{k_2}\gamma_\nu P_R v_{k_3})\epsilon_\mu^*(k_1)\,,
\end{aligned} \tag{30}$$

$$\begin{aligned}
i\mathcal{M}_{C4} &\equiv \mathcal{M}_{C4}^L + \mathcal{M}_{C4}^R\,, \\
i\mathcal{M}_{C4}^L &= iA_{15}v\Pi^{\mu\nu}(\bar{u}_{k_2}\gamma_\nu P_L v_{k_3})\epsilon_\mu^*(k_1)\,, \\
i\mathcal{M}_{C4}^R &= iB_{15}v\Pi^{\mu\nu}(\bar{u}_{k_2}\gamma_\nu P_R v_{k_3})\epsilon_\mu^*(k_1)\,.
\end{aligned} \tag{31}$$

In addition to the above there are the contributions with operators which induce chiral flips:

$$i\mathcal{M}_{C5} = \frac{\bar{e}\hat{m}_\ell}{v}(1+\Delta_{H\bar{\ell}\ell})\,\bar{u}_{k_2}\gamma^\mu\slashed{k}_1\bar{v}_{k_3}\epsilon_\mu^*(k_1)\left[\frac{1}{s_1}-\frac{1}{s_3}\right]\,, \tag{32}$$

$$i\mathcal{M}_{C6} = c_{DP}\bar{u}_{k_2}\sigma^{\mu\nu}v_{k_3}k_1^\nu\epsilon_\mu^*(k_1)\,. \tag{33}$$

where $\Delta_{H\bar{\ell}\ell}$ is defined in Eq. B.2 of App B. We have also defined:

$$s_3 = (k_\gamma + k_{\bar{\psi}})^2 = \bar{m}_H^2 - s_1 - s_2\,. \tag{34}$$

Of these cases of amplitudes, only Cases 1, 2, 5, and 6 can generate results independent of the SM loop. For cases 1 and 2, this is because they can be generated at dimension-six, Cases 3 and 4 are first generated at $\mathcal{O}(1/\Lambda^4)$. For cases 5 and 6 this is because they can interfere with the SM tree amplitude. In Tables 6–7 we present Dalitz-like plots for the squares of the tree-level SMEFT contributions as well as the loop-tree interference between the SMEFT and the SM loops. These Dalitz plots are the ratio of the SMEFT contribution to the sum of the SM tree- and loop- contributions:

$$\begin{aligned}
R_{Ci}^{(0)} &= \frac{|\mathcal{M}_{Ci}|^2}{|\mathcal{M}_{SM}^{(0)}+\mathcal{M}_{SM}^{(1)}|^2} \\[2mm]
R_{CiCj^*}^{(0)} &= \frac{2\mathrm{Re}[\mathcal{M}_{Ci}\mathcal{M}_{Cj}^*]}{|\mathcal{M}_{SM}^{(0)}+\mathcal{M}_{SM}^{(1)}|^2} \\[2mm]
R_{Ci}^{(1)} &= \frac{2\mathrm{Re}[\mathcal{M}_{SM}^{(1)}\mathcal{M}_{Ci}^*]}{|\mathcal{M}_{SM}^{(0)}+\mathcal{M}_{SM}^{(1)}|^2}
\end{aligned} \tag{35}$$

---

[5]Here for convenience we take $k_\gamma = k_1$, $k_\psi = k_2$, and $k_{\bar{\psi}} = k_3$.

These plots are specifically for the case of the muon, but hold qualitatively for the other charged lepton flavors. The normalization of the plots is arbitrary as we have pulled out the dependence on the Wilson coefficients from the expressions (as indicated by the titles). Quantities such as $\bar{g}_Z$ and $\bar{e}$ are set to their SM values for simplicity.

The most striking feature of these plots is that in certain regions the overall sign of the ratio $R$ can flip. This sign flip is shown in the figures with a solid black line across the plot, the top most region is always chosen to have to a positive value (e.g. the plot of $R_{C1}^{(1)}$ has an explicit minus sign in its title to ensure the top region is $+$). These sign flips occur as these interference terms can generally have an arbitrary sign, as $s_2$ changes different SM-loop contributions are emphasized and can result in an overall sign change. While this theoretically could be used to define cuts in $s_2$ which emphasize the effects of these SMEFT contributions, in general one region's contribution is much larger than that in another. We do, however, explore this option in the analysis below.

These plots show separately the left and right-handed components as they can behave differently due to their interference with the left- and right-handed SM loops which differ (as discussed above). The most stark difference can be seen in the plots for $R_{C3}^{(1)}$ and $R_{C4}^{(1)}$ where the contribution from left handed leptons flips signs twice while for the right handed case there is only one sign flip.

## 4.2 Parameterized partial widths

Including all SMEFT corrections to the partial width of the Higgs boson decay to $\bar{\ell}\ell\gamma$ we have the following expression for the amplitude squared truncated at order $1/\Lambda^4$:

$$
\begin{aligned}
|\mathcal{M}|^2 &= |\mathcal{M}_{\mathrm{SM}}^{(0)}|^2(1 + 2\Delta_{H\bar{\ell}\ell} + [\Delta_{H\bar{\ell}\ell}^{(6)}]^2) + |\mathcal{M}_{\mathrm{C1}} + \mathcal{M}_{\mathrm{C2}} + \mathcal{M}_{\mathrm{C6}}|^2 \\
&+ \frac{1}{16\pi^2}2\mathrm{Re}\left[\mathcal{M}_{\mathrm{SM}}^{(1)}\left(\mathcal{M}_{\mathrm{C1}} + \mathcal{M}_{\mathrm{C2}} + \mathcal{M}_{\mathrm{C3}} + \mathcal{M}_{\mathrm{C4}} + \mathcal{M}_{\mathrm{C5}} + \mathcal{M}_{\mathrm{C6}}\right)^*\right] \quad (36)\\
&+ \frac{1}{(16\pi^2)^2}|\mathcal{M}_{\mathrm{SM}}^{(1)}|^2
\end{aligned}
$$

where $\Delta_{H\bar{\ell}\ell}^{(6)}$ is the dimension-six part of $\Delta_{H\bar{\ell}\ell}$ defined in B.2. Truncation at order $1/\Lambda^4$ of the square of the sum of amplitudes is implied. Writing the amplitude squared in this way emphasizes the enhancement of the SMEFT terms over the SM loops – we have explicitly written $1/(16\pi^2)$ to demonstrate that the SMEFT contributes to the amplitude-squared at tree- and one-loop level, where the SM has the chirally suppressed tree level contribution and the two-loop (one-loop squared) contribution.

In [8], the authors identify regions focused on optimizing sensitivity to the SM contributions. For example, from Fig. 4 we can see that the large $s_2$ region is dominated, for muons, by the tree-level contribution, cutting $s_2$ around the $Z$-pole region will isolate the resonant loop contributions, while cutting for lower $s_2$ will isolate the non-resonant loop contributions. The Dalitz-like plots of Figs. 6 and 7 show no obvious regions for isolating the SMEFT contributions due to novel kinematics not present in the SM. $R_{C1}$ and $R_{C2}$ mimic the effects of the resonant loops in the SM, and so can be isolated by looking in the corresponding SM kinematic regions for excesses. However, the sign changes coming from the interference with the SM loops present an interesting opportunity for combining different regions to emphasize the effects of the SMEFT. In Table 3 we define possible choices of cuts on $s_2$ and combinations of regions in $s_2$ to emphasize contributions in the SMEFT. The last column gives the

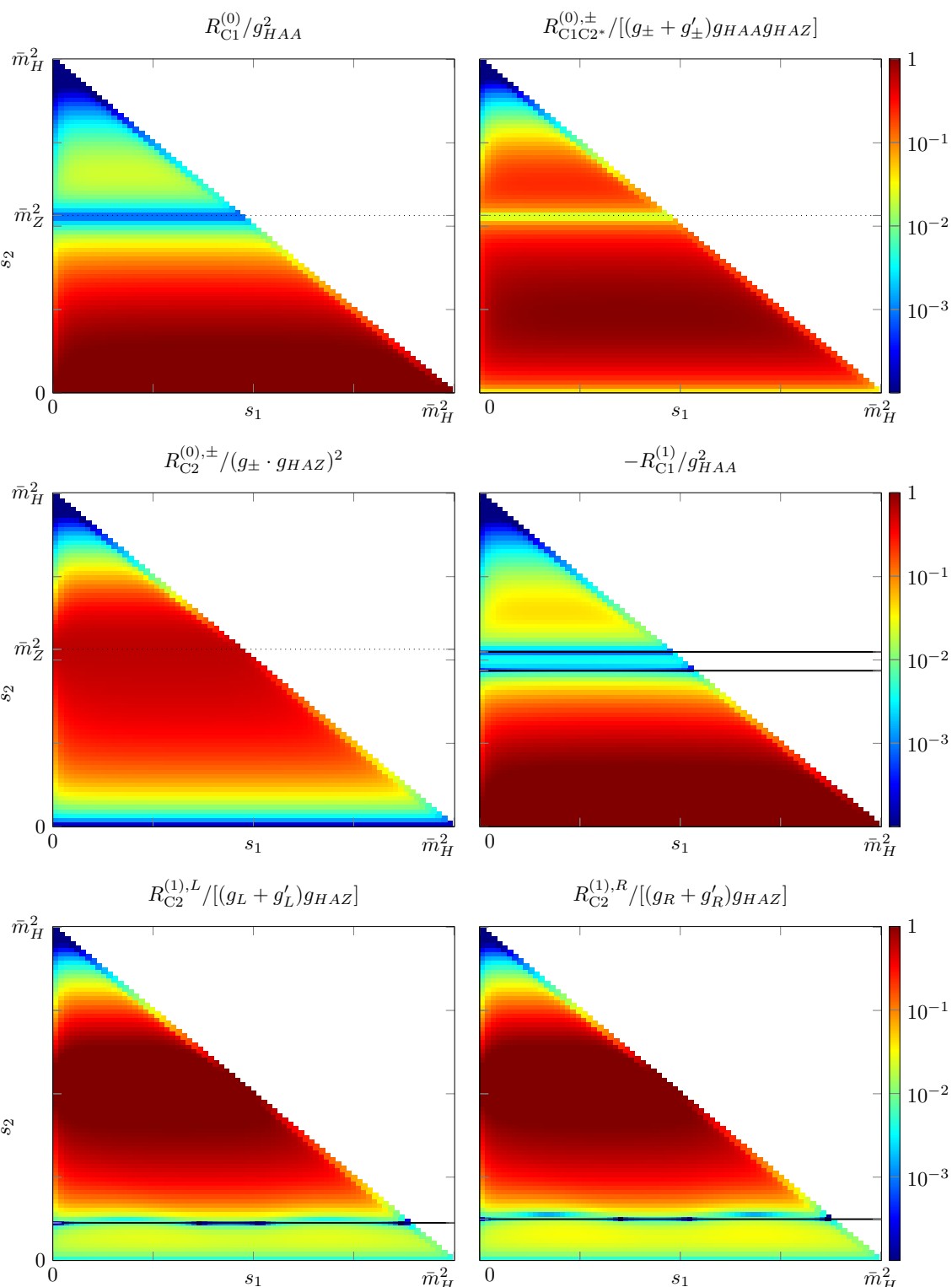

Figure 6: Dalitz-like plots showing the ratios defined in Eq. 35. Normalization is arbitrary as prefactors related to the Wilson coefficients have been pulled out of the ratios. In some cases a sign has been pulled out of the ratio as well. "±" is used when both the left and right hand parts are equal up to the normalizations factored out in the plot (as indicated by the title). Solid black lines show where the ratio flips signs, the top-most region is always positive.

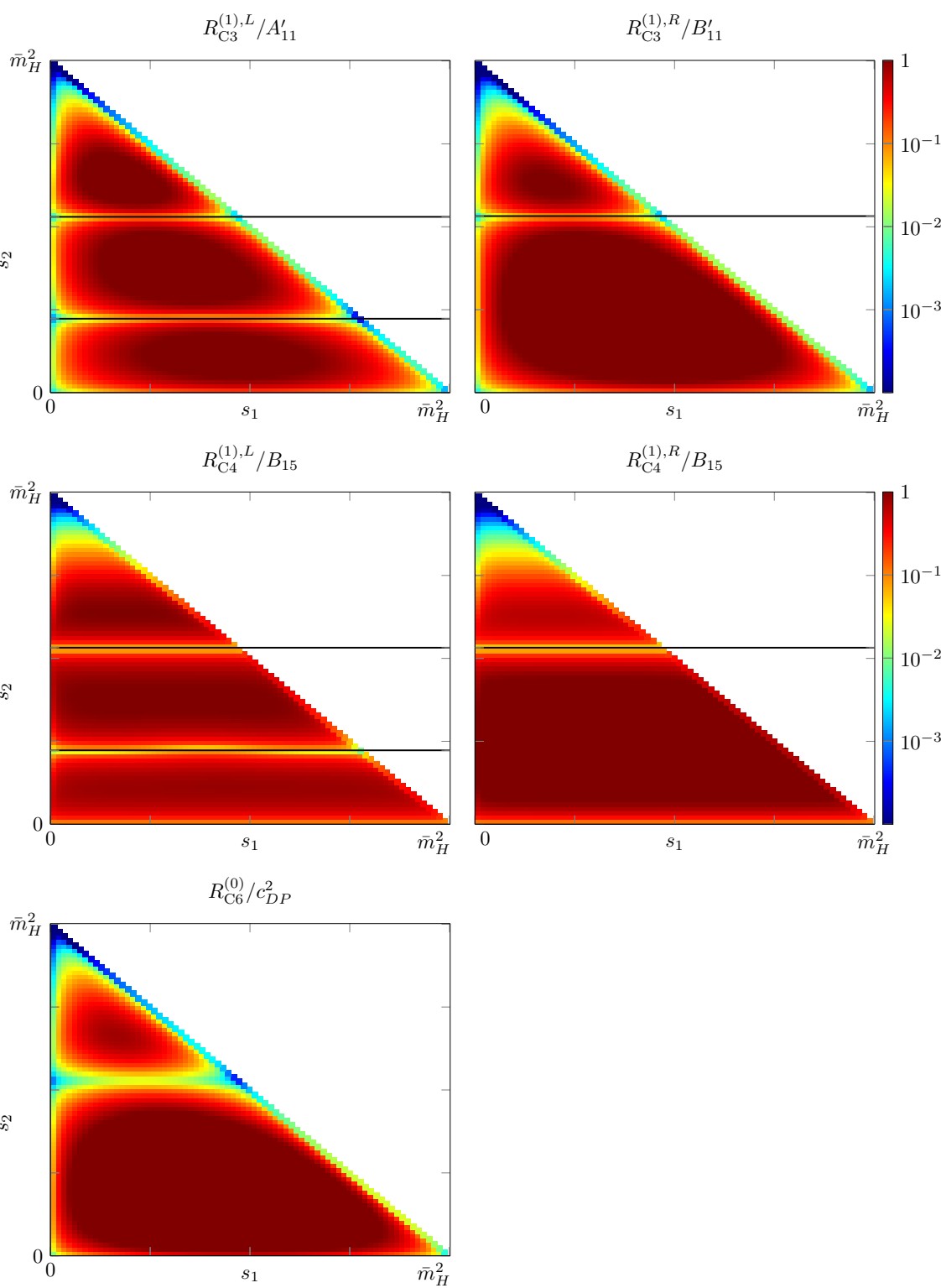

Figure 7: Dalitz-like plots as described in Figure 6.

| # | Region [GeV]$^2$ | Purpose | # SM |
|---|---|---|---|
| 1 | $(0 \leq s_2 \leq \bar{m}_H^2)$ | Full phase space/general | $22 \cdot 10^3$ |
| 2 | $(10^2 \leq s_2 \leq 40^2)$ | SM nonresonant & $R_{C1}^{(0,1)}$ | $2 \cdot 10^3$ |
| 3 | $(70^2 \leq s_2 \leq 100^2)$ | SM resonant & $R_{C2}^{(0,1)}$ & $R_{C3}^{(1)}$ | $10 \cdot 10^3$ |
| 4 | $(100^2 \leq s_2 \leq \bar{m}_H^2)$ | SM tree & $R_{C5}^{(0)}$ | $3 \cdot 10^3$ |
| 5 | $(45^2 \leq s_2 \leq 50^2) - (65^2 \leq s_2 \leq 80^2)$ | $R_{C3}^{(1),L}$ & $R_{C4}^{(1),L}$ | "$-$"900 |
| 6 | $(45^2 \leq s_2 \leq 50^2) + (65^2 \leq s_2 \leq 80^2)$ | control for $R_{C3}^{(1),L}$ & $R_{C4}^{(1),L}$ | 900 |

Table 3: Regions in $s_2$ chosen to emphasize various SMEFT effects. All regions should be understood to be limited to the physical phase space region, for example $\bar{m}_H^2$ may be outside the physical region and instead $s_2$ is integrated to the maximum value. The combined regions 5 and 6 are understood as the sum or difference between the integrated regions in $s_2$. $s_1$ is always integrated from the lowest to highest physical values. The first four regions approximately correspond to those proposed in [8]. The column labeled "# SM" is the approximate number of SM decays $H \to \mu\mu\gamma$ expected at HL-LHC with 3/ab integrated luminosity. The negative sign for the number of events in region 5 simply indicates that more decays occur in one region of the phase space than the other and we are taking the difference.

approximate number of decays $H \to \mu\mu\gamma$ expected for a SM Higgs at the HL-LHC given 3/ab of integrated luminosity. The number of decays is determines as the production cross section multiplied by the decay width (in the SM) multiplied by the integrated luminosity, $\sigma_{pp \to h} \times \mathrm{BR}_{\mathrm{SM}}(h \to \bar{\ell}\ell\gamma) \times 3/\mathrm{ab}$. In the first four regions the number is well over 1,000. In the case of Regions 5 and 6 the number is lower which could be problematic statistically – although the presence of SMEFT corrections could result in a higher number of events and be a strong indicator of NP.

To simplify the following expressions we adopt the following convention:

$$\tilde{c}_i^{(6)} = \hat{v}^2 c_i^{(6)}, \tag{37}$$

$$\tilde{c}_i^{(8)} = \hat{v}^4 c_i^{(8)}, \tag{38}$$

$$\hat{v}^2 = \frac{1}{\sqrt{2}\hat{G}_F}. \tag{39}$$

We also define the ratio of the SMEFT partial width to that of the SM in input parameter scheme $S$ over Region $R_i$ as defined in Table 3 as:

$$\Delta_S^{R_i} = \left.\frac{\Gamma_{H \to \bar{\mu}\mu\gamma}^{\mathrm{SMEFT}}}{\Gamma_{H \to \bar{\mu}\mu\gamma}^{\mathrm{SM}}}\right|_S^{R_i} - 1. \tag{40}$$

The relations between the various Lagrangian parameters and the two input parameter schemes can be derived from the Appendices of [34]. We emphasize that in determining the $\Delta$s we have used the full dependence of $\Gamma$ and not the qualitative normalization-independent quantities used to derive the Dalitz-like plots. These cuts are very simple and purely phenomenologically motivated. In future studies should be extended to include kinematic limitations of the experiments (e.g. photon isolation requirements, thresholds for various kinematic measurements). We have further neglected the experimental efficiency for muon detection in

the total number of events in the # SM column of Table 3, the efficiencies for electrons and taus has also been neglected in the discussion below.

Beginning with the fully integrated region (i.e. Region 1 of Tab. 3), we find the ratio of the partial width in the presence of the SMEFT to the SM value is:

$$
\begin{aligned}
\Delta_\alpha^{R_1} = {} & 0.47\tilde{c}_{H\Box}^{(6)} - 0.12\tilde{c}_{HD}^{(6)} - 0.059\tilde{c}_{HD}^{(8)} - 0.059\tilde{c}_{HD,2}^{(8)} + 0.94[\tilde{c}_{H\Box}^{(6)}]^2 - 0.47\tilde{c}_{H\Box}^{(6)}\tilde{c}_{HD}^{(6)} + 0.059[\tilde{c}_{HD}^{(6)}]^2 \\
& - 0.50\frac{\hat{v}}{\hat{m}_\mu}\tilde{c}_{eH}^{(6)} + 0.27\frac{\hat{v}^2}{\hat{m}_\mu^2}[\tilde{c}_{eH}^{(6)}]^2 - 0.25\frac{\hat{v}}{\hat{m}_\mu}\tilde{c}_{eH}^{(8)} - 0.50\frac{\hat{v}}{\hat{m}_\mu}\tilde{c}_{eH}^{(6)}\tilde{c}_{H\Box}^{(6)} + 0.13\frac{\hat{v}}{\hat{m}_\mu}\tilde{c}_{eH}^{(6)}\tilde{c}_{HD}^{(6)} \\
& + 0.0038\tilde{c}_{H\Box}^{(6)}\frac{\delta G_F}{\hat{G}_F} - 0.00096\tilde{c}_{HD}^{(6)}\frac{\delta G_F}{\hat{G}_F} - 0.0061\frac{\hat{v}}{\hat{m}_\mu}\tilde{c}_{eH}^{(6)}\frac{\delta G_F}{\hat{G}_F} + 9.4\cdot10^{-9}\frac{\delta G_F}{\hat{G}_F} \\
& + 10^4\left(5.6[\tilde{c}_{HB}^{(6)}]^2 + 1.8\tilde{c}_{HB}^{(6)}\tilde{c}_{HW}^{(6)} + 1.1[\tilde{c}_{HW}^{(6)}]^2 - 4.7\tilde{c}_{HB}^{(6)}\tilde{c}_{HWB}^{(6)} - 2.4\tilde{c}_{HW}^{(6)}\tilde{c}_{HWB}^{(6)} + 1.7[\tilde{c}_{HWB}^{(6)}]^2\right) \\
& + 10^{11}\left(2.4[\tilde{c}_{eB}^{(6)}]^2 - 2.7\tilde{c}_{eB}^{(6)}\tilde{c}_{eW}^{(6)} + 0.73[\tilde{c}_{eW}^{(6)}]^2\right) \\
& - 260\left(\tilde{c}_{HB}^{(6)} + \tilde{c}_{HB}^{(8)}\right) - 76\left(\tilde{c}_{HW}^{(6)} + \tilde{c}_{HW}^{(8)} + \tilde{c}_{HW,2}^{(8)}\right) + 140\left(\tilde{c}_{HWB}^{(6)} + \tilde{c}_{HWB}^{(8)}\right) \\
& - 520\left[\tilde{c}_{HB}^{(6)}\right]^2 - 150\left[\tilde{c}_{HW}^{(6)}\right]^2 + 16\left[\tilde{c}_{HWB}^{(6)}\right]^2 + 560\tilde{c}_{HB}^{(6)}\tilde{c}_{HWB}^{(6)} + 0.03\tilde{c}_{HW}^{(6)}\tilde{c}_{HWB}^{(6)} \\
& - 260\tilde{c}_{HB}^{(6)}\tilde{c}_{H\Box}^{(6)} + 120\tilde{c}_{HB}^{(6)}\tilde{c}_{HD}^{(6)} - 76\tilde{c}_{HW}^{(6)}\tilde{c}_{H\Box}^{(6)} - 41\tilde{c}_{HW}^{(6)}\tilde{c}_{HD}^{(6)} + 140\tilde{c}_{HWB}^{(6)}\tilde{c}_{H\Box}^{(6)} + 4.0\tilde{c}_{HWB}^{(6)}\tilde{c}_{HD}^{(6)} \\
& - 4.9\tilde{c}_{HB}^{(6)}\tilde{c}_{He}^{(6)} - 7.7\tilde{c}_{HB}^{(6)}\left(\tilde{c}_{Hl}^{(6),1} + \tilde{c}_{Hl}^{(6),3}\right) + 4.9\tilde{c}_{HW}^{(6)}\tilde{c}_{He}^{(6)} + 7.7\tilde{c}_{HW}^{(6)}\left(\tilde{c}_{Hl}^{(6),1} + \tilde{c}_{Hl}^{(6),3}\right) - 3.2\tilde{c}_{HWB}^{(6)}\tilde{c}_{He}^{(6)} \\
& - 4.9\tilde{c}_{HWB}^{(6)}\left(\tilde{c}_{Hl}^{(6),1} + \tilde{c}_{Hl}^{(6),3}\right) - 0.09\tilde{c}_{BH^4D^2}^{(8),1} + 0.025\tilde{c}_{WH^4D^2}^{(8),1} \\
& + 0.000016\tilde{c}_{HB}^{(6)}\frac{\delta G_F}{\hat{G}_F} - 0.000016\tilde{c}_{HW}^{(6)}\frac{\delta G_F}{\hat{G}_F} + 0.000010\tilde{c}_{HWB}^{(6)}\frac{\delta G_F}{\hat{G}_F} \\
& - 1.3\tilde{c}_{e^2BH^2D}^{(8),1} - 0.22\left(\tilde{c}_{e^2H^2D^3}^{(8),1} + \tilde{c}_{e^2H^2D^3}^{(8),2}\right) + 0.69\tilde{c}_{e^2WH^2D}^{(8),1} - 0.12\left(\tilde{c}_{L^2BH^2D}^{(8),1} + \tilde{c}_{L^2BH^2D}^{(8),5}\right) \\
& - 0.023\left(\tilde{c}_{l^2H^2D^3}^{(8),1} + \tilde{c}_{l^2H^2D^3}^{(8),2}\right) - 0.012\left(\tilde{c}_{l^2H^2D^3}^{(8),3} + \tilde{c}_{l^2H^2D^3}^{(8),4}\right) + 0.067\left(\tilde{c}_{l^2WH^2D}^{(8),1} + \tilde{c}_{l^2H^2D^3}^{(8),5}\right)
\end{aligned}
\tag{41}
$$

The lines of this expression are organized as follows: the first through third lines are the contribution from $\Delta_{H\bar{\ell}\ell}$, the fourth and fifth come from the square of the tree-level SMEFT contributions, and the subsequent lines are the interference of the SMEFT with the SM loops. All contributions are written only to two digits accuracy. Immediately we can see that the contribution from the square of the SMEFT tree contributions is by far the most dominant (note the factor of $10^4$, and $10^{11}$ pulled out from the expression). Interestingly, the dipole operators are dominant by several orders of magnitude and are generally neglected in the literature, including articles discussing NLO in the SMEFT expansion. From a purely bottom-up EFT interpretation we might argue all other terms are negligible, however making a flavor $U(3)^5$ assumption is sufficient for removing them (see, for example, the discussion of $U(3)^5$ symmetry in [38]). For the remainder of the text we will make this assumption, the dipole operator dependence is maintained in the full expressions in Appendix D and further dependence on the dipole operators which may be relevant to taus is discussed in App. F.

Further, the contributions coming from the shift in the definition of the fermi constant,

$$
v^2 = \frac{1}{\sqrt{2}\hat{G}_F} + \frac{\delta G_F}{\hat{G}_F}\,,
\tag{42}
$$

are strongly suppressed compared to other contributions. For this reason we drop them in

the remainder of the main text, they are also written in Appendix D. $\delta G_F$ at leading order is given by [55]:

$$\delta G_F = \sqrt{2}\tilde{c}_{Hl}^{(6),3} - \frac{\tilde{c}_{ll}^{(6)}}{\sqrt{2}} \, . \tag{43}$$

Where $c_{ll}^{(6)}$ is the four fermion operator affecting muon decay, $(\bar{L}L)^2$. It can be found to dimension-eight using the system of equations in the appendices of [34].

In the case of the contributions from $c_{eH}^{(d)}$ we could be concerned that these explode for the lighter generations as $\frac{v}{\hat{m}}$ will be large, however constraints on these Wilson coefficients (See e.g. [56]) constrain them for the muon and tau to be,

$$-\frac{14}{[\text{TeV}]^2} \leq c_{\mu H}^{(6)} \frac{v}{\hat{m}_\mu} \leq \frac{12}{[\text{TeV}]^2} \, , \qquad\qquad -\frac{2.5}{[\text{TeV}]^2} \leq c_{\tau H}^{(6)} \frac{v}{\hat{m}_\tau} \leq \frac{4.2}{[\text{TeV}]^2} \, , \tag{44}$$

as such we will neglect them in the following discussion, they are reproduced in Appendix D. Constraints on the Class 3 and 4 operators can also be found in the literature, however since $H \to \gamma\gamma$ and $H \to Z\gamma$ are used in inferring these constraints, they cannot be applied to this analysis. The couplings $c_{eH}^{(6)}$ can be constrained from the Higgs decays to two leptons independent of other channels. The coupling to the electron is not (directly) constrained by experiment, however we will still neglect the possibility of large changes to the electron yukawa due to $c_{eH}^{(6)}$ in the following discussion.

If we make no assumptions about the size of these Wilson coefficients the discussion above seemingly implies a unique opportunity to directly study the $\frac{1}{\Lambda^4}$ contributions from the SMEFT while the leading contributions at $\frac{1}{\Lambda^2}$ are strongly loop or chirally suppressed. We see that the contribution of the SMEFT at one loop and $\mathcal{O}(1/\Lambda^2)$ interfering with a tree level SMEFT amplitude would have a negligible effect compared to the tree-level squared SMEFT contributions in the fourth line above. This holds throughout the discussions below.

However, perturbativity in the SMEFT requires that, generally, higher order terms in $1/\Lambda^2$ should have smaller contributions to observables than the leading order terms. Here we understand the reason for the larger impact of the tree-level terms of the fourth and fifth lines is the loop suppression of the subsequent lines and the chiral suppression of the first lines. However, we can consider some scenarios in which the perturbativity of the expansion is more manifest:

1. NP Scale Assumption: Restoring the NP scale and the vev, i.e. $\tilde{c}_i^{(4+n)} \to \frac{\hat{v}^n}{\Lambda^n} c_i^{(4+n)}$, we can assume that the $c_i$ are order 1 and infer a scale at which the terms of the sixth line of Eq. 41 (i.e. linear in Class 4 operators) become larger than those of the fourth (quadratic in Class 4 operators, we neglect the dipole operators). Generally the contributions from the sixth line exceed those of the fourth before any other line under this assumption.

2. Loop Assumption: For weakly interacting new physics the Wilson coefficients $c_{HB}^{(6)}$, $c_{HW}^{(6)}$, and $c_{HWB}^{(6)}$ are generally expected to be generated at one loop in the UV [57][6]. These arguments do not hold for the Class 3 and contact operators, so we can use this assumption to infer if, for weakly interacting NP, the terms on the fourth line of Eq. 41 remain dominant. In this assumption all the $c_i$ are taken to be order 1 except for the class 4 operators which are taken to be order $1/(16\pi^2)$.

---

[6]It should be noted, however, that the logic typically used to categorize these Class 4 operators as loop generated [58] does not necessarily apply at dimension-eight [34]

3. Combined Assumption: We can combine the above two assumptions to infer a scale at which the Class 3 or contact operators become dominant. Generally with modest NP scales the Class 3 operators will dominate. Changing from muon final states to electrons can drive down the Class 3 operator contribution resulting in the dominance of the contact operators. The combined assumption leaves a significant amount of freedom to manipulate the outcome that it is arguably too model dependent for discussion. While we generally do not discuss it below, we do occasionally invoke flavor as a way to discriminate the contributions of different operator classes.

The above assumptions are very much model dependent. However, as noted above, if we make no assumptions the squares and interference of the tree level amplitudes of Eqs. 28 and 29 (Cases 1 and 2) dominate regardless of regions considered from Table 3.

For Region 1, under the NP scale assumption, for a NP scale of $\Lambda \sim 1$ TeV, the squares of the tree level C1 and C2 amplitudes are still dominant. This holds for $\Lambda$ up to around 3.5 TeV. Under the Loop Assumption the $[c_{HB}^{(6)}]^2$ term of the fourth line become roughly order $6 \cdot 10^4/(16\pi^2)^2 \sim 2$ while those linear in $c_{HB}^{(6)}$ are roughly $270/(16\pi^2) \sim 1.7$ so these terms are of roughly the same order. Class 4 operators still dominate the decay width. Considering the case of taus would, however, push the contributions from the first line beyond loop suppressed Class 4 operators. Considering both loop suppression and assuming a scale of NP the Class 4 operators cease to be dominant below an assumed scale of 1 TeV.

Region 2 is chosen to enhance the non-resonant SM loop contributions as well as the Case 1 amplitude's interference with the SM loop. We find the dominance of the tree-level SMEFT contributions is enhanced by at most a factor of 3. The contact operators have a larger impact as well as their contributions are concentrated in the low $s_2$ area of phase space. The full expression can be found in Appendix D, instead of giving the full expression for $\Delta_\alpha^{R_i}$ from this point in the text we will give the expressions under the UV assumptions presented above, keeping only the leading and subleading contributions for clarity.

If we again assume a scale associated with new physics as described above, we find that Class 4 operator contributions at order $\frac{1}{\Lambda^2}$ begin to dominate over these tree-level SMEFT contributions again at a scale of around $\Lambda \sim 3.5$ TeV:

$$
\begin{aligned}
\Delta_\alpha^{R_2} &= 3.4[c_{HB}^{(6)}]^2 + 2.0c_{HB}^{(6)}c_{HW}^{(6)} + 0.31[c_{HW}^{(6)}]^2 - 3.7c_{HB}^{(6)}c_{HWB}^{(6)} - 1.1c_{HW}^{(6)}c_{HWB}^{(6)} + 1.0[c_{HWB}^{(6)}]^2 \\
&\quad - 3.6c_{HB}^{(6)} - 1.1c_{HW}^{(6)} + 2.0c_{HWB}^{(6)}
\end{aligned}
\tag{45}
$$

If we retained the dipole operators, the scale at which the dimension-six operators would begin to be larger than the squares of the dipole contribution is over 100 TeV. Taking the loop assumption the contributions from the contact operators of Classes 11 and 15 are of greater importance than in the previous region, Class 4 operators are still dominant:

$$
\begin{aligned}
\Delta_\alpha^{R_2} &= 3.4[c_{HB}^{(6)}]^2 + 2.0c_{HB}^{(6)}c_{HW}^{(6)} + 0.31[c_{HW}^{(6)}]^2 - 3.7c_{HB}^{(6)}c_{HWB}^{(6)} - 1.1c_{HW}^{(6)}c_{HWB}^{(6)} + 1.0[c_{HWB}^{(6)}]^2 \\
&\quad - 4.6\left(\tilde{c}_{HB}^{(6)} + \tilde{c}_{HB}^{(8)}\right) - 1.4\left(\tilde{c}_{HW}^{(6)} + \tilde{c}_{HW}^{(8)} + \tilde{c}_{HW,2}^{(8)}\right) + 2.5\left(\tilde{c}_{HWB}^{(6)} + \tilde{c}_{HWB}^{(8)}\right) \\
&\quad - 3.1\tilde{c}_{e^2BH^2D}^{(8),1}
\end{aligned}
\tag{46}
$$

The enhancement from considering final state taus changes this such that class 3 operators are dominant.

Region 3 is chosen to enhance the resonant SM contributions as well as Case 2. However, we find this region, seemingly counterintuitively, results in a drop in the tree-level SMEFT

contributions of Cases 1 and 2. This is because in Regions 1 and 2 the phase space integral corresponding to the square of Case 1 (and the interference of C1 and C2) are two orders of magnitude larger than the contribution of Case 2. Region 3 successfully cuts out the dominant low energy part of the phase space integral dropping the contribution of the squares of the tree-level yukawa-like SMEFT contributions by an order of magnitude. As the interference of C1 and C2 has the opposite sign of the square of C1 some cancellation results in only a single order of magnitude drop.

For $R_3$, assuming a scale of new physics, the $\frac{1}{\Lambda^2}$ contributions begin to dominate over the tree level $\frac{1}{\Lambda^4}$ class 4 operator contributions for a scale of $\Lambda \sim 8$ TeV:

$$
\begin{aligned}
\Delta_\alpha^{R_3} \quad &= 0.013[c_{HB}^{(6)}]^2 - 0.022c_{HB}^{(6)}c_{HW}^{(6)} + 0.011[c_{HW}^{(6)}]^2 + 0.013c_{HB}^{(6)}c_{HWB}^{(6)} - 0.015\tilde{c}_{HW}^{(6)}\tilde{c}_{HWB}^{(6)} + 0.0049[\tilde{c}_{HWB}^{(6)}]^2 \\
&\quad -0.015c_{HB}^{(6)} + 0.0085\tilde{c}_{HW}^{(6)} - 0.0037\tilde{c}_{HWB}^{(6)}
\end{aligned}
\tag{47}
$$

If we again consider the possibility the Class 4 operators are loop generated then we find the contact operators become competitive with the class 4 operators:

$$
\begin{aligned}
\Delta_\alpha^{R_3} \quad &= 0.57[\tilde{c}_{HB}^{(6)}]^2 - 0.99\tilde{c}_{HB}^{(6)}\tilde{c}_{HW}^{(6)} + 0.51[\tilde{c}_{HW}^{(6)}]^2 + 0.59\tilde{c}_{HB}^{(6)}\tilde{c}_{HWB}^{(6)} - 0.66\tilde{c}_{HW}^{(6)}\tilde{c}_{HWB}^{(6)} + 0.22[\tilde{c}_{HWB}^{(6)}]^2 \\
&\quad -1.0\tilde{c}_{e^2BH^2D}^{(8),1} - 0.18\left(\tilde{c}_{e^2H^2D^3}^{(8),1} + \tilde{c}_{e^2H^2D^3}^{(8),2}\right) + 0.56\tilde{c}_{e^2WH^2D}^{(8),1} + 0.51\left(\tilde{c}_{L^2BH^2D}^{(8),1} + \tilde{c}_{L^2BH^2D}^{(8),5}\right) \\
&\quad +0.089\left(\tilde{c}_{l^2H^2D^3}^{(8),1} + \tilde{c}_{l^2H^2D^3}^{(8),2}\right) + 0.045\left(\tilde{c}_{l^2H^2D^3}^{(8),3} + \tilde{c}_{l^2H^2D^3}^{(8),4}\right) - 0.28\left(\tilde{c}_{l^2WH^2D}^{(8),1} + \tilde{c}_{l^2H^2D^3}^{(8),5}\right)
\end{aligned}
\tag{48}
$$

Region 4 is chosen to enhance the SM tree contributions and the corresponding shifts from $\Delta_{H\bar{\ell}\ell}$. In this region we fould that the SMEFT C1 and C2 tree contributions are strongly suppressed, but still largely dominant. Making assumptions about the scale of new physics as above, the linear terms in Class 4 operators begin to dominate for $\Lambda \sim 2.5$ TeV:

$$
\begin{aligned}
\Delta_\alpha^{R_4} \quad &= 0.046[c_{HB}^{(6)}]^2 - 0.055c_{HB}^{(6)}c_{HW}^{(6)} + 0.036[c_{HW}^{(6)}]^2 + 0.024c_{HB}^{(6)}c_{HWB}^{(6)} - 0.051c_{HW}^{(6)}c_{HWB}^{(6)} + 0.02[c_{HWB}^{(6)}]^2 \\
&\quad +0.044c_{HB}^{(6)} - 0.059c_{HW}^{(6)} + 0.042c_{HWB}^{(6)}
\end{aligned}
\tag{49}
$$

Further, should the Class 4 operators be loop suppressed this is the first region in which the Class 3 operators of Eq. 24 become dominant for the muon:

$$
\begin{aligned}
\Delta_\alpha^{R_4} \quad &= 1.9\tilde{c}_{H\square}^{(6)} - 0.49\tilde{c}_{HD}^{(6)} - 0.24\tilde{c}_{HD}^{(8)} - 0.24\tilde{c}_{HD,2}^{(8)} + 3.9[\tilde{c}_{H\square}^{(6)}]^2 - 1.9\tilde{c}_{H\square}^{(6)}\tilde{c}_{HD}^{(6)} + 0.24[\tilde{c}_{HD}^{(6)}]^2 \\
&\quad +0.22\tilde{c}_{e^2BH^2D}^{(8),1} - 0.12\tilde{c}_{e^2WH^2D}^{(8),1} - 0.42\left(\tilde{c}_{L^2BH^2D}^{(8),1} + \tilde{c}_{L^2BH^2D}^{(8),5}\right) + 0.23\left(\tilde{c}_{l^2WH^2D}^{(8),1} + \tilde{c}_{l^2H^2D^3}^{(8),5}\right)
\end{aligned}
\tag{50}
$$

While a subset of the contact operators are second-most dominant. The enhancement of the Class 3 operators will be improved by a factor of $m_\tau^2/m_\mu^2$ for the tau, and heavily suppressed for the electron in which case the contact operators become dominant.

Region 5 is chosen such that the phase space integral for the first subregion, $45^2 \leq s_2 \leq 50^2$, approximately cancels that of the second region, $65^2 \leq s_2 \leq 80^2$, for the tree level SMEFT contributions from Case 1. This has the effect of reducing the contributions from the Class 4 operators while enhancing the contribution from the left-handed contact operators (see Fig. 7).

Unfortunately the region only suppresses the tree-level Class 4 operator contributions by a factor of about 10. This is because for Cases 1, 2, and their interference, the regions cannot be chosen to simultaneously cancel all contributions. A very careful analysis could outperform the regions chosen here, but are beyond the scope of this article as they should

also take into account detector effects, such as the detector's ability resolve different flavors of charged leptons in these particular regions of phase space. While this region was defined with the intention of emphasizing the left-handed contact operators we also find enhanced contributions from the right-handed contact operators. This is simply because the dominant area of phase space for these operators is the low $s_2$ region.

The terms linear in Class 4 operators begin to dominate under an assumption of a new physics scale of about 3 TeV:

$$
\begin{aligned}
\Delta_\alpha^{R_5} &= 1.7[c_{HB}^{(6)}]^2 - 3.9c_{HB}^{(6)}c_{HW}^{(6)} + 1.6[c_{HW}^{(6)}]^2 + 2.6c_{HB}^{(6)}c_{HWB}^{(6)} - 1.8c_{HW}^{(6)}c_{HWB}^{(6)} + 0.48[c_{HWB}^{(6)}]^2 \\
&\quad + 2.3c_{HB}^{(6)} - 1.8c_{HW}^{(6)} + 1.0c_{HWB}^{(6)}
\end{aligned}
\tag{51}
$$

Under the assumption Class 4 operators are loop suppressed, the contact operators are dominant for final state electrons and muons:

$$
\begin{aligned}
\Delta_\alpha^{R_5} &= 0.95\left(\tilde{c}_{HB}^{(6)} + \tilde{c}_{HB}^{(8)}\right) - 0.76\left(\tilde{c}_{HW}^{(6)} + \tilde{c}_{HW}^{(8)} + \tilde{c}_{HW,2}^{(8)}\right) + 0.43\left(\tilde{c}_{HWB}^{(6)} + \tilde{c}_{HWB}^{(8)}\right) \\
&\quad - 9.6\tilde{c}_{e^2BH^2D}^{(8),1} - 1.7\left(\tilde{c}_{e^2H^2D^3}^{(8),1} + \tilde{c}_{e^2H^2D^3}^{(8),2}\right) + 5.3\tilde{c}_{e^2WH^2D}^{(8),1} + 6.4\left(\tilde{c}_{L^2BH^2D}^{(8),1} + \tilde{c}_{L^2BH^2D}^{(8),5}\right) \\
&\quad + 1.1\left(\tilde{c}_{l^2H^2D^3}^{(8),1} + \tilde{c}_{l^2H^2D^3}^{(8),2}\right) + 0.56\left(\tilde{c}_{l^2H^2D^3}^{(8),3} + \tilde{c}_{l^2H^2D^3}^{(8),4}\right) - 3.5\left(\tilde{c}_{l^2WH^2D}^{(8),1} + \tilde{c}_{l^2H^2D^3}^{(8),5}\right)
\end{aligned}
\tag{52}
$$

Comparing the Region 5 result with Region 6, which is just the sum of the two regions used for Region 5, we find that the tree-level contributions from Class 4 operators as much as a factor of 5 times larger than in the previously considered region. The calculation of the difference between regions in $R_5$ could mean the difference between resolving the impact of contact operators or only seeing the Class 4 operator contributions.

We have found that, invoking $U(3)^5$ symmetry, the Class 4 operator contributions are dominant in all defined regions unless we make certain assumptions. Assuming a scale associated with the NP allows us to infer a scale at which the $\frac{1}{\Lambda^2}$ contributions dominate over the contributions from the squares of Class 4 operators, the lowest scale for which this occurs is 2.5 TeV. This indicates (under the NP scale assumption) that the $H \to \ell\ell\gamma$ decay channel presents an excellent opportunity to directly study the $(1/\Lambda^2)^2$ terms in the SMEFT. Table 4 summarizes which Class of operator is dominant for a given region under the loop assumption for Class 4 operators. By considering the electron and muon flavors, these different regions present the opportunity to separately study the effects of operator Classes 3, 4, and the contact operators (Classes 11 and 15). Adding in the tau allows for clearer distinguishing in Region 3, but may be less promising when the instability of, and ability to resolve, taus is taken into account. In this way, Table 4 shows that this decay channel of the Higgs boson supplemented by various cuts provides an excellent opportunity to study the interplay between different operator classes as well as different orders in the SMEFT power counting. It is important to note that these statements all rely heavily on assumptions about the UV physics, and while these assumptions may be motivated, they may be misleading. By taking a more general approach, improving the quality of our predictions, and continuing to perform high quality global fits we will allow the physics to speak for itself and lead us to the correct conclusions.

|     | Cl4 operators | Class 3 operators | Class 11 & 15 operators | $\Lambda$ |
|-----|---------------|-------------------|-------------------------|-----------|
| R1  | D             | –                 | –                       | 3.5 TeV   |
| R2  | $D_{\mu,e}$   | $D_\tau$          | –                       | 3.5 TeV   |
| R3  | $C_{\mu,e}$   | $C_\mu$, $D_\tau$ | $C_{\mu,e}$             | 8 TeV     |
| R4  | –             | $D_\tau$          | $D_{e,\mu}$             | 2.5 TeV   |
| R5  | –             | $D_\tau$          | $D_{e,\mu}$             | 3 TeV     |

Table 4: Summary of the dominant operator contributions under the assumption the Class 4 operators are generated at one loop. $D_i$ indicates this operator class dominates for flavor $i$, $D$ indicates for all flavors. $C_i$ indicates that there is no dominant contribution and that the marked classes are of comparable sizes.

## 5 Conclusions

After laying out our conventions we derived both the SM tree and loop contributions to $H \to \bar{\ell}\ell\gamma$. We then derived the tree level results in the SMEFT up to and including terms of order $1/\Lambda^4$. Interfering the SMEFT results with the SM we created Dalitz-like plots demonstrating the behavior of these SMEFT contributions over the full phase space. By considering different regions of this phase space we were able to deduce regions in which various SMEFT contributions would be emphasized. The interference of SMEFT amplitudes with the SM were found to sometimes switch signs due to the different terms in the SM loops, which presented an interesting opportunity to sum regions of phase space in such a manner that the dominant contributions from the tree-level SMEFT amplitudes squared would be suppressed. In all cases, however, we found that these dominant terms prevailed.

If we make the assumption that the UV physics which generates the SMEFT in the IR is weakly interacting it is generally believed that the Class 4 operators (at dimension-six) will be suppressed by a factor of $\frac{1}{16\pi^2}$. In this case we have identified that the process $H \to \bar{\ell}\ell\gamma$, when different flavors of lepton are considered, can discriminate Class 4 operators from Class 3, and the contact operators of Classes 11 and 15. Given a measured deviation from the SM in this channel, this provides an opportunity to attempt to infer whether or not the UV physics is strongly or weakly interacting (i.e. the strength of the Class 4 operator contributions[7]) as well as to phenomenologically study the size of the $\frac{1}{\Lambda^4}$ terms.

The nature of the three body phase space makes three body decays an excellent opportunity for studying the novel kinematics of the SMEFT. The only other three body decays of the Higgs involve quarks, as such they present a challenge to precision measurements[8]. Another key aspect of this study was that the interplay between the tree and one-loop processes in the SM which cannot interfere. This combined with the fact the tree level contributions vary over many orders of magnitude depending on the flavor of the final state leptons gives a very broad phenomenology presenting many different ways to study the contributions from the SMEFT.

In performing this analysis we neglected the dimension-six one-loop terms and potential issues with detector resolution of the leptons and photon. As a result our quantitative results of Appendix D have room for improvement before and during the run of the HL-LHC. Our qualitative results are expected to approximately hold and therefore the conclusions of the

---

[7]It is important to note that if the Class 4 operators have small Wilson coefficients this does not necessarily preclude new physics which is strongly interacting, but is consistent with weakly interacting new physics.

[8]We neglect the narrow width approximation of, e.g. $H \to WW$ and $H \to ZZ$ in this statement [30].

ability of this channel to distinguish between different UV physics as well as to provide an opportunity to look for interesting flavor phenomenology remain.

## Acknowledgements

The authors thank I. Brivio, A. Martin, H. Patel, and M. Trott for useful discussions.

**Funding information** TC acknowledges funding from European Union's Horizon 2020 research and innovation program under the Marie Sklodowska-Curie grant agreement No. 890787.

# A  Feynman rules for the effective action

The Lagrangian corresponding to the effective action is given by Eq. 15, it is reproduced here:

$$
\begin{aligned}
\mathcal{L}_{1-\text{loop}} &= f^{(1)}_{H\gamma\gamma}(k_1, k_2)\hat{h}\hat{F}_{\mu\nu}\hat{F}^{\mu\nu} + f^{(2)}_{H\gamma\gamma}(k_1, k_2)\hat{h}(\partial^\mu \hat{F}_{\mu\nu})(\partial_\sigma \hat{F}^{\sigma\nu}) \\
&\quad + f^{(1)}_{HZ\gamma}(k_1, k_2)\hat{h}\hat{F}_{\mu\nu}\hat{Z}^{\mu\nu} + f^{(2)}_{HZ\gamma}(k_1, k_2)\hat{h}(\partial^\mu \hat{F}_{\mu\nu})\partial_\sigma\partial^\nu \hat{Z}^\sigma \quad\quad\quad\text{(A.1)}
\end{aligned}
$$

$$
+ f^{(3)}_{HZ\gamma}(k_1, k_2)\hat{h}\hat{F}_{\mu\nu}\Box\hat{Z}^\nu \,, \quad\quad\quad\quad\quad\quad\quad\quad\quad\quad\quad\quad\text{(A.2)}
$$

The rules follow `FeynRules` formatting, e.g. a field "$\phi$" with subscript "1" has corresponding four-momenta $k_1$ and if relevant Lorentz index $\mu_1$. The rule corresponding to the form factor $f^{(1)}_{H\gamma\gamma}$ is:

$$
\begin{pmatrix} \hat{h} \\ \hat{A}_1 \\ \hat{A}_2 \end{pmatrix} = 4if^{(1)}_{H\gamma\gamma}(k_1, k_2)\left[k_1^{\mu_2}k_2^{\mu_1} - k_1 \cdot k_2\, \eta^{\mu_1,\mu_2}\right] \quad\quad\quad\text{(A.3)}
$$

The rule corresponding to the form factor $f^{(2)}_{H\gamma\gamma}$ is:

$$
\begin{pmatrix} \hat{h} \\ \hat{A}_1 \\ \hat{A}_2 \end{pmatrix} = 2if^{(2)}_{H\gamma\gamma}(k_1, k_2)\left[k_1 \cdot k_2\, k_1^{\mu_1}k_2^{\mu_2} - k_1^2\, k_2^{\mu_1}k_2^{\mu_2} - k_2^2\, k_1^{\mu_1}k_1^{\mu_2} + k_1^2\, k_2^2\, \eta^{\mu_1,\mu_2}\right] \quad\text{(A.4)}
$$

For the operators corresponding to the $\hat{h}\hat{Z}\hat{\gamma}$ operators we have:

$$
\begin{pmatrix} \hat{h} \\ \hat{A}_1 \\ \hat{Z}_2 \end{pmatrix} = 2if^{(1)}_{HZ\gamma}(k_1, k_2)\left[k_1^{\mu_2}k_2^{\mu_1} - k_1 \cdot k_2\, \eta^{\mu_1\mu_2}\right] \quad\quad\quad\text{(A.5)}
$$

The Lorentz forms corresponding to $f^{(2)}_{H\gamma\gamma}$ are split between $f^{(2)}_{HZ\gamma}$ and $f^{(3)}_{HZ\gamma}$:

$$
\begin{pmatrix} \hat{h} \\ \hat{A}_1 \\ \hat{Z}_2 \end{pmatrix} = if^{(2)}_{HZ\gamma}(k_1, k_2)\left[k_1 \cdot k_2\, k_1^{\mu_1}k_2^{\mu_2} - k_1^2\, k_2^{\mu_1}k_2^{\mu_2}\right] \quad\quad\quad\text{(A.6)}
$$

$$
\begin{pmatrix} \hat{h} \\ \hat{A}_1 \\ \hat{Z}_2 \end{pmatrix} = if^{(3)}_{HZ\gamma}(k_1, k_2)\left[k_2^2\, k_1^{\mu_1}k_1^{\mu_2} - k_1^2\, k_2^2\, \eta^{\mu_1,\mu_2}\right] \quad\quad\quad\text{(A.7)}
$$

The following tree level Feynman rules are need to generate the process $H \to \ell\ell\gamma$ from the effective action:

$$
\begin{pmatrix} \bar{\ell} \\ \ell \\ \hat{A}_1 \end{pmatrix} = -i\bar{e}Q_\ell\gamma^{\mu_1} \quad\quad\quad\quad\quad\quad\quad\quad\quad\quad\quad\text{(A.8)}
$$

$$
\begin{pmatrix} \bar{\ell} \\ \ell \\ \hat{Z}_1 \end{pmatrix} = i\frac{\bar{g}_Z}{2}\gamma^{\mu_1}\left[(1 + 2Q_\ell s_Z^2)P_L + 2Q_\ell s_Z^2 P_R\right] \equiv i\frac{\bar{g}_Z}{2}\gamma^{\mu_1}(g_L P_L + g_R P_R) \quad\text{(A.9)}
$$

## B  Feynman rules in the SMEFT

This appendix follows the conventions set out in App. A. Note that there is implicit dependence on the Wilson coefficients of the SMEFT in Eqs. A.8 and A.9 (i.e. in $\bar{e}$ and $\bar{g}_Z$) which needs to be considered in addition to the rules outlined below. Shifts in the coupling of the Higgs boson to two fermions are given by:

$$
\begin{pmatrix} H \\ \bar{\ell} \\ \ell \end{pmatrix} = -i\frac{\hat{m}_\ell}{v}(1 + \Delta_{H\bar{\ell}\ell}) \tag{B.1}
$$

$$
\Delta_{H\bar{\ell}\ell} = v^2 \left[ c_{H\Box}^{(6)} - \frac{1}{4}c_{HD}^{(6)} - \frac{v^2}{8}(c_{HD}^{(8)} + c_{HD,2}^{(8)}) + \frac{3v^2}{32}(c_{HD}^{(6)} - 4c_{H\Box}^{(6)})^2 \right]
$$
$$
- \frac{3v^3}{2\sqrt{2}\hat{m}_\ell} \left( c_{eH}^{(6)} + \frac{v^2}{2}c_{eH}^{(8)} \right) \tag{B.2}
$$

We note in the above, and what follows $v$ is not rewritten in terms of the input parameter $\hat{G}_F$ and so the shift defined in Eq. 42 is implicit in these expressions. The direct coupling of the Higgs boson to $Z\gamma$ and $\gamma\gamma$ is given by:

$$
\begin{pmatrix} H \\ A_1 \\ A_2 \end{pmatrix} = ig_{HAA}v(k_1^{\mu_2}k_2^{\mu_1} - k_1 \cdot k_2\eta^{\mu_1\mu_2}) \tag{B.3}
$$

$$
\begin{pmatrix} H \\ A_1 \\ Z_2 \end{pmatrix} = -\frac{ig_{HAZ}v}{2}(k_1^{\mu_2}k_2^{\mu_1} - k_1 \cdot k_2\eta^{\mu_1\mu_2}) \tag{B.4}
$$

Where we have defined:

$$
g_{HAZ} = \left[ 8(c_{HB}^{(6)} - c_{HW}^{(6)})\bar{c}_W\bar{s}_W + 4c_{HWB}^{(6)}(\bar{c}_W^2 - \bar{s}_W^2) \right](1 + \Delta_{HAZ}v^2)
$$
$$
+ v^2 \left[ 8(c_{HB}^{(8)} - c_{HW}^{(8)} - c_{HW,2}^{(8)})\bar{c}_W\bar{s}_W + 4c_{HWB}^{(8)}(\bar{c}_W^2 - \bar{s}_W^2) \right] \tag{B.5}
$$

$$
+ \frac{\bar{e}(2c_{BH^4D^2}^{(8),1}\bar{c}_W - c_{WH^4D^2}^{(8),1}\bar{s}_W)}{8\bar{c}_W\bar{s}_W}v^2 \tag{B.6}
$$

$$
\Delta_{HAZ} = 2c_{HB}^{(6)} + 2c_{HW}^{(6)} + c_{H\Box}^{(6)} - \frac{1}{4}c_{HD}^{(6)} \tag{B.7}
$$

$$
g_{HAA} = 4 \left[ c_{HB}^{(6)}\bar{c}_W^2 + c_{HW}^{(6)}\bar{s}_W^2 - \bar{s}_W\bar{c}_W c_{HWB}^{(6)} \right]
$$
$$
+ v^2\bar{c}_W^2 \left[ c_{HB}^{(6)}(8c_{HB}^{(6)} + 4c_{H\Box}^{(6)} - c_{HD}^{(6)}) + 2(\bar{c}_{HWB}^{(6)})^2 \right]
$$
$$
+ v^2\bar{s}_W^2 \left[ c_{HW}^{(6)}(8c_{HW}^{(6)} + 4c_{H\Box}^{(6)} - c_{HD}^{(6)}) + 2(\bar{c}_{HWB}^{(6)})^2 \right]
$$
$$
- v^2\bar{c}_W\bar{s}_W c_{HWB}^{(6)} \left[ 8c_{HB}^{(6)} + 4c_{H\Box}^{(6)} - c_{HD}^{(6)} + 8c_{HW}^{(6)} \right]
$$
$$
+ 4v^2 \left[ c_{HB}^{(8)}\bar{c}_W^2 + (c_{HW}^{(8)} + c_{HW,2}^{(8)})\bar{s}_W^2 - \bar{s}_W\bar{c}_W c_{HWB}^{(8)} \right] \tag{B.8}
$$

In addition to the implicit SMEFT shifts in Eq. A.9, the $Z$ coupling to leptons is shifted by the Class 7 operators:

$$
\begin{pmatrix} \bar{\ell} \\ \ell \\ \hat{Z}_1 \end{pmatrix} = i\frac{\bar{g}_Z}{2}\gamma^{\mu_1}\left(g'_L P_L + g'_R P_R\right) \tag{B.9}
$$

$$
g'_L = v^2\left(c^{(6),1}_{Hl} + c^{(6),3}_{Hl}\right)(1 + \Delta_{Z\ell\ell}) + \frac{v^4}{2}\left(c^{(8),1}_{Hl} + c^{(8),2}_{Hl} + c^{(8),3}_{Hl}\right) \tag{B.10}
$$

$$
g'_R = v^2 c^{(6)}_{He}(1 + \Delta_{Z\ell\ell}) + \frac{v^4}{2}c^{(8)}_{He} \tag{B.11}
$$

$$
\Delta_{Z\ell\ell} = \frac{v^2}{2}\left[2\bar{c}_W g_2 c^{(6)}_{HW} + 2\bar{s}_W g_1 \bar{c}^{(6)}_{HB} + (g_2\bar{s}_W + g_1\bar{c}_W)c^{(6)}_{HWB}\right] \tag{B.12}
$$

The $Z$ coupling is also shifted by the Class 11 operators:

$$
\begin{pmatrix} \bar{\ell} \\ \ell_1 \\ \hat{Z}_2 \end{pmatrix} = i\frac{\bar{g}_Z}{8}v^2\left[A_{11}(k_1 \cdot k_2\gamma^{\mu_2} + k_1^{\mu_2}\slashed{k}_2)P_L + B_{11}(k_1 \cdot k_2\gamma^{\mu_2} + k_1^{\mu_2}\slashed{k}_2)P_R\right] \tag{B.13}
$$

$$
A_{11} = 2c^{(8),1}_{l^2H^2D^3} - 2c^{(8),2}_{l^2H^2D^3} + c^{(8),3}_{l^2H^2D^3} - c^{(8),4}_{l^2H^2D^3} \tag{B.14}
$$

$$
B_{11} = 2c^{(8),1}_{e^2H^2D^3} - 2c^{(8),2}_{e^2H^2D^3} \tag{B.15}
$$

However, these terms make no contribution to our a calculations as they occur at dimension-eight and no tree level coupling $HAZ$ exists in the SM.

For dipole operators we have the following rules:

$$
\begin{pmatrix} \bar{\ell} \\ \ell \\ \hat{A}_1 \\ \hat{H}_2 \end{pmatrix} = -ic_{\mathrm{DP}}\sigma^{\mu_1\nu}p_3^\nu \tag{B.16}
$$

$$
\begin{pmatrix} \bar{\ell} \\ \ell \\ \hat{A}_1 \end{pmatrix} = -ic'_{\mathrm{DP}}v\sigma^{\mu_1\nu}p_1^\nu \tag{B.17}
$$

$$
\begin{pmatrix} \bar{\ell} \\ \ell \\ \hat{Z}_1 \end{pmatrix} = -ic'_{\mathrm{DP2}}v\sigma^{\mu_1\nu}p_1^\nu \tag{B.18}
$$

$$
c_{\mathrm{DP}} = c'_{DP} = \frac{\bar{c}_W c_{eB,\ell} - \bar{s}_W c_{eW,\ell}}{2} \tag{B.19}
$$

$$
c'_{DP2} = \frac{\bar{s}_W c_{eB,\ell} + \bar{c}_W c_{eW,\ell}}{2} \tag{B.20}
$$

Finally, at dimension eight both Class 11 and Class 15 operators can generate the four-point

contact interaction:

$$
\begin{pmatrix} \bar{\ell} \\ \ell \\ \hat{A}_1 \\ \hat{H}_2 \end{pmatrix} = \quad i\frac{Q_\ell \bar{e} v}{2} k_2^{\mu_1} \slashed{k}_2 (A'_{11} P_L + B'_{11} P_R) \tag{B.21}
$$
$$
-iv\left[ A_{15}(k_1 \cdot k_2 \gamma^{\mu_1} - k_2^{\mu_1} \slashed{k}_1) P_L + B_{15}(k_1 \cdot k_2 \gamma^{\mu_1} - k_2^{\mu_1} \slashed{k}_1) P_R \right]
$$

$$
A'_{11} = \quad 2c^{(8),1}_{l^2 H^2 D^3} + 2c^{(8),2}_{l^2 H^2 D^3} + c^{(8),3}_{l^2 H^2 D^3} + c^{(8),4}_{l^2 H^2 D^3} \tag{B.22}
$$
$$
B'_{11} = \quad 2c^{(8),1}_{e^2 H^2 D^3} + 2c^{(8),2}_{e^2 H^2 D^3} \tag{B.23}
$$
$$
A_{15} = \quad \bar{c}_W (c^{(8),1}_{l^2 B H^2 D} + c^{(8),5}_{l^2 B H^2 D}) - \bar{s}_W (c^{(8),1}_{l^2 W H^2 D} + c^{(8),5}_{l^2 W H^2 D}) \tag{B.24}
$$
$$
B_{15} = \quad \bar{c}_W c^{(8),1}_{e^2 B H^2 D} - \bar{s}_W c^{(8),1}_{e^2 W H^2 D} \tag{B.25}
$$

## C   Matching to the effective action

Here we reproduce the matching expressions in the small off-shell momenta limit. Discussions in the main text generally use the full result where momenta are allowed to be arbitrarily off shell as appropriate to the process. The Higgs boson is taken on shell, and the momenta of photons, $k_i^2$ are taken to be small. We separate the bosonic (i.e. $W^\pm$, $\phi^\pm$, inclusive of ghosts $u^\pm$) contribution from the top contribution. Further, we expand in an appropriate small ratio of masses for comparison with [38, 39, 59] and to demonstrate the agreement between our and their expressions.

We leave the gauge parameter arbitrary, and note that the term of order $(k_i^2)^0$ terms are gauge invariant, as they must be, as they correspond to physical processes. Certain short hands are used for more complicated functions, such as the scalar Passarino-Veltman function $C_0^S$, these can be found after the full set of expressions. For the bosonic (including ghosts) contribution we find:

$$f_i^{(j)} = \left(f_i^{(j)}\Big|_{\rm B}\right) + \left(f_i^{(j)}\Big|_t\right) \tag{C.1}$$

$$\begin{aligned}
f_{H\gamma\gamma}^{(1)}\Big|_{\rm B} = & -\frac{\bar{e}^2\bar{m}_W^4}{16\pi^2\bar{m}_H^4 v}\left[\bar{m}_H^2(\bar{m}_H^2 + 6\bar{m}_W^2) - 3\bar{m}_W^2(\bar{m}_H^2 - 2\bar{m}_W^2)\log\left(1 + \frac{\bar{m}_H(\sqrt{\bar{m}_H^2 - 4\bar{m}_W^2} - \bar{m}_H)}{2\bar{m}_W^2}\right)^2\right] \\[2mm]
& -\frac{\bar{e}^2\bar{m}_W^4}{64\pi^2\bar{m}_H^6 v}(k_1^2 + k_2^2)\left[2(\bar{m}_H^4 + 6\bar{m}_H^2\bar{m}_W^2)\mathrm{Disc}[\bar{m}_H^2, \bar{m}_W, \bar{m}_W] + (\bar{m}_H^4 - 6\bar{m}_H^2\bar{m}_W^2 + 24\bar{m}_W^4)\log\left(1 + \frac{\bar{m}_H(\sqrt{\bar{m}_H^2 - 4\bar{m}_W^2} - \bar{m}_H)}{2\bar{m}_W^2}\right)^2\right. \\[2mm]
& \qquad\qquad - \bar{m}_H^2\left(48\bar{m}_W^2 - \xi(\bar{m}_H^2 - 2\bar{m}_W^2\xi)\log\left[1 + \frac{\bar{m}_H(\sqrt{\bar{m}_H^2 - 4\bar{m}_W^2} - \bar{m}_H)}{2\bar{m}_W^2}\right]^2\right) \\[2mm]
& \qquad\qquad + 2\bar{m}_H^2\left([\bar{m}_H^2 + \bar{m}_W^2(\xi - 1)]C_0^S[0, 0, \bar{m}_H^2, \sqrt{\xi}\bar{m}_W, \bar{m}_W, \bar{m}_W]\right) \\[2mm]
& \qquad\qquad \left. + (\bar{m}_W^2[\xi - 1] - \bar{m}_H^2)\xi C_0^S[0, 0, \bar{m}_H^2, \sqrt{\xi}\bar{m}_W, \sqrt{\xi}\bar{m}_W, \bar{m}_W]\right] + \mathcal{O}(k_i^4) \tag{C.2} \\[3mm]
= & \frac{\bar{e}^2}{8\pi^2 v}\left(-\frac{7}{4} - \frac{11}{30}a - \frac{19}{105}a^2 - \frac{58}{525}a^3 - \cdots\right) + \mathcal{O}(k_i^2) \qquad a \equiv \frac{\bar{m}_H^2}{4\bar{m}_W^2} \tag{C.3}
\end{aligned}$$

In the case of the operator corresponding to $f_{H\gamma\gamma}^{(2)}$ the gauge dependence does not vanish for the leading term, this is because this term only contributes for off-shell photons. This can be understood from Eq. A.4 where the first three terms are proportional to $k_i^{\mu_i}$ which vanishes when contracted with the on-shell polarization vector, while the $\eta^{\mu_1,\mu_2}$ term is preceded by $k_i^2$ which vanishes for on-shell photons.

$$
\left. f_{H\gamma\gamma}^{(2)}\right|_B = \frac{\bar{e}^2}{16\pi^2 \bar{m}_H^6 v}\left[ -40\bar{m}_H^2(\bar{m}_H^2 + 6\bar{m}_W^2) + \bar{m}_H^2\left(\bar{m}_W^2(\xi-9)(\xi-1)^2 - 2\bar{m}_H^2[\xi(\xi-5)-4]\right)\log\left[\frac{1}{\xi}\right]\right.
$$

(C.4)

$$
+ \left(2\bar{m}_H^4 - \frac{\bar{m}_H^6}{\bar{m}_W^2} - 20\bar{m}_H^2\bar{m}_W^2 - 48\bar{m}_W^4\right)\log\left(1 + \frac{\bar{m}_H(\sqrt{\bar{m}_H^2 - 4\bar{m}_W^2} - \bar{m}_H)}{2\bar{m}_W^2}\right)
$$

$$
+ \frac{1}{\bar{m}_W^2}\left[\bar{m}_H^6(\xi-1) + 2\bar{m}_H^4\bar{m}_W^2(\xi-9) - 96\bar{m}_H^2\bar{m}_W^4\right]\operatorname{Disc}[\bar{m}_H^2, \bar{m}_W, \bar{m}_W]
$$

$$
+ \frac{\bar{m}_H^4}{\bar{m}_W^2}(\xi-1)\left((\bar{m}_H^2 - 2\bar{m}_W^2\xi)\operatorname{Disc}[\bar{m}_H^2, \sqrt{\xi}\bar{m}_W, \sqrt{\xi}\bar{m}_W] - 2(\bar{m}_H^2 + (\xi-1)\bar{m}_W^2)\operatorname{Disc}[\bar{m}_H^2, \bar{m}_W, \sqrt{\xi}\bar{m}_W]\right)
$$

$$
+ \bar{m}_H^4\left[2\bar{m}_W^2(\xi-1)^2 + \bar{m}_H^2(\xi+1)^2\right]C_0^S[0, 0\bar{m}_H^2, \bar{m}_W, \sqrt{\xi}\bar{m}_W, \bar{m}_W]
$$

$$
+ \frac{4\bar{m}_H^2}{\bar{m}_W^2}\left[\bar{m}_H^6 + 2\bar{m}_W^6(\xi-1)^3 - \bar{m}_H^4\bar{m}_W^2(\xi+1) + \bar{m}_H^2\bar{m}_W^4(1 + (2-3\xi)\xi)\right]C_0^S[0, 0\bar{m}_H^2, \sqrt{\xi}\bar{m}_W, \bar{m}_W, \bar{m}_W]
$$

$$
- \frac{\bar{m}_H^4}{\bar{m}_W^2}(2\bar{m}_H^2 + \bar{m}_W^2(\xi-1)^2)(\bar{m}_H^2 - 2\bar{m}_W^2\xi)C_0^S[0, 0, \bar{m}_H^2, \sqrt{\xi}\bar{m}_W, \bar{m}_W, \sqrt{\xi}\bar{m}_W]
$$

$$
\left. + 4\bar{m}_H^2[\bar{m}_H^2 + 2\bar{m}_W^2(\xi-1)][\bar{m}_W^2(\xi-1)^2 - \bar{m}_H^2(\xi+1)]C_0^S[0, 0, \bar{m}_H^2, \sqrt{\xi}\bar{m}_W, \sqrt{\xi}\bar{m}_W, \bar{m}_W]\right] + \mathcal{O}(k_i^2)
$$

(C.5)

While for the top quark contributions we have:

$$
\begin{aligned}
f^{(1)}_{H\gamma\gamma}\Big|_{\rm t} &= \frac{Q_u^2 \bar{e}^2 N_c}{4\pi^2 \bar{m}_H^4 v}\left[4\bar{m}_t^2\bar{m}_H^2 - (\bar{m}_H^2\bar{m}_t^2 - 4\bar{m}_t^4)\log\left(\frac{2\bar{m}_t^2 - \bar{m}_H^2 + \sqrt{\bar{m}_H^4 - 4\bar{m}_H^2\bar{m}_t^2}}{2\bar{m}_t^2}\right)^2\right. \\
&\left. - \frac{k_1^2 + k_2^2}{\bar{m}_H^2}\left((\bar{m}_H^4 - 16\bar{m}_H^2\bar{m}_t^2 - 4\bar{m}_H^2\bar{m}_t^2\mathrm{Disc}[\bar{m}_H^2, m_t, m_t] + \bar{m}_t^2(\bar{m}_H^2 - 8\bar{m}_t^2)\log\left[\frac{2\bar{m}_t^2 - \bar{m}_H^2 + \sqrt{\bar{m}_H^4 - 4\bar{m}_H^2\bar{m}_t^2}}{2\bar{m}_t^2}\right]^2\right)\right] + \mathcal{O}(k_i^4) \quad (\mathrm{C.6}) \\
&= \frac{Q_u^2 \bar{e}^2 N_c}{8\pi^2 v}\left[\frac{1}{3} + \frac{7}{90}a + \cdots\right] + \mathcal{O}(k_i^2) \qquad a \equiv \frac{\bar{m}_H^2}{4\bar{m}_t^2} \quad\quad (\mathrm{C.7})
\end{aligned}
$$

$$
\begin{aligned}
f^{(2)}_{H\gamma\gamma}\Big|_{\rm t} &= \frac{Q_u^2 \bar{e}^2 N_c}{2\pi^2 \bar{m}_H^6 v}\left[20\bar{m}_H^2\bar{m}_t^2 + 8\bar{m}_t^2\bar{m}_H^2\mathrm{Disc}[\bar{m}_H^2, \bar{m}_t, \bar{m}_t] + \bar{m}_t^2(\bar{m}_H^2 + 4\bar{m}_t^2)\log\left(\frac{2\bar{m}_t^2 - \bar{m}_H^2 + \sqrt{\bar{m}_H^4 - 4\bar{m}_H^2\bar{m}_t^2}}{2\bar{m}_t^2}\right)^2\right. \\
&\left. + \frac{k_1^2 + k_2^2}{3\bar{m}_H^2}\left(\bar{m}_H^4 + 204\bar{m}_H^2\bar{m}_t^2 + 84\bar{m}_t^2\bar{m}_H^2\mathrm{Disc}[\bar{m}_H^2, \bar{m}_t, \bar{m}_t] + 12(\bar{m}_t^2\bar{m}_H^2 + 3\bar{m}_t^4)\log\left[\frac{2\bar{m}_t^2 - \bar{m}_H^2 + \sqrt{\bar{m}_h^4 - 4\bar{m}_H^2\bar{m}_t^2}}{2\bar{m}_t^2}\right]\right)\right] \quad (\mathrm{C.8})
\end{aligned}
$$

The form factors for the $\hat{h}\hat{Z}\hat{\gamma}$ couplings are again expanded in terms of small off-shell momenta, in this case we take the gauge parameter $\xi \to 1$ and expand in small off-shell momenta. For the $Z$ we take $k_2^2 \to \bar{m}_Z^2 + \mu^2$, where $\mu^2$ represents the small perturbation from an on-shell $Z$:

$$N_1 = -\frac{2\bar{e}^2 t_W}{16\pi^2(\bar{m}_H^2 - \bar{m}_Z^2)^2(\bar{m}_Z^2 - \bar{m}_W^2)v} \tag{C.9}$$

$$
\begin{aligned}
f^{(1)}_{HZ\gamma}\Big|_{\mathrm{B}} = N_1 &\left[ (\bar{m}_H^2 - \bar{m}_Z^2) \left[ 2\bar{m}_W^2(\bar{m}_H^2 + 6\bar{m}_W^2) - (\bar{m}_H^2 + 2\bar{m}_W^2)\bar{m}_Z^2 \right] + \bar{m}_Z^2 \left[ 2\bar{m}_W^2(\bar{m}_H^2 + 6\bar{m}_W^2) - (\bar{m}_H^2 + 2\bar{m}_W^2)\bar{m}_Z^2 \right] \mathrm{Disc}[\bar{m}_H^2, \bar{m}_W, \bar{m}_W] \right.\\[2mm]
&+ \bar{m}_Z^2 \left[ (\bar{m}_H^2 + 2\bar{m}_W^2)\bar{m}_Z^2 - 2\bar{m}_W^2(\bar{m}_H^2 + 6\bar{m}_W^2) \right] \mathrm{Disc}[\bar{m}_Z^2, \bar{m}_W, \bar{m}_W] \\[2mm]
&\left. + \bar{m}_W^2 \left[ 12\bar{m}_W^4 + 6\bar{m}_W^2\bar{m}_Z^2 - 2\bar{m}_Z^4 + \bar{m}_H^2(\bar{m}_Z^2 - 6\bar{m}_W^2) \right] \left( \log\left[ 1 + \frac{\bar{m}_H(\sqrt{\bar{m}_H^2 - 4\bar{m}_W^2} - \bar{m}_H)}{2\bar{m}_W^2} \right] + \log\left[ 1 + \frac{\bar{m}_Z(\sqrt{\bar{m}_Z^2 - 4\bar{m}_W^2} - \bar{m}_Z)}{2\bar{m}_W^2} \right] \right) \right]
\end{aligned}
\tag{C.10}
$$

$$+ \mathcal{O}(k_i^2, \mu^2)$$

$$= \frac{\bar{e}^2 t_W}{3 \cdot 16\pi^2 v} \left[ \left( 11 - \frac{31}{t_W^2} \right) + \frac{22}{3}\left( \frac{1}{5} - \frac{1}{t_W^2} \right) a + \frac{8}{3}\left( \frac{7}{5} - \frac{4}{t_W^2} \right) b + \mathcal{O}(a^2, b^2) \right] + \mathcal{O}(k_i^2, \mu^2), \quad a \equiv \frac{\bar{m}_H^2}{4\bar{m}_W^2}, \quad b \equiv \frac{\bar{m}_Z^2}{4\bar{m}_W^2}. \tag{C.11}$$

In the above, the overall normalization was chosen to match that of [39].

$$N_2 = \frac{\bar{e}^2}{16\pi^2(\bar{m}_H - \bar{m}_Z)^3(\bar{m}_H + \bar{m}_Z)^3\sqrt{\bar{m}_W^2(\bar{m}_Z^2 - \bar{m}_W^2)}v} \tag{C.12}$$

$$
\begin{aligned}
f_{HZ\gamma}^{(2)}\Big|_{\text{B}} = N_2 \Bigg[ & 12(\bar{m}_H^2 - \bar{m}_Z^2)[2\bar{m}_W^2(\bar{m}_H^2 + 6\bar{m}_W^2) - \bar{m}_Z^2(\bar{m}_H^2 + 2\bar{m}_W^2)] \\[4pt]
& - 4(2\bar{m}_H^2 + \bar{m}_Z^2)[\bar{m}_Z^2(\bar{m}_H^2 + 2\bar{m}_W^2) - 2\bar{m}_W^2(\bar{m}_H^2 + 6\bar{m}_W^2)]\text{Disc}[\bar{m}_H^2, \bar{m}_W, \bar{m}_W] \\
& + 4(\bar{m}_H^2 + 2\bar{m}_Z^2)[\bar{m}_Z^2(\bar{m}_H^2 + 2\bar{m}_W^2) - 2\bar{m}_W^2(\bar{m}_H^2 + 6\bar{m}_W^2)]\text{Disc}[\bar{m}_Z^2, \bar{m}_W, \bar{m}_W] \\
& - \big[6\bar{m}_W^2\bar{m}_Z^4 + \bar{m}_H^4(2\bar{m}_W^2 + \bar{m}_Z^2) + \bar{m}_H^2(\bar{m}_Z^4 - 20\bar{m}_W^4 - 4\bar{m}_W^2\bar{m}_Z^2) - 48\bar{m}_W^6 - 4\bar{m}_W^4\bar{m}_Z^2\big] \\
& \times \log\left(\frac{2\bar{m}_W^2 + \bar{m}_H(\sqrt{\bar{m}_H^2 - 4\bar{m}_W^2} - \bar{m}_H)}{2\bar{m}_W^2 + \bar{m}_Z(\sqrt{\bar{m}_Z^2 - 4\bar{m}_W^2} - \bar{m}_Z)}\right)\left(\log\left[1 + \frac{\bar{m}_H(\sqrt{\bar{m}_H^2 - 4\bar{m}_W^2} - \bar{m}_H)}{2\bar{m}_W^2}\right] + \log\left[1 + \frac{\bar{m}_Z(\sqrt{\bar{m}_Z^2 - 4\bar{m}_W^2} - \bar{m}_Z)}{2\bar{m}_W^2}\right]\right) \Bigg] \\
& + \mathcal{O}(k_i^2, \mu^2)
\end{aligned}
\tag{C.13}
$$

$$N_3 = \frac{\bar{e}^2[2\bar{m}_W^2(\bar{m}_H^2 + 6\bar{m}_W^2) - (\bar{m}_H^2 + 2\bar{m}_W^2)\bar{m}_Z^2]}{16\pi^2\bar{m}_W(\bar{m}_Z^2 - \bar{m}_H^2)^3\sqrt{\bar{m}_Z^2 - \bar{m}_W^2}v} \tag{C.14}$$

$$
\begin{aligned}
f_{HZ\gamma}^{(3)}\Big|_{\text{B}} = N_3 \Bigg[ & 12(\bar{m}_H^2 - \bar{m}_Z^2) + 4(2\bar{m}_H^2 + \bar{m}_Z^2)\text{Disc}[\bar{m}_H^2, \bar{m}_W, \bar{m}_W] - 4(\bar{m}_H^2 + \bar{m}_Z^2)\text{Disc}[\bar{m}_Z^2, \bar{m}_W, \bar{m}_W] \\[4pt]
& + (\bar{m}_H^2 + 4\bar{m}_W^2 + \bar{m}_Z^2)\left(\log\left[1 + \frac{\bar{m}_H(\sqrt{\bar{m}_H^2 - 4\bar{m}_W^2} - \bar{m}_H)}{2\bar{m}_W^2}\right] - \log\left[1 + \frac{\bar{m}_Z(\sqrt{\bar{m}_Z^2 - 4\bar{m}_W^2} - \bar{m}_Z)}{2\bar{m}_W^2}\right]\right) \Bigg] \\
& + \mathcal{O}(k_i^2, \mu^2)
\end{aligned}
\tag{C.15}
$$

For the top quark contribution we have:

$$N_4 = \frac{\bar{e}\bar{g}_Z(g_L^t + g_R^t)\bar{m}_t^2 N_c Q_u}{32\pi^2 v(\bar{m}_H^2 - \bar{m}_Z^2)^2} \qquad g_L^t = (1 - 2Q_u s_z^2) \qquad g_R^t = -2Q_u s_Z^2 \tag{C.17}$$

$$f_{HZ\gamma}^{(1)}\Big|_t = N_4 \Big[ 4(\bar{m}_H^2 - \bar{m}_Z^2) + 4\bar{m}_Z^2 \left( \text{Disc}[\bar{m}_H^2, \bar{m}_t, \bar{m}_t] - \text{Disc}[\bar{m}_Z^2, \bar{m}_t, \bar{m}_t] \right)$$

$$+ (\bar{m}_H^2 - 4\bar{m}_t^2 - \bar{m}_Z^2) \left( \log \left[ \frac{2\bar{m}_t^2 - \bar{m}_Z^2 + \sqrt{\bar{m}_Z^4 - 4\bar{m}_Z^2\bar{m}_t^2}}{2\bar{m}_t^2} \right]^2 - \log \left[ \frac{2\bar{m}_t^2 - \bar{m}_h^2 + \sqrt{\bar{m}_H^4 - 4\bar{m}_H^2\bar{m}_t^2}}{2\bar{m}_t^2} \right]^2 \right) \Big] + \mathcal{O}(k_i^2, \mu^2) \tag{C.18}$$

$$= \frac{\bar{e}^2}{4\pi^2} \frac{N_c Q_u (3 - 8\bar{s}_W^2)}{12\bar{s}_W \bar{c}_W v} \left[ \frac{1}{3} + \frac{7}{90}a + \frac{11}{90}b + \mathcal{O}(a^2, b^2) \right] + \mathcal{O}(k_i^2, \mu^2) \qquad a \equiv \frac{\bar{m}_H^2}{4\bar{m}_t^2} \qquad b \equiv \frac{\bar{m}_Z^2}{4\bar{m}_t^2} \tag{C.19}$$

Below are the special functions used in the above expressions. Not all $C_0^S$ functions are listed as the expressions become complicated and depend on the relation between $b$ and $c$ which is generally a function of the gauge parameter $\xi$. The unlisted $C_0^S$ functions can be obtained from `Package-X` [49].

$$\text{Disc}[a^2, b, c] = \frac{\sqrt{\lambda[a^2, b^2, c^2]}}{a^2} \log \left( \frac{b^2 + c^2 - a^2 + \sqrt{\lambda[a^2 + b^2 + c^2]}}{2bc} \right) \tag{C.20}$$

$$\lambda[a^2, b^2, c^2] = a^4 + b^4 + c^4 - 2a^2b^2 - 2a^2c^2 - 2b^2c^2 \tag{C.21}$$

$$C_0^S[0, 0, a^2, b, c, c] = \log \left( 1 - \frac{2a^2}{a^2 + b^2 - c^2 + \sqrt{\lambda[a^2, b^2, c^2]}} \right)^2 + 2\text{Li}_2 \left( 1 - \frac{b^2}{c^2} \right) \tag{C.22}$$

$$- 2\text{Li}_2 \left( \frac{2a^2}{a^2 + b^2 - c^2 + \sqrt{\lambda[a^2, b^2, c^2]}} \right) + 2\text{Li}_2 \left( \frac{2a^2}{a^2 - b^2 + c^2 + \sqrt{\lambda[a^2, b^2, c^2]}} \right)$$

$$= C_0^S[0, 0, a^2, c, c, b] \tag{C.23}$$

# D    Full Parameterized partial widths

## D.1    Full $\alpha$–scheme results

$$\Delta_\alpha^{R_1} = +0.47\tilde{c}_{H\square}^{(6)} - 0.12\tilde{c}_{HD}^{(6)} - 0.059\tilde{c}_{HD}^{(8)} - 0.059\tilde{c}_{HD,2}^{(8)} + 0.94[\tilde{c}_{H\square}^{(6)}]^2 - 0.47\tilde{c}_{H\square}^{(6)}\tilde{c}_{HD}^{(6)} + 0.059[\tilde{c}_{HD}^{(6)}]^2 \quad (D.1)$$

$$-0.5\frac{\hat{v}}{\hat{m}_\mu}\tilde{c}_{eH}^{(6)} + 0.27\frac{\hat{v}^2}{\hat{m}_\mu^2}[\tilde{c}_{eH}^{(6)}]^2 - 0.25\frac{\hat{v}}{\hat{m}_\mu}\tilde{c}_{eH}^{(8)} - 0.5\frac{\hat{v}}{\hat{m}_\mu}\tilde{c}_{eH}^{(6)}\tilde{c}_{H\square}^{(6)} + 0.13\frac{\hat{v}}{\hat{m}_\mu}\tilde{c}_{eH}^{(6)}\tilde{c}_{HD}^{(6)}$$

$$+0.0038\tilde{c}_{H\square}^{(6)}\frac{\delta G_F}{\hat{G}_F} - 0.00096\tilde{c}_{HD}^{(6)}\frac{\delta G_F}{\hat{G}_F} - 0.0061\frac{\hat{v}}{\hat{m}_\mu}\tilde{c}_{eH}^{(6)}\frac{\delta G_F}{\hat{G}_F} + 9.4\cdot10^{-9}\frac{\delta G_F}{\hat{G}_F}$$

$$+10^4\left(+5.6[\tilde{c}_{HB}^{(6)}]^2 + 1.8\tilde{c}_{HB}^{(6)}\tilde{c}_{HW}^{(6)} + 1.1[\tilde{c}_{HW}^{(6)}]^2 - 4.7\tilde{c}_{HB}^{(6)}\tilde{c}_{HWB}^{(6)} - 2.4\tilde{c}_{HW}^{(6)}\tilde{c}_{HWB}^{(6)} + 1.7[\tilde{c}_{HWB}^{(6)}]^2\right)$$

$$+10^{11}\left(+2.4[\tilde{c}_{eB}^{(6)}]^2 - 2.7\tilde{c}_{eB}^{(6)}\tilde{c}_{eW}^{(6)} + 0.73[\tilde{c}_{eW}^{(6)}]^2\right)$$

$$-260.\left(\tilde{c}_{HB}^{(6)} + \tilde{c}_{HB}^{(8)}\right) - 76.\left(\tilde{c}_{HW}^{(6)} + \tilde{c}_{HW}^{(8)} + \tilde{c}_{HW,2}^{(8)}\right) + 140.\left(\tilde{c}_{HWB}^{(6)} + \tilde{c}_{HWB}^{(8)}\right)$$

$$-520.\left[\tilde{c}_{HB}^{(6)}\right]^2 - 150.\left[\tilde{c}_{HW}^{(6)}\right]^2 + 16.\left[\tilde{c}_{HWB}^{(6)}\right]^2 + 560.\tilde{c}_{HB}^{(6)}\tilde{c}_{HWB}^{(6)} + 0.03\tilde{c}_{HW}^{(6)}\tilde{c}_{HWB}^{(6)}$$

$$-260.\tilde{c}_{HB}^{(6)}\tilde{c}_{H\square}^{(6)} + 120.\tilde{c}_{HB}^{(6)}\tilde{c}_{HD}^{(6)} - 76.\tilde{c}_{HW}^{(6)}\tilde{c}_{H\square}^{(6)} - 41.\tilde{c}_{HW}^{(6)}\tilde{c}_{HD}^{(6)} + 140.\tilde{c}_{HWB}^{(6)}\tilde{c}_{H\square}^{(6)} + 4.\tilde{c}_{HWB}^{(6)}\tilde{c}_{HD}^{(6)}$$

$$-4.9\tilde{c}_{HB}^{(6)}\tilde{c}_{He}^{(6)} - 7.7\tilde{c}_{HB}^{(6)}\left(\tilde{c}_{Hl}^{(6),1} + \tilde{c}_{Hl}^{(6),3}\right) + 4.9\tilde{c}_{HW}^{(6)}\tilde{c}_{He}^{(6)} + 7.7\tilde{c}_{HW}^{(6)}\left(\tilde{c}_{Hl}^{(6),1} + \tilde{c}_{Hl}^{(6),3}\right) - 3.2\tilde{c}_{HWB}^{(6)}\tilde{c}_{He}^{(6)}$$

$$-4.9\tilde{c}_{HWB}^{(6)}\left(\tilde{c}_{Hl}^{(6),1} + \tilde{c}_{Hl}^{(6),3}\right)$$

$$+0.000016\tilde{c}_{HB}^{(6)}\frac{\delta G_F}{\hat{G}_F} - 0.000016\tilde{c}_{HW}^{(6)}\frac{\delta G_F}{\hat{G}_F} + 0.00001\tilde{c}_{HWB}^{(6)}\frac{\delta G_F}{\hat{G}_F}$$

$$-0.09\tilde{c}_{BH^4D^2}^{(8),1} + 0.025\tilde{c}_{WH^4D^2}^{(8),1}$$

$$-1.3\tilde{c}_{e^2BH^2D}^{(8),1} - 0.22\left(\tilde{c}_{e^2H^2D^3}^{(8),1} + \tilde{c}_{e^2H^2D^3}^{(8),2}\right) + 0.69\tilde{c}_{e^2WH^2D}^{(8),1} - 0.12\left(\tilde{c}_{L^2BH^2D}^{(8),1} + \tilde{c}_{L^2BH^2D}^{(8),5}\right)$$

$$-0.023\left(\tilde{c}_{l^2H^2D^3}^{(8),1} + \tilde{c}_{l^2H^2D^3}^{(8),2}\right) - 0.012\left(\tilde{c}_{l^2H^2D^3}^{(8),3} + \tilde{c}_{l^2H^2D^3}^{(8),4}\right) + 0.067\left(\tilde{c}_{l^2WH^2D}^{(8),1} + \tilde{c}_{l^2H^2D^3}^{(8),5}\right)$$

$$\Delta_\alpha^{R_2} = +0.11\tilde{c}_{H\Box}^{(6)} - 0.029\tilde{c}_{HD}^{(6)} - 0.014\tilde{c}_{HD}^{(8)} - 0.014\tilde{c}_{HD,2}^{(8)} + 0.23[\tilde{c}_{H\Box}^{(6)}]^2 - 0.11\tilde{c}_{H\Box}^{(6)}\tilde{c}_{HD}^{(6)} + 0.014[\tilde{c}_{HD}^{(6)}]^2 \quad \text{(D.2)}$$

$$-0.12\frac{\hat{v}}{\hat{m}_\mu}\tilde{c}_{eH}^{(6)} + 0.065\frac{\hat{v}^2}{\hat{m}_\mu^2}[\tilde{c}_{eH}^{(6)}]^2 - 0.061\frac{\hat{v}}{\hat{m}_\mu}\tilde{c}_{eH}^{(8)} - 0.12\frac{\hat{v}}{\hat{m}_\mu}\tilde{c}_{eH}^{(6)}\tilde{c}_{H\Box}^{(6)} + 0.03\frac{\hat{v}}{\hat{m}_\mu}\tilde{c}_{eH}^{(6)}\tilde{c}_{HD}^{(6)}$$

$$+0.00093\tilde{c}_{H\Box}^{(6)}\frac{\delta G_F}{\hat{G}_F} - 0.00023\tilde{c}_{HD}^{(6)}\frac{\delta G_F}{\hat{G}_F} - 0.0015\frac{\hat{v}}{\hat{m}_\mu}\tilde{c}_{eH}^{(6)}\frac{\delta G_F}{\hat{G}_F} + 2.3\cdot 10^{-9}\frac{\delta G_F}{\hat{G}_F}$$

$$+10^5\left(+1.4[\tilde{c}_{HB}^{(6)}]^2 + 0.83\tilde{c}_{HB}^{(6)}\tilde{c}_{HW}^{(6)} + 0.13[\tilde{c}_{HW}^{(6)}]^2 - 1.5\tilde{c}_{HB}^{(6)}\tilde{c}_{HWB}^{(6)} - 0.46\tilde{c}_{HW}^{(6)}\tilde{c}_{HWB}^{(6)} + 0.42[\tilde{c}_{HWB}^{(6)}]^2\right)$$

$$+10^{11}\left(+9.[\tilde{c}_{eB}^{(6)}]^2 - 9.8\tilde{c}_{eB}^{(6)}\tilde{c}_{eW}^{(6)} + 2.7[\tilde{c}_{eW}^{(6)}]^2\right)$$

$$-720.\left(\tilde{c}_{HB}^{(6)} + \tilde{c}_{HB}^{(8)}\right) - 220.\left(\tilde{c}_{HW}^{(6)} + \tilde{c}_{HW}^{(8)} + \tilde{c}_{HW,2}^{(8)}\right) + 400.\left(\tilde{c}_{HWB}^{(6)} + \tilde{c}_{HWB}^{(8)}\right)$$

$$-1400.\left[\tilde{c}_{HB}^{(6)}\right]^2 - 430.\left[\tilde{c}_{HW}^{(6)}\right]^2 + 30.\left[\tilde{c}_{HWB}^{(6)}\right]^2 + 1600.\tilde{c}_{HB}^{(6)}\tilde{c}_{HWB}^{(6)} + 6.4\tilde{c}_{HW}^{(6)}\tilde{c}_{HWB}^{(6)}$$

$$-720.\tilde{c}_{HB}^{(6)}\tilde{c}_{H\Box}^{(6)} + 340.\tilde{c}_{HB}^{(6)}\tilde{c}_{HD}^{(6)} - 220.\tilde{c}_{HW}^{(6)}\tilde{c}_{H\Box}^{(6)} - 110.\tilde{c}_{HW}^{(6)}\tilde{c}_{HD}^{(6)} + 400.\tilde{c}_{HWB}^{(6)}\tilde{c}_{H\Box}^{(6)} + 5.9\tilde{c}_{HWB}^{(6)}\tilde{c}_{HD}^{(6)}$$

$$-18.\tilde{c}_{HB}^{(6)}\tilde{c}_{He}^{(6)} - 14.\tilde{c}_{HB}^{(6)}\left(\tilde{c}_{Hl}^{(6),1} + \tilde{c}_{Hl}^{(6),3}\right) + 18.\tilde{c}_{HW}^{(6)}\tilde{c}_{He}^{(6)} + 14.\tilde{c}_{HW}^{(6)}\left(\tilde{c}_{Hl}^{(6),1} + \tilde{c}_{Hl}^{(6),3}\right) - 12.\tilde{c}_{HWB}^{(6)}\tilde{c}_{He}^{(6)}$$

$$-8.7\tilde{c}_{HWB}^{(6)}\left(\tilde{c}_{Hl}^{(6),1} + \tilde{c}_{Hl}^{(6),3}\right)$$

$$+0.000044\tilde{c}_{HB}^{(6)}\frac{\delta G_F}{\hat{G}_F} - 0.000044\tilde{c}_{HW}^{(6)}\frac{\delta G_F}{\hat{G}_F} + 0.000028\tilde{c}_{HWB}^{(6)}\frac{\delta G_F}{\hat{G}_F}$$

$$+0.05\tilde{c}_{BH^4D^2}^{(8),1} - 0.014\tilde{c}_{WH^4D^2}^{(8),1}$$

$$-3.1\tilde{c}_{e^2BH^2D}^{(8),1} - 0.55\left(\tilde{c}_{e^2H^2D^3}^{(8),1} + \tilde{c}_{e^2H^2D^3}^{(8),2}\right) + 1.7\tilde{c}_{e^2WH^2D}^{(8),1} - 2.4\left(\tilde{c}_{L^2BH^2D}^{(8),1} + \tilde{c}_{L^2BH^2D}^{(8),5}\right)$$

$$-0.42\left(\tilde{c}_{l^2H^2D^3}^{(8),1} + \tilde{c}_{l^2H^2D^3}^{(8),2}\right) - 0.21\left(\tilde{c}_{l^2H^2D^3}^{(8),3} + \tilde{c}_{l^2H^2D^3}^{(8),4}\right) + 1.3\left(\tilde{c}_{l^2WH^2D}^{(8),1} + \tilde{c}_{l^2H^2D^3}^{(8),5}\right)$$

$$\Delta_\alpha^{R_3} = +0.21\tilde{c}_{H\square}^{(6)} - 0.053\tilde{c}_{HD}^{(6)} - 0.026\tilde{c}_{HD}^{(8)} - 0.026\tilde{c}_{HD,2}^{(8)} + 0.42[\tilde{c}_{H\square}^{(6)}]^2 - 0.21\tilde{c}_{H\square}^{(6)}\tilde{c}_{HD}^{(6)} + 0.026[\tilde{c}_{HD}^{(6)}]^2 \quad (\text{D.3})$$

$$-0.22\frac{\hat{v}}{\hat{m}_\mu}\tilde{c}_{eH}^{(6)} + 0.12\frac{\hat{v}^2}{\hat{m}_\mu^2}[\tilde{c}_{eH}^{(6)}]^2 - 0.11\frac{\hat{v}}{\hat{m}_\mu}\tilde{c}_{eH}^{(8)} - 0.22\frac{\hat{v}}{\hat{m}_\mu}\tilde{c}_{eH}^{(6)}\tilde{c}_{H\square}^{(6)} + 0.056\frac{\hat{v}}{\hat{m}_\mu}\tilde{c}_{eH}^{(6)}\tilde{c}_{HD}^{(6)}$$

$$+0.0017\tilde{c}_{H\square}^{(6)}\frac{\delta G_F}{\hat{G}_F} - 0.00043\tilde{c}_{HD}^{(6)}\frac{\delta G_F}{\hat{G}_F} - 0.0027\frac{\hat{v}}{\hat{m}_\mu}\tilde{c}_{eH}^{(6)}\frac{\delta G_F}{\hat{G}_F} + 4.2*10^{-9}\frac{\delta G_F}{\hat{G}_F}$$

$$+10^4\left(+1.4[\tilde{c}_{HB}^{(6)}]^2 - 2.5\tilde{c}_{HB}^{(6)}\tilde{c}_{HW}^{(6)} + 1.3[\tilde{c}_{HW}^{(6)}]^2 + 1.5\tilde{c}_{HB}^{(6)}\tilde{c}_{HWB}^{(6)} - 1.6\tilde{c}_{HW}^{(6)}\tilde{c}_{HWB}^{(6)} + 0.55[\tilde{c}_{HWB}^{(6)}]^2\right)$$

$$+10^{11}\left(+1.1[\tilde{c}_{eB}^{(6)}]^2 - 1.2\tilde{c}_{eB}^{(6)}\tilde{c}_{eW}^{(6)} + 0.32[\tilde{c}_{eW}^{(6)}]^2\right)$$

$$-16.\left(\tilde{c}_{HB}^{(6)} + \tilde{c}_{HB}^{(8)}\right) + 9.\left(\tilde{c}_{HW}^{(6)} + \tilde{c}_{HW}^{(8)} + \tilde{c}_{HW,2}^{(8)}\right) - 3.9\left(\tilde{c}_{HWB}^{(6)} + \tilde{c}_{HWB}^{(8)}\right)$$

$$-31.\left[\tilde{c}_{HB}^{(6)}\right]^2 + 18.\left[\tilde{c}_{HW}^{(6)}\right]^2 + 31.\left[\tilde{c}_{HWB}^{(6)}\right]^2 + 2.7\tilde{c}_{HB}^{(6)}\tilde{c}_{HWB}^{(6)} - 18.\tilde{c}_{HW}^{(6)}\tilde{c}_{HWB}^{(6)}$$

$$-16.\tilde{c}_{HB}^{(6)}\tilde{c}_{H\square}^{(6)} + 8.7\tilde{c}_{HB}^{(6)}\tilde{c}_{HD}^{(6)} + 9.\tilde{c}_{HW}^{(6)}\tilde{c}_{H\square}^{(6)} - 7.1\tilde{c}_{HW}^{(6)}\tilde{c}_{HD}^{(6)} - 3.9\tilde{c}_{HWB}^{(6)}\tilde{c}_{H\square}^{(6)} + 9.8\tilde{c}_{HWB}^{(6)}\tilde{c}_{HD}^{(6)}$$

$$+4.4\tilde{c}_{HB}^{(6)}\tilde{c}_{He}^{(6)} - 16.\tilde{c}_{HB}^{(6)}\left(\tilde{c}_{Hl}^{(6),1} + \tilde{c}_{Hl}^{(6),3}\right) - 4.4\tilde{c}_{HW}^{(6)}\tilde{c}_{He}^{(6)} + 16.\tilde{c}_{HW}^{(6)}\left(\tilde{c}_{Hl}^{(6),1} + \tilde{c}_{Hl}^{(6),3}\right) + 2.8\tilde{c}_{HWB}^{(6)}\tilde{c}_{He}^{(6)}$$

$$-10.\tilde{c}_{HWB}^{(6)}\left(\tilde{c}_{Hl}^{(6),1} + \tilde{c}_{Hl}^{(6),3}\right)$$

$$+1.3\cdot10^{-6}\tilde{c}_{HB}^{(6)}\frac{\delta G_F}{\hat{G}_F} - 1.3\cdot10^{-6}\tilde{c}_{HW}^{(6)}\frac{\delta G_F}{\hat{G}_F} + 2.4\cdot10^{-6}\tilde{c}_{HWB}^{(6)}\frac{\delta G_F}{\hat{G}_F}$$

$$-0.51\tilde{c}_{BH^4D^2}^{(8),1} + 0.14\tilde{c}_{WH^4D^2}^{(8),1}$$

$$-1.\tilde{c}_{e^2BH^2D}^{(8),1} - 0.18\left(\tilde{c}_{e^2H^2D^3}^{(8),1} + \tilde{c}_{e^2H^2D^3}^{(8),2}\right) + 0.56\tilde{c}_{e^2WH^2D}^{(8),1} + 0.51\left(\tilde{c}_{L^2BH^2D}^{(8),1} + \tilde{c}_{L^2BH^2D}^{(8),5}\right)$$

$$+0.089\left(\tilde{c}_{l^2H^2D^3}^{(8),1} + \tilde{c}_{l^2H^2D^3}^{(8),2}\right) + 0.045\left(\tilde{c}_{l^2H^2D^3}^{(8),3} + \tilde{c}_{l^2H^2D^3}^{(8),4}\right) - 0.28\left(\tilde{c}_{l^2WH^2D}^{(8),1} + \tilde{c}_{l^2H^2D^3}^{(8),5}\right)$$

$$
\begin{aligned}
\Delta_\alpha^{R_4} = {} & +1.9\tilde{c}_{H\Box}^{(6)} - 0.49\tilde{c}_{HD}^{(6)} - 0.24\tilde{c}_{HD}^{(8)} - 0.24\tilde{c}_{HD,2}^{(8)} + 3.9[\tilde{c}_{H\Box}^{(6)}]^2 - 1.9\tilde{c}_{H\Box}^{(6)}\tilde{c}_{HD}^{(6)} + 0.24[\tilde{c}_{HD}^{(6)}]^2 \quad \text{(D.4)} \\[4pt]
& -2.1\frac{\hat{v}}{\hat{m}_\mu}\tilde{c}_{eH}^{(6)} + 1.1\frac{\hat{v}^2}{\hat{m}_\mu^2}[\tilde{c}_{eH}^{(6)}]^2 - 1.\frac{\hat{v}}{\hat{m}_\mu}\tilde{c}_{eH}^{(8)} - 2.1\frac{\hat{v}}{\hat{m}_\mu}\tilde{c}_{eH}^{(6)}\tilde{c}_{H\Box}^{(6)} + 0.52\frac{\hat{v}}{\hat{m}_\mu}\tilde{c}_{eH}^{(6)}\tilde{c}_{HD}^{(6)} \\[4pt]
& +0.016\tilde{c}_{H\Box}^{(6)}\frac{\delta G_F}{\hat{G}_F} - 0.004\tilde{c}_{HD}^{(6)}\frac{\delta G_F}{\hat{G}_F} - 0.025\frac{\hat{v}}{\hat{m}_\mu}\tilde{c}_{eH}^{(6)}\frac{\delta G_F}{\hat{G}_F} + 3.9 \cdot 10^{-8}\frac{\delta G_F}{\hat{G}_F} \\[4pt]
& +10^2\left(+4.9[\tilde{c}_{HB}^{(6)}]^2 - 5.8\tilde{c}_{HB}^{(6)}\tilde{c}_{HW}^{(6)} + 3.8[\tilde{c}_{HW}^{(6)}]^2 + 2.6\tilde{c}_{HB}^{(6)}\tilde{c}_{HWB}^{(6)} - 5.4\tilde{c}_{HW}^{(6)}\tilde{c}_{HWB}^{(6)} + 2.1[\tilde{c}_{HWB}^{(6)}]^2\right) \\[4pt]
& +10^{10}\left(+2.4[\tilde{c}_{eB}^{(6)}]^2 - 2.7\tilde{c}_{eB}^{(6)}\tilde{c}_{eW}^{(6)} + 0.73[\tilde{c}_{eW}^{(6)}]^2\right) \\[4pt]
& +4.6\left(\tilde{c}_{HB}^{(6)} + \tilde{c}_{HB}^{(8)}\right) - 6.1\left(\tilde{c}_{HW}^{(6)} + \tilde{c}_{HW}^{(8)} + \tilde{c}_{HW,2}^{(8)}\right) + 4.3\left(\tilde{c}_{HWB}^{(6)} + \tilde{c}_{HWB}^{(8)}\right) \\[4pt]
& +9.1\left[\tilde{c}_{HB}^{(6)}\right]^2 - 12.\left[\tilde{c}_{HW}^{(6)}\right]^2 - 14.\left[\tilde{c}_{HWB}^{(6)}\right]^2 + 11.\tilde{c}_{HB}^{(6)}\tilde{c}_{HWB}^{(6)} + 5.7\tilde{c}_{HW}^{(6)}\tilde{c}_{HWB}^{(6)} \\[4pt]
& +4.6\tilde{c}_{HB}^{(6)}\tilde{c}_{H\Box}^{(6)} - 2.\tilde{c}_{HB}^{(6)}\tilde{c}_{HD}^{(6)} - 6.1\tilde{c}_{HW}^{(6)}\tilde{c}_{H\Box}^{(6)} + 2.3\tilde{c}_{HW}^{(6)}\tilde{c}_{HD}^{(6)} + 4.3\tilde{c}_{HWB}^{(6)}\tilde{c}_{H\Box}^{(6)} - 4.8\tilde{c}_{HWB}^{(6)}\tilde{c}_{HD}^{(6)} \\[4pt]
& -4.\tilde{c}_{HB}^{(6)}\tilde{c}_{He}^{(6)} + 7.2\tilde{c}_{HB}^{(6)}\left(\tilde{c}_{Hl}^{(6),1} + \tilde{c}_{Hl}^{(6),3}\right) + 4.\tilde{c}_{HW}^{(6)}\tilde{c}_{He}^{(6)} - 7.2\tilde{c}_{HW}^{(6)}\left(\tilde{c}_{Hl}^{(6),1} + \tilde{c}_{Hl}^{(6),3}\right) - 2.5\tilde{c}_{HWB}^{(6)}\tilde{c}_{He}^{(6)} \\[4pt]
& +4.6\tilde{c}_{HWB}^{(6)}\left(\tilde{c}_{Hl}^{(6),1} + \tilde{c}_{Hl}^{(6),3}\right) \\[4pt]
& -2.2 \cdot 10^{-7}\tilde{c}_{HB}^{(6)}\frac{\delta G_F}{\hat{G}_F} + 2.2 \cdot 10^{-7}\tilde{c}_{HW}^{(6)}\frac{\delta G_F}{\hat{G}_F} - 9.9 \cdot 10^{-7}\tilde{c}_{HWB}^{(6)}\frac{\delta G_F}{\hat{G}_F} \\[4pt]
& +0.28\tilde{c}_{BH^4D^2}^{(8),1} - 0.076\tilde{c}_{WH^4D^2}^{(8),1} \\[4pt]
& +0.22\tilde{c}_{e^2BH^2D}^{(8),1} + 0.04\left(\tilde{c}_{e^2H^2D^3}^{(8),1} + \tilde{c}_{e^2H^2D^3}^{(8),2}\right) - 0.12\tilde{c}_{e^2WH^2D}^{(8),1} - 0.42\left(\tilde{c}_{L^2BH^2D}^{(8),1} + \tilde{c}_{L^2BH^2D}^{(8),5}\right) \\[4pt]
& -0.074\left(\tilde{c}_{l^2H^2D^3}^{(8),1} + \tilde{c}_{l^2H^2D^3}^{(8),2}\right) - 0.037\left(\tilde{c}_{l^2H^2D^3}^{(8),3} + \tilde{c}_{l^2H^2D^3}^{(8),4}\right) + 0.23\left(\tilde{c}_{l^2WH^2D}^{(8),1} + \tilde{c}_{l^2H^2D^3}^{(8),5}\right)
\end{aligned}
$$

$$\Delta_\alpha^{R_5} = +0.93\tilde{c}_{H\Box}^{(6)} - 0.23\tilde{c}_{HD}^{(6)} - 0.12\tilde{c}_{HD}^{(8)} - 0.12\tilde{c}_{HD,2}^{(8)} + 1.9[\tilde{c}_{H\Box}^{(6)}]^2 - 0.93\tilde{c}_{H\Box}^{(6)}\tilde{c}_{HD}^{(6)} + 0.12[\tilde{c}_{HD}^{(6)}]^2 \qquad \text{(D.5)}$$

$$-0.99\frac{\hat{v}}{\hat{m}_\mu}\tilde{c}_{eH}^{(6)} + 0.52\frac{\hat{v}^2}{\hat{m}_\mu^2}[\tilde{c}_{eH}^{(6)}]^2 - 0.49\frac{\hat{v}}{\hat{m}_\mu}\tilde{c}_{eH}^{(8)} - 0.99\frac{\hat{v}}{\hat{m}_\mu}\tilde{c}_{eH}^{(6)}\tilde{c}_{H\Box}^{(6)} + 0.25\frac{\hat{v}}{\hat{m}_\mu}\tilde{c}_{eH}^{(6)}\tilde{c}_{HD}^{(6)}$$

$$+0.0076\tilde{c}_{H\Box}^{(6)}\frac{\delta G_F}{\hat{G}_F} - 0.0019\tilde{c}_{HD}^{(6)}\frac{\delta G_F}{\hat{G}_F} - 0.012\frac{\hat{v}}{\hat{m}_\mu}\tilde{c}_{eH}^{(6)}\frac{\delta G_F}{\hat{G}_F} + 1.9\cdot 10^{-8}\frac{\delta G_F}{\hat{G}_F}$$

$$+10^3\left(+7.4[\tilde{c}_{HB}^{(6)}]^2 - 17.\tilde{c}_{HB}^{(6)}\tilde{c}_{HW}^{(6)} + 6.9[\tilde{c}_{HW}^{(6)}]^2 + 11.\tilde{c}_{HB}^{(6)}\tilde{c}_{HWB}^{(6)} - 8.\tilde{c}_{HW}^{(6)}\tilde{c}_{HWB}^{(6)} + 2.1[\tilde{c}_{HWB}^{(6)}]^2\right)$$

$$+10^{11}\left(+7.7[\tilde{c}_{eB}^{(6)}]^2 - 8.5\tilde{c}_{eB}^{(6)}\tilde{c}_{eW}^{(6)} + 2.3[\tilde{c}_{eW}^{(6)}]^2\right)$$

$$+150.\left(\tilde{c}_{HB}^{(6)} + \tilde{c}_{HB}^{(8)}\right) - 120.\left(\tilde{c}_{HW}^{(6)} + \tilde{c}_{HW}^{(8)} + \tilde{c}_{HW,2}^{(8)}\right) + 68.\left(\tilde{c}_{HWB}^{(6)} + \tilde{c}_{HWB}^{(8)}\right)$$

$$+300.\left[\tilde{c}_{HB}^{(6)}\right]^2 - 240.\left[\tilde{c}_{HW}^{(6)}\right]^2 - 160.\left[\tilde{c}_{HWB}^{(6)}\right]^2 + 380.\tilde{c}_{HB}^{(6)}\tilde{c}_{HWB}^{(6)} - 110.\tilde{c}_{HW}^{(6)}\tilde{c}_{HWB}^{(6)}$$

$$+150.\tilde{c}_{HB}^{(6)}\tilde{c}_{H\Box}^{(6)} - 18.\tilde{c}_{HB}^{(6)}\tilde{c}_{HD}^{(6)} - 120.\tilde{c}_{HW}^{(6)}\tilde{c}_{H\Box}^{(6)} + 10.\tilde{c}_{HW}^{(6)}\tilde{c}_{HD}^{(6)} + 68.\tilde{c}_{HWB}^{(6)}\tilde{c}_{H\Box}^{(6)} - 74.\tilde{c}_{HWB}^{(6)}\tilde{c}_{HD}^{(6)}$$

$$-170.\tilde{c}_{HB}^{(6)}\tilde{c}_{He}^{(6)} + 92.\tilde{c}_{HB}^{(6)}\left(\tilde{c}_{Hl}^{(6),1} + \tilde{c}_{Hl}^{(6),3}\right) + 170.\tilde{c}_{HW}^{(6)}\tilde{c}_{He}^{(6)} - 92.\tilde{c}_{HW}^{(6)}\left(\tilde{c}_{Hl}^{(6),1} + \tilde{c}_{Hl}^{(6),3}\right) - 110.\tilde{c}_{HWB}^{(6)}\tilde{c}_{He}^{(6)}$$

$$+59.\tilde{c}_{HWB}^{(6)}\left(\tilde{c}_{Hl}^{(6),1} + \tilde{c}_{Hl}^{(6),3}\right)$$

$$+5.2\cdot 10^{-6}\tilde{c}_{HB}^{(6)}\frac{\delta G_F}{\hat{G}_F} - 5.2\cdot 10^{-6}\tilde{c}_{HW}^{(6)}\frac{\delta G_F}{\hat{G}_F} - 0.000015\tilde{c}_{HWB}^{(6)}\frac{\delta G_F}{\hat{G}_F}$$

$$+6.1\tilde{c}_{BH^4D^2}^{(8),1} - 1.7\tilde{c}_{WH^4D^2}^{(8),1}$$

$$-9.6\tilde{c}_{e^2BH^2D}^{(8),1} - 1.7\left(\tilde{c}_{e^2H^2D^3}^{(8),1} + \tilde{c}_{e^2H^2D^3}^{(8),2}\right) + 5.3\tilde{c}_{e^2WH^2D}^{(8),1} + 6.4\left(\tilde{c}_{L^2BH^2D}^{(8),1} + \tilde{c}_{L^2BH^2D}^{(8),5}\right)$$

$$+1.1\left(\tilde{c}_{l^2H^2D^3}^{(8),1} + \tilde{c}_{l^2H^2D^3}^{(8),2}\right) + 0.56\left(\tilde{c}_{l^2H^2D^3}^{(8),3} + \tilde{c}_{l^2H^2D^3}^{(8),4}\right) - 3.5\left(\tilde{c}_{l^2WH^2D}^{(8),1} + \tilde{c}_{l^2H^2D^3}^{(8),5}\right)$$

$$\Delta_\alpha^{R_6} = +0.82\tilde{c}_{H\square}^{(6)} - 0.2\tilde{c}_{HD}^{(6)} - 0.1\tilde{c}_{HD}^{(8)} - 0.1\tilde{c}_{HD,2}^{(8)} + 1.6[\tilde{c}_{H\square}^{(6)}]^2 - 0.82\tilde{c}_{H\square}^{(6)}\tilde{c}_{HD}^{(6)} + 0.1[\tilde{c}_{HD}^{(6)}]^2 \qquad \text{(D.6)}$$

$$-0.87\frac{\hat{v}}{\hat{m}_\mu}\tilde{c}_{eH}^{(6)} + 0.46\frac{\hat{v}^2}{\hat{m}_\mu^2}[\tilde{c}_{eH}^{(6)}]^2 - 0.43\frac{\hat{v}}{\hat{m}_\mu}\tilde{c}_{eH}^{(8)} - 0.87\frac{\hat{v}}{\hat{m}_\mu}\tilde{c}_{eH}^{(6)}\tilde{c}_{H\square}^{(6)} + 0.22\frac{\hat{v}}{\hat{m}_\mu}\tilde{c}_{eH}^{(6)}\tilde{c}_{HD}^{(6)}$$

$$+0.0066\tilde{c}_{H\square}^{(6)}\frac{\delta G_F}{\hat{G}_F} - 0.0017\tilde{c}_{HD}^{(6)}\frac{\delta G_F}{\hat{G}_F} - 0.011\frac{\hat{v}}{\hat{m}_\mu}\tilde{c}_{eH}^{(6)}\frac{\delta G_F}{\hat{G}_F} - 8 + 1.6*10\frac{\delta G_F}{\hat{G}_F}$$

$$+10^4\left(+3.8[\tilde{c}_{HB}^{(6)}]^2 + 0.66\tilde{c}_{HB}^{(6)}\tilde{c}_{HW}^{(6)} + 0.78[\tilde{c}_{HW}^{(6)}]^2 - 2.7\tilde{c}_{HB}^{(6)}\tilde{c}_{HWB}^{(6)} - 1.6\tilde{c}_{HW}^{(6)}\tilde{c}_{HWB}^{(6)} + 1.1[\tilde{c}_{HWB}^{(6)}]^2\right)$$

$$+10^{12}\left(+1.4[\tilde{c}_{eB}^{(6)}]^2 - 1.6\tilde{c}_{eB}^{(6)}\tilde{c}_{eW}^{(6)} + 0.43[\tilde{c}_{eW}^{(6)}]^2\right)$$

$$-50.\left(\tilde{c}_{HB}^{(6)} + \tilde{c}_{HB}^{(8)}\right) - 130.\left(\tilde{c}_{HW}^{(6)} + \tilde{c}_{HW}^{(8)} + \tilde{c}_{HW,2}^{(8)}\right) + 140.\left(\tilde{c}_{HWB}^{(6)} + \tilde{c}_{HWB}^{(8)}\right)$$

$$-100.\left[\tilde{c}_{HB}^{(6)}\right]^2 - 270.\left[\tilde{c}_{HW}^{(6)}\right]^2 - 82.\left[\tilde{c}_{HWB}^{(6)}\right]^2 + 660.\tilde{c}_{HB}^{(6)}\tilde{c}_{HWB}^{(6)} - 120.\tilde{c}_{HW}^{(6)}\tilde{c}_{HWB}^{(6)}$$

$$-50.\tilde{c}_{HB}^{(6)}\tilde{c}_{H\square}^{(6)} + 72.\tilde{c}_{HB}^{(6)}\tilde{c}_{HD}^{(6)} - 130.\tilde{c}_{HW}^{(6)}\tilde{c}_{H\square}^{(6)} - 26.\tilde{c}_{HW}^{(6)}\tilde{c}_{HD}^{(6)} + 140.\tilde{c}_{HWB}^{(6)}\tilde{c}_{H\square}^{(6)} - 47.\tilde{c}_{HWB}^{(6)}\tilde{c}_{HD}^{(6)}$$

$$-140.\tilde{c}_{HB}^{(6)}\tilde{c}_{He}^{(6)} + 51.\tilde{c}_{HB}^{(6)}\left(\tilde{c}_{Hl}^{(6),1} + \tilde{c}_{Hl}^{(6),3}\right) + 140.\tilde{c}_{HW}^{(6)}\tilde{c}_{He}^{(6)} - 51.\tilde{c}_{HW}^{(6)}\left(\tilde{c}_{Hl}^{(6),1} + \tilde{c}_{Hl}^{(6),3}\right) - 88.\tilde{c}_{HWB}^{(6)}\tilde{c}_{He}^{(6)}$$

$$+33.\tilde{c}_{HWB}^{(6)}\left(\tilde{c}_{Hl}^{(6),1} + \tilde{c}_{Hl}^{(6),3}\right)$$

$$+0.000016\tilde{c}_{HB}^{(6)}\frac{\delta G_F}{\hat{G}_F} - 0.000016\tilde{c}_{HW}^{(6)}\frac{\delta G_F}{\hat{G}_F} - 3.4\cdot10^{-6}\tilde{c}_{HWB}^{(6)}\frac{\delta G_F}{\hat{G}_F}$$

$$+4.4\tilde{c}_{BH^4D^2}^{(8),1} - 1.2\tilde{c}_{WH^4D^2}^{(8),1}$$

$$-10.\tilde{c}_{e^2BH^2D}^{(8),1} - 1.8\left(\tilde{c}_{e^2H^2D^3}^{(8),1} + \tilde{c}_{e^2H^2D^3}^{(8),2}\right) + 5.6\tilde{c}_{e^2WH^2D}^{(8),1} + 2.8\left(\tilde{c}_{L^2BH^2D}^{(8),1} + \tilde{c}_{L^2BH^2D}^{(8),5}\right)$$

$$+0.48\left(\tilde{c}_{l^2H^2D^3}^{(8),1} + \tilde{c}_{l^2H^2D^3}^{(8),2}\right) + 0.24\left(\tilde{c}_{l^2H^2D^3}^{(8),3} + \tilde{c}_{l^2H^2D^3}^{(8),4}\right) - 1.5\left(\tilde{c}_{l^2WH^2D}^{(8),1} + \tilde{c}_{l^2H^2D^3}^{(8),5}\right)$$

## D.2   $M_W$ scheme results

The parameterized partial widths in the $M_W$ input parameter scheme for each region defined in the main text are given below. When solving for $\bar{s}_W$ in terms of the input parameters we obtain terms proportional to $\sqrt{[c_{HWB}^{(6)}]^2} = |c_{HWB}^{(6)}|$. This is not mentioned explicitly in the Appendix of [13] where equations useful for solving barred quantities for input parameters are found. The authors of [13] have found a work around, but this was not apparent to the authors of this article. Note that in the $\hat{M}_W$ scheme Case 5 has corrections from the Class 4 operators which shift the definition of the $\alpha$, in the $\hat{\alpha}$ scheme this is not the case as $\alpha$ is an input parameter.

$$
\begin{aligned}
\Delta^{R_1}_{M_W} = {} & +0.48\tilde{c}^{(6)}_{H\square} - 0.12\tilde{c}^{(6)}_{HD} - 0.06\tilde{c}^{(8)}_{HD} - 0.06\tilde{c}^{(8)}_{HD,2} + 0.96[\tilde{c}^{(6)}_{H\square}]^2 - 1.1\tilde{c}^{(6)}_{H\square}\tilde{c}^{(6)}_{HD} + 0.22[\tilde{c}^{(6)}_{HD}]^2 \quad \text{(D.7)} \\[4pt]
& -0.51\frac{\hat{v}}{\bar{m}_\mu}\tilde{c}^{(6)}_{eH} + 0.27\frac{\hat{v}^2}{\bar{m}_\mu^2}[\tilde{c}^{(6)}_{eH}]^2 - 0.25\frac{\hat{v}}{\bar{m}_\mu}\tilde{c}^{(8)}_{eH} - 0.51\frac{\hat{v}}{\bar{m}_\mu}\tilde{c}^{(6)}_{eH}\tilde{c}^{(6)}_{H\square} + 0.8\frac{\hat{v}}{\bar{m}_\mu}\tilde{c}^{(6)}_{eH}\tilde{c}^{(6)}_{HD} \\[4pt]
& +0.0039\tilde{c}^{(6)}_{H\square}\frac{\delta G_F}{\hat{G}_F} - 0.00097\tilde{c}^{(6)}_{HD}\frac{\delta G_F}{\hat{G}_F} - 0.0062\frac{\hat{v}}{\bar{m}_\mu}\tilde{c}^{(6)}_{eH}\frac{\delta G_F}{\hat{G}_F} + 9.5\cdot 10^{-9}\frac{\delta G_F}{\hat{G}_F} \\[4pt]
& -0.68\tilde{c}^{(6)}_{H\square}|\tilde{c}^{(6)}_{HWB}| + 0.17\tilde{c}^{(6)}_{HD}|\tilde{c}^{(6)}_{HWB}| + 0.72\frac{\hat{v}}{\bar{m}_\mu}\tilde{c}^{(6)}_{eH}|\tilde{c}^{(6)}_{HWB}| \\[4pt]
& +10^4\left(+5.8[\tilde{c}^{(6)}_{HB}]^2 + 1.7\tilde{c}^{(6)}_{HB}\tilde{c}^{(6)}_{HW} + 1.[\tilde{c}^{(6)}_{HW}]^2 - 4.7\tilde{c}^{(6)}_{HB}\tilde{c}^{(6)}_{HWB} - 2.4\tilde{c}^{(6)}_{HW}\tilde{c}^{(6)}_{HWB} + 1.7[\tilde{c}^{(6)}_{HWB}]^2\right) \\[4pt]
& +10^{11}\left(+2.6[\tilde{c}^{(6)}_{eB}]^2 - 2.7\tilde{c}^{(6)}_{eB}\tilde{c}^{(6)}_{eW} + 0.73[\tilde{c}^{(6)}_{eW}]^2\right) \\[4pt]
& -260.\left(\tilde{c}^{(6)}_{HB} + \tilde{c}^{(8)}_{HB}\right) - 71.\left(\tilde{c}^{(6)}_{HW} + \tilde{c}^{(8)}_{HW} + \tilde{c}^{(8)}_{HW,2}\right) + 140.\left(\tilde{c}^{(6)}_{HWB} + \tilde{c}^{(8)}_{HWB}\right) \\[4pt]
& -520.\left[\tilde{c}^{(6)}_{HB}\right]^2 - 140.\left[\tilde{c}^{(6)}_{HW}\right]^2 - 230.\left[\tilde{c}^{(6)}_{HWB}\right]^2 + 400.\tilde{c}^{(6)}_{HB}\tilde{c}^{(6)}_{HWB} + 310.\tilde{c}^{(6)}_{HW}\tilde{c}^{(6)}_{HWB} \\[4pt]
& -260.\tilde{c}^{(6)}_{HB}\tilde{c}^{(6)}_{H\square} + 44.\tilde{c}^{(6)}_{HB}\tilde{c}^{(6)}_{HD} - 71.\tilde{c}^{(6)}_{HW}\tilde{c}^{(6)}_{H\square} + 190.\tilde{c}^{(6)}_{HW}\tilde{c}^{(6)}_{HD} + 140.\tilde{c}^{(6)}_{HWB}\tilde{c}^{(6)}_{H\square} - 190.\tilde{c}^{(6)}_{HWB}\tilde{c}^{(6)}_{HD} \\[4pt]
& -4.8\tilde{c}^{(6)}_{HB}\tilde{c}^{(6)}_{He} - 7.7\tilde{c}^{(6)}_{HB}\left(\tilde{c}^{(6),1}_{Hl} + \tilde{c}^{(6),3}_{Hl}\right) + 4.8\tilde{c}^{(6)}_{HW}\tilde{c}^{(6)}_{He} + 7.7\tilde{c}^{(6)}_{HW}\left(\tilde{c}^{(6),1}_{Hl} + \tilde{c}^{(6),3}_{Hl}\right) - 3.2\tilde{c}^{(6)}_{HWB}\tilde{c}^{(6)}_{He} \\[4pt]
& -5.1\tilde{c}^{(6)}_{HWB}\left(\tilde{c}^{(6),1}_{Hl} + \tilde{c}^{(6),3}_{Hl}\right) \\[4pt]
& +9.0\cdot 10^{-6}\tilde{c}^{(6)}_{HB}\frac{\delta G_F}{\hat{G}_F} + 2.5\cdot 10^{-6}\tilde{c}^{(6)}_{HW}\frac{\delta G_F}{\hat{G}_F} - 4.7\cdot 10^{-6}\tilde{c}^{(6)}_{HWB}\frac{\delta G_F}{\hat{G}_F} \\[4pt]
& -0.1\tilde{c}^{(8),1}_{BH^4D^2} + 0.028\tilde{c}^{(8),1}_{WH^4D^2} \\[4pt]
& -1.3\tilde{c}^{(8),1}_{e^2BH^2D} - 0.22\left(\tilde{c}^{(8),1}_{e^2H^2D^3} + \tilde{c}^{(8),2}_{e^2H^2D^3}\right) + 0.68\tilde{c}^{(8),1}_{e^2WH^2D} - 0.085\left(\tilde{c}^{(8),1}_{L^2BH^2D} + \tilde{c}^{(8),5}_{L^2BH^2D}\right) \\[4pt]
& -0.016\left(\tilde{c}^{(8),1}_{l^2H^2D^3} + \tilde{c}^{(8),2}_{l^2H^2D^3}\right) - 0.0082\left(\tilde{c}^{(8),3}_{l^2H^2D^3} + \tilde{c}^{(8),4}_{l^2H^2D^3}\right) + 0.045\left(\tilde{c}^{(8),1}_{l^2WH^2D} + \tilde{c}^{(8),5}_{l^2H^2D^3}\right)
\end{aligned}
$$

$$\Delta_{M_W}^{R_2} = +0.12\tilde{c}_{H\square}^{(6)} - 0.03\tilde{c}_{HD}^{(6)} - 0.015\tilde{c}_{HD}^{(8)} - 0.015\tilde{c}_{HD,2}^{(8)} + 0.24[\tilde{c}_{H\square}^{(6)}]^2 - 0.32\tilde{c}_{H\square}^{(6)}\tilde{c}_{HD}^{(6)} + 0.065[\tilde{c}_{HD}^{(6)}]^2 \quad \text{(D.8)}$$

$$-0.13\frac{\hat{v}}{\bar{m}_\mu}\tilde{c}_{eH}^{(6)} + 0.069\frac{\hat{v}^2}{\bar{m}_\mu^2}[\tilde{c}_{eH}^{(6)}]^2 - 0.065\frac{\hat{v}}{\bar{m}_\mu}\tilde{c}_{eH}^{(8)} - 0.13\frac{\hat{v}}{\bar{m}_\mu}\tilde{c}_{eH}^{(6)}\tilde{c}_{H\square}^{(6)} + 0.24\frac{\hat{v}}{\bar{m}_\mu}\tilde{c}_{eH}^{(6)}\tilde{c}_{HD}^{(6)}$$

$$+0.00099\tilde{c}_{H\square}^{(6)}\frac{\delta G_F}{\hat{G}_F} - 0.00025\tilde{c}_{HD}^{(6)}\frac{\delta G_F}{\hat{G}_F} - 0.0016\frac{\hat{v}}{\bar{m}_\mu}\tilde{c}_{eH}^{(6)}\frac{\delta G_F}{\hat{G}_F} + 2.4\cdot10^{-9}\frac{\delta G_F}{\hat{G}_F}$$

$$-0.21\tilde{c}_{H\square}^{(6)}|\tilde{c}_{HWB}^{(6)}| + 0.053\tilde{c}_{HD}^{(6)}|\tilde{c}_{HWB}^{(6)}| + 0.23\frac{\hat{v}}{\bar{m}_\mu}\tilde{c}_{eH}^{(6)}|\tilde{c}_{HWB}^{(6)}|$$

$$+10^5\left(+1.5[\tilde{c}_{HB}^{(6)}]^2 + 0.86\tilde{c}_{HB}^{(6)}\tilde{c}_{HW}^{(6)} + 0.12[\tilde{c}_{HW}^{(6)}]^2 - 1.6\tilde{c}_{HB}^{(6)}\tilde{c}_{HWB}^{(6)} - 0.46\tilde{c}_{HW}^{(6)}\tilde{c}_{HWB}^{(6)} + 0.43[\tilde{c}_{HWB}^{(6)}]^2\right)$$

$$+10^{11}\left(+9.8[\tilde{c}_{eB}^{(6)}]^2 - 11.\tilde{c}_{eB}^{(6)}\tilde{c}_{eW}^{(6)} + 2.8[\tilde{c}_{eW}^{(6)}]^2\right)$$

$$-750.\left(\tilde{c}_{HB}^{(6)} + \tilde{c}_{HB}^{(8)}\right) - 220.\left(\tilde{c}_{HW}^{(6)} + \tilde{c}_{HW}^{(8)} + \tilde{c}_{HW,2}^{(8)}\right) + 400.\left(\tilde{c}_{HWB}^{(6)} + \tilde{c}_{HWB}^{(8)}\right)$$

$$-1500.\left[\tilde{c}_{HB}^{(6)}\right]^2 - 430.\left[\tilde{c}_{HW}^{(6)}\right]^2 - 810.\left[\tilde{c}_{HWB}^{(6)}\right]^2 + 1400.\tilde{c}_{HB}^{(6)}\tilde{c}_{HWB}^{(6)} + 970.\tilde{c}_{HW}^{(6)}\tilde{c}_{HWB}^{(6)}$$

$$-750.\tilde{c}_{HB}^{(6)}\tilde{c}_{H\square}^{(6)} + 360.\tilde{c}_{HB}^{(6)}\tilde{c}_{HD}^{(6)} - 220.\tilde{c}_{HW}^{(6)}\tilde{c}_{H\square}^{(6)} + 620.\tilde{c}_{HW}^{(6)}\tilde{c}_{HD}^{(6)} + 400.\tilde{c}_{HWB}^{(6)}\tilde{c}_{H\square}^{(6)} - 670.\tilde{c}_{HWB}^{(6)}\tilde{c}_{HD}^{(6)}$$

$$-19.\tilde{c}_{HB}^{(6)}\tilde{c}_{He}^{(6)} - 14.\tilde{c}_{HB}^{(6)}\left(\tilde{c}_{Hl}^{(6),1} + \tilde{c}_{Hl}^{(6),3}\right) + 19.\tilde{c}_{HW}^{(6)}\tilde{c}_{He}^{(6)} + 14.\tilde{c}_{HW}^{(6)}\left(\tilde{c}_{Hl}^{(6),1} + \tilde{c}_{Hl}^{(6),3}\right) - 12.\tilde{c}_{HWB}^{(6)}\tilde{c}_{He}^{(6)}$$

$$-9.2\tilde{c}_{HWB}^{(6)}\left(\tilde{c}_{Hl}^{(6),1} + \tilde{c}_{Hl}^{(6),3}\right)$$

$$+0.000044\tilde{c}_{HB}^{(6)}\frac{\delta G_F}{\hat{G}_F} + 0.000013\tilde{c}_{HW}^{(6)}\frac{\delta G_F}{\hat{G}_F} - 0.000024\tilde{c}_{HWB}^{(6)}\frac{\delta G_F}{\hat{G}_F}$$

$$+0.03\tilde{c}_{BH^4D^2}^{(8),1} - 0.008\tilde{c}_{WH^4D^2}^{(8),1}$$

$$-3.3\tilde{c}_{e^2BH^2D}^{(8),1} - 0.57\left(\tilde{c}_{e^2H^2D^3}^{(8),1} + \tilde{c}_{e^2H^2D^3}^{(8),2}\right) + 1.8\tilde{c}_{e^2WH^2D}^{(8),1} - 2.5\left(\tilde{c}_{L^2BH^2D}^{(8),1} + \tilde{c}_{L^2BH^2D}^{(8),5}\right)$$

$$-0.43\left(\tilde{c}_{l^2H^2D^3}^{(8),1} + \tilde{c}_{l^2H^2D^3}^{(8),2}\right) - 0.22\left(\tilde{c}_{l^2H^2D^3}^{(8),3} + \tilde{c}_{l^2H^2D^3}^{(8),4}\right) + 1.3\left(\tilde{c}_{l^2WH^2D}^{(8),1} + \tilde{c}_{l^2H^2D^3}^{(8),5}\right)$$

$$\Delta_{M_W}^{R_3} = +0.21\tilde{c}_{H\square}^{(6)} - 0.052\tilde{c}_{HD}^{(6)} - 0.026\tilde{c}_{HD}^{(8)} - 0.026\tilde{c}_{HD,2}^{(8)} + 0.41[\tilde{c}_{H\square}^{(6)}]^2 - 0.53\tilde{c}_{H\square}^{(6)}\tilde{c}_{HD}^{(6)} + 0.11[\tilde{c}_{HD}^{(6)}]^2 \quad (D.9)$$

$$-0.22\frac{\hat{v}}{\bar{m}_\mu}\tilde{c}_{eH}^{(6)} + 0.12\frac{\hat{v}^2}{\bar{m}_\mu^2}[\tilde{c}_{eH}^{(6)}]^2 - 0.11\frac{\hat{v}}{\bar{m}_\mu}\tilde{c}_{eH}^{(8)} - 0.22\frac{\hat{v}}{\bar{m}_\mu}\tilde{c}_{eH}^{(6)}\tilde{c}_{H\square}^{(6)} + 0.4\frac{\hat{v}}{\bar{m}_\mu}\tilde{c}_{eH}^{(6)}\tilde{c}_{HD}^{(6)}$$

$$+0.0017\tilde{c}_{H\square}^{(6)}\frac{\delta G_F}{\hat{G}_F} - 0.00042\tilde{c}_{HD}^{(6)}\frac{\delta G_F}{\hat{G}_F} - 0.0027\frac{\hat{v}}{\bar{m}_\mu}\tilde{c}_{eH}^{(6)}\frac{\delta G_F}{\hat{G}_F} + 4.1\cdot10^{-9}\frac{\delta G_F}{\hat{G}_F}$$

$$-0.35\tilde{c}_{H\square}^{(6)}|\tilde{c}_{HWB}^{(6)}| + 0.087\tilde{c}_{HD}^{(6)}|\tilde{c}_{HWB}^{(6)}| + 0.37\frac{\hat{v}}{\bar{m}_\mu}\tilde{c}_{eH}^{(6)}|\tilde{c}_{HWB}^{(6)}|$$

$$+10^4\left(+1.4[\tilde{c}_{HB}^{(6)}]^2 - 2.4\tilde{c}_{HB}^{(6)}\tilde{c}_{HW}^{(6)} + 1.3[\tilde{c}_{HW}^{(6)}]^2 + 1.5\tilde{c}_{HB}^{(6)}\tilde{c}_{HWB}^{(6)} - 1.7\tilde{c}_{HW}^{(6)}\tilde{c}_{HWB}^{(6)} + 0.58[\tilde{c}_{HWB}^{(6)}]^2\right)$$

$$+10^{11}\left(+1.1[\tilde{c}_{eB}^{(6)}]^2 - 1.2\tilde{c}_{eB}^{(6)}\tilde{c}_{eW}^{(6)} + 0.31[\tilde{c}_{eW}^{(6)}]^2\right)$$

$$-15.\left(\tilde{c}_{HB}^{(6)} + \tilde{c}_{HB}^{(8)}\right) + 9.4\left(\tilde{c}_{HW}^{(6)} + \tilde{c}_{HW}^{(8)} + \tilde{c}_{HW,2}^{(8)}\right) - 4.7\left(\tilde{c}_{HWB}^{(6)} + \tilde{c}_{HWB}^{(8)}\right)$$

$$-30.\left[\tilde{c}_{HB}^{(6)}\right]^2 + 19.\left[\tilde{c}_{HW}^{(6)}\right]^2 - 5.\left[\tilde{c}_{HWB}^{(6)}\right]^2 - 6.9\tilde{c}_{HB}^{(6)}\tilde{c}_{HWB}^{(6)} - 7.9\tilde{c}_{HW}^{(6)}\tilde{c}_{HWB}^{(6)}$$

$$-15.\tilde{c}_{HB}^{(6)}\tilde{c}_{H\square}^{(6)} + 3.\tilde{c}_{HB}^{(6)}\tilde{c}_{HD}^{(6)} + 9.4\tilde{c}_{HW}^{(6)}\tilde{c}_{H\square}^{(6)} + 2.3\tilde{c}_{HW}^{(6)}\tilde{c}_{HD}^{(6)} - 4.7\tilde{c}_{HWB}^{(6)}\tilde{c}_{H\square}^{(6)} - 17.\tilde{c}_{HWB}^{(6)}\tilde{c}_{HD}^{(6)}$$

$$+4.3\tilde{c}_{HB}^{(6)}\tilde{c}_{He}^{(6)} - 16.\tilde{c}_{HB}^{(6)}\left(\tilde{c}_{Hl}^{(6),1} + \tilde{c}_{Hl}^{(6),3}\right) - 4.3\tilde{c}_{HW}^{(6)}\tilde{c}_{He}^{(6)} + 16.\tilde{c}_{HW}^{(6)}\left(\tilde{c}_{Hl}^{(6),1} + \tilde{c}_{Hl}^{(6),3}\right) + 2.8\tilde{c}_{HWB}^{(6)}\tilde{c}_{He}^{(6)}$$

$$-11.\tilde{c}_{HWB}^{(6)}\left(\tilde{c}_{Hl}^{(6),1} + \tilde{c}_{Hl}^{(6),3}\right)$$

$$+8.\cdot10^{-7}\tilde{c}_{HB}^{(6)}\frac{\delta G_F}{\hat{G}_F} - 5.\cdot10^{-7}\tilde{c}_{HW}^{(6)}\frac{\delta G_F}{\hat{G}_F} + 2.5\cdot10^{-7}\tilde{c}_{HWB}^{(6)}\frac{\delta G_F}{\hat{G}_F}$$

$$-0.52\tilde{c}_{BH^4D^2}^{(8),1} + 0.14\tilde{c}_{WH^4D^2}^{(8),1}$$

$$-1.\tilde{c}_{e^2BH^2D}^{(8),1} - 0.17\left(\tilde{c}_{e^2H^2D^3}^{(8),1} + \tilde{c}_{e^2H^2D^3}^{(8),2}\right) + 0.53\tilde{c}_{e^2WH^2D}^{(8),1} + 0.55\left(\tilde{c}_{L^2BH^2D}^{(8),1} + \tilde{c}_{L^2BH^2D}^{(8),5}\right)$$

$$+0.094\left(\tilde{c}_{l^2H^2D^3}^{(8),1} + \tilde{c}_{l^2H^2D^3}^{(8),2}\right) + 0.047\left(\tilde{c}_{l^2H^2D^3}^{(8),3} + \tilde{c}_{l^2H^2D^3}^{(8),4}\right) - 0.29\left(\tilde{c}_{l^2WH^2D}^{(8),1} + \tilde{c}_{l^2H^2D^3}^{(8),5}\right)$$

$$\Delta^{R_4}_{M_W} = +1.9\tilde{c}^{(6)}_{H\square} - 0.49\tilde{c}^{(6)}_{HD} - 0.24\tilde{c}^{(8)}_{HD} - 0.24\tilde{c}^{(8)}_{HD,2} + 3.9[\tilde{c}^{(6)}_{H\square}]^2 - 2.\tilde{c}^{(6)}_{H\square}\tilde{c}^{(6)}_{HD} + 0.27[\tilde{c}^{(6)}_{HD}]^2 \qquad \text{(D.10)}$$

$$-2.1\frac{\hat{v}}{\bar{m}_\mu}\tilde{c}^{(6)}_{eH} + 1.1\frac{\hat{v}^2}{\bar{m}^2_\mu}[\tilde{c}^{(6)}_{eH}]^2 - 1.\frac{\hat{v}}{\bar{m}_\mu}\tilde{c}^{(8)}_{eH} - 2.1\frac{\hat{v}}{\bar{m}_\mu}\tilde{c}^{(6)}_{eH}\tilde{c}^{(6)}_{H\square} + 0.61\frac{\hat{v}}{\bar{m}_\mu}\tilde{c}^{(6)}_{eH}\tilde{c}^{(6)}_{HD}$$

$$+0.016\tilde{c}^{(6)}_{H\square}\frac{\delta G_F}{\hat{G}_F} - 0.004\tilde{c}^{(6)}_{HD}\frac{\delta G_F}{\hat{G}_F} - 0.025\frac{\hat{v}}{\bar{m}_\mu}\tilde{c}^{(6)}_{eH}\frac{\delta G_F}{\hat{G}_F} + 3.9\cdot10^{-8}\frac{\delta G_F}{\hat{G}_F}$$

$$-0.094\tilde{c}^{(6)}_{H\square}|\tilde{c}^{(6)}_{HWB}| + 0.024\tilde{c}^{(6)}_{HD}|\tilde{c}^{(6)}_{HWB}| + 0.1\frac{\hat{v}}{\bar{m}_\mu}\tilde{c}^{(6)}_{eH}|\tilde{c}^{(6)}_{HWB}|$$

$$+10^2\left(+4.8[\tilde{c}^{(6)}_{HB}]^2 - 5.8\tilde{c}^{(6)}_{HB}\tilde{c}^{(6)}_{HW} + 3.9[\tilde{c}^{(6)}_{HW}]^2 + 2.8\tilde{c}^{(6)}_{HB}\tilde{c}^{(6)}_{HWB} - 5.7\tilde{c}^{(6)}_{HW}\tilde{c}^{(6)}_{HWB} + 2.3[\tilde{c}^{(6)}_{HWB}]^2\right)$$

$$+10^{10}\left(+2.5[\tilde{c}^{(6)}_{eB}]^2 - 2.7\tilde{c}^{(6)}_{eB}\tilde{c}^{(6)}_{eW} + 0.72[\tilde{c}^{(6)}_{eW}]^2\right)$$

$$+4.6\left(\tilde{c}^{(6)}_{HB} + \tilde{c}^{(8)}_{HB}\right) - 6.2\left(\tilde{c}^{(6)}_{HW} + \tilde{c}^{(8)}_{HW} + \tilde{c}^{(8)}_{HW,2}\right) + 4.6\left(\tilde{c}^{(6)}_{HWB} + \tilde{c}^{(8)}_{HWB}\right)$$

$$+9.1\left[\tilde{c}^{(6)}_{HB}\right]^2 - 12.\left[\tilde{c}^{(6)}_{HW}\right]^2 + 6.6\left[\tilde{c}^{(6)}_{HWB}\right]^2 + 18.\tilde{c}^{(6)}_{HB}\tilde{c}^{(6)}_{HWB} - 1.4\tilde{c}^{(6)}_{HW}\tilde{c}^{(6)}_{HWB}$$

$$+4.6\tilde{c}^{(6)}_{HB}\tilde{c}^{(6)}_{H\square} + 4.8\tilde{c}^{(6)}_{HB}\tilde{c}^{(6)}_{HD} - 6.2\tilde{c}^{(6)}_{HW}\tilde{c}^{(6)}_{H\square} - 5.7\tilde{c}^{(6)}_{HW}\tilde{c}^{(6)}_{HD} + 4.6\tilde{c}^{(6)}_{HWB}\tilde{c}^{(6)}_{H\square} + 12.\tilde{c}^{(6)}_{HWB}\tilde{c}^{(6)}_{HD}$$

$$-3.9\tilde{c}^{(6)}_{HB}\tilde{c}^{(6)}_{He} + 7.4\tilde{c}^{(6)}_{HB}\left(\tilde{c}^{(6),1}_{Hl} + \tilde{c}^{(6),3}_{Hl}\right) + 3.9\tilde{c}^{(6)}_{HW}\tilde{c}^{(6)}_{He} - 7.4\tilde{c}^{(6)}_{HW}\left(\tilde{c}^{(6),1}_{Hl} + \tilde{c}^{(6),3}_{Hl}\right) - 2.6\tilde{c}^{(6)}_{HWB}\tilde{c}^{(6)}_{He}$$

$$+4.9\tilde{c}^{(6)}_{HWB}\left(\tilde{c}^{(6),1}_{Hl} + \tilde{c}^{(6),3}_{Hl}\right)$$

$$+2.9\cdot10^{-7}\tilde{c}^{(6)}_{HB}\frac{\delta G_F}{\hat{G}_F} - 3.9\cdot10^{-7}\tilde{c}^{(6)}_{HW}\frac{\delta G_F}{\hat{G}_F} + 2.9\cdot10^{-7}\tilde{c}^{(6)}_{HWB}\frac{\delta G_F}{\hat{G}_F}$$

$$+0.29\tilde{c}^{(8),1}_{BH^4D^2} - 0.077\tilde{c}^{(8),1}_{WH^4D^2}$$

$$+0.22\tilde{c}^{(8),1}_{e^2BH^2D} + 0.039\left(\tilde{c}^{(8),1}_{e^2H^2D^3} + \tilde{c}^{(8),2}_{e^2H^2D^3}\right) - 0.12\tilde{c}^{(8),1}_{e^2WH^2D} - 0.43\left(\tilde{c}^{(8),1}_{L^2BH^2D} + \tilde{c}^{(8),5}_{L^2BH^2D}\right)$$

$$-0.076\left(\tilde{c}^{(8),1}_{l^2H^2D^3} + \tilde{c}^{(8),2}_{l^2H^2D^3}\right) - 0.038\left(\tilde{c}^{(8),3}_{l^2H^2D^3} + \tilde{c}^{(8),4}_{l^2H^2D^3}\right) + 0.23\left(\tilde{c}^{(8),1}_{l^2WH^2D} + \tilde{c}^{(8),5}_{l^2H^2D^3}\right)$$

$$\Delta_{M_W}^{R_5} = +0.93\tilde{c}_{H\Box}^{(6)} - 0.23\tilde{c}_{HD}^{(6)} - 0.12\tilde{c}_{HD}^{(8)} - 0.12\tilde{c}_{HD,2}^{(8)} + 1.9[\tilde{c}_{H\Box}^{(6)}]^2 - 1.8\tilde{c}_{H\Box}^{(6)}\tilde{c}_{HD}^{(6)} + 0.33[\tilde{c}_{HD}^{(6)}]^2 \qquad \text{(D.11)}$$

$$-0.99\frac{\hat{v}}{\bar{m}_\mu}\tilde{c}_{eH}^{(6)} + 0.53\frac{\hat{v}^2}{\bar{m}_\mu^2}[\tilde{c}_{eH}^{(6)}]^2 - 0.5\frac{\hat{v}}{\bar{m}_\mu}\tilde{c}_{eH}^{(8)} - 0.99\frac{\hat{v}}{\bar{m}_\mu}\tilde{c}_{eH}^{(6)}\tilde{c}_{H\Box}^{(6)} + 1.2\frac{\hat{v}}{\bar{m}_\mu}\tilde{c}_{eH}^{(6)}\tilde{c}_{HD}^{(6)}$$

$$+0.0076\tilde{c}_{H\Box}^{(6)}\frac{\delta G_F}{\hat{G}_F} - 0.0019\tilde{c}_{HD}^{(6)}\frac{\delta G_F}{\hat{G}_F} - 0.012\frac{\hat{v}}{\bar{m}_\mu}\tilde{c}_{eH}^{(6)}\frac{\delta G_F}{\hat{G}_F} + 1.9\cdot 10^{-8}\frac{\delta G_F}{\hat{G}_F}$$

$$-0.93\tilde{c}_{H\Box}^{(6)}|\tilde{c}_{HWB}^{(6)}| + 0.23\tilde{c}_{HD}^{(6)}|\tilde{c}_{HWB}^{(6)}| + 0.99\frac{\hat{v}}{\bar{m}_\mu}\tilde{c}_{eH}^{(6)}|\tilde{c}_{HWB}^{(6)}|$$

$$+10^3\left(+8.1[\tilde{c}_{HB}^{(6)}]^2 - 18.\tilde{c}_{HB}^{(6)}\tilde{c}_{HW}^{(6)} + 6.8[\tilde{c}_{HW}^{(6)}]^2 + 12.\tilde{c}_{HB}^{(6)}\tilde{c}_{HWB}^{(6)} - 8.\tilde{c}_{HW}^{(6)}\tilde{c}_{HWB}^{(6)} + 2.1[\tilde{c}_{HWB}^{(6)}]^2\right)$$

$$+10^{11}\left(+8.[\tilde{c}_{eB}^{(6)}]^2 - 8.6\tilde{c}_{eB}^{(6)}\tilde{c}_{eW}^{(6)} + 2.3[\tilde{c}_{eW}^{(6)}]^2\right)$$

$$+160.\left(\tilde{c}_{HB}^{(6)} + \tilde{c}_{HB}^{(8)}\right) - 120.\left(\tilde{c}_{HW}^{(6)} + \tilde{c}_{HW}^{(8)} + \tilde{c}_{HW,2}^{(8)}\right) + 69.\left(\tilde{c}_{HWB}^{(6)} + \tilde{c}_{HWB}^{(8)}\right)$$

$$+310.\left[\tilde{c}_{HB}^{(6)}\right]^2 - 240.\left[\tilde{c}_{HW}^{(6)}\right]^2 + 99.\left[\tilde{c}_{HWB}^{(6)}\right]^2 + 250.\tilde{c}_{HB}^{(6)}\tilde{c}_{HWB}^{(6)} + 19.\tilde{c}_{HW}^{(6)}\tilde{c}_{HWB}^{(6)}$$

$$+160.\tilde{c}_{HB}^{(6)}\tilde{c}_{H\Box}^{(6)} - 83.\tilde{c}_{HB}^{(6)}\tilde{c}_{HD}^{(6)} - 120.\tilde{c}_{HW}^{(6)}\tilde{c}_{H\Box}^{(6)} + 71.\tilde{c}_{HW}^{(6)}\tilde{c}_{HD}^{(6)} + 69.\tilde{c}_{HWB}^{(6)}\tilde{c}_{H\Box}^{(6)} + 130.\tilde{c}_{HWB}^{(6)}\tilde{c}_{HD}^{(6)}$$

$$-160.\tilde{c}_{HB}^{(6)}\tilde{c}_{He}^{(6)} + 98.\tilde{c}_{HB}^{(6)}\left(\tilde{c}_{Hl}^{(6),1} + \tilde{c}_{Hl}^{(6),3}\right) + 160.\tilde{c}_{HW}^{(6)}\tilde{c}_{He}^{(6)} - 98.\tilde{c}_{HW}^{(6)}\left(\tilde{c}_{Hl}^{(6),1} + \tilde{c}_{Hl}^{(6),3}\right) - 110.\tilde{c}_{HWB}^{(6)}\tilde{c}_{He}^{(6)}$$

$$+66.\tilde{c}_{HWB}^{(6)}\left(\tilde{c}_{Hl}^{(6),1} + \tilde{c}_{Hl}^{(6),3}\right)$$

$$-6.9\cdot 10^{-7}\tilde{c}_{HB}^{(6)}\frac{\delta G_F}{\hat{G}_F} + 5.2\cdot 10^{-7}\tilde{c}_{HW}^{(6)}\frac{\delta G_F}{\hat{G}_F} - 3.\cdot 10^{-7}\tilde{c}_{HWB}^{(6)}\frac{\delta G_F}{\hat{G}_F}$$

$$+6.2\tilde{c}_{BH^4D^2}^{(8),1} - 1.7\tilde{c}_{WH^4D^2}^{(8),1}$$

$$-9.5\tilde{c}_{e^2BH^2D}^{(8),1} - 1.7\left(\tilde{c}_{e^2H^2D^3}^{(8),1} + \tilde{c}_{e^2H^2D^3}^{(8),2}\right) + 5.1\tilde{c}_{e^2WH^2D}^{(8),1} + 6.8\left(\tilde{c}_{L^2BH^2D}^{(8),1} + \tilde{c}_{L^2BH^2D}^{(8),5}\right)$$

$$+1.2\left(\tilde{c}_{l^2H^2D^3}^{(8),1} + \tilde{c}_{l^2H^2D^3}^{(8),2}\right) + 0.59\left(\tilde{c}_{l^2H^2D^3}^{(8),3} + \tilde{c}_{l^2H^2D^3}^{(8),4}\right) - 3.7\left(\tilde{c}_{l^2WH^2D}^{(8),1} + \tilde{c}_{l^2H^2D^3}^{(8),5}\right)$$

$$\Delta_{M_W}^{R_6} = +0.83\tilde{c}_{H\Box}^{(6)} - 0.21\tilde{c}_{HD}^{(6)} - 0.1\tilde{c}_{HD}^{(8)} - 0.1\tilde{c}_{HD,2}^{(8)} + 1.7[\tilde{c}_{H\Box}^{(6)}]^2 - 1.7\tilde{c}_{H\Box}^{(6)}\tilde{c}_{HD}^{(6)} + 0.32[\tilde{c}_{HD}^{(6)}]^2 \tag{D.12}$$

$$-0.88\frac{\hat{v}}{\bar{m}_\mu}\tilde{c}_{eH}^{(6)} + 0.47\frac{\hat{v}^2}{\bar{m}_\mu^2}[\tilde{c}_{eH}^{(6)}]^2 - 0.44\frac{\hat{v}}{\bar{m}_\mu}\tilde{c}_{eH}^{(8)} - 0.88\frac{\hat{v}}{\bar{m}_\mu}\tilde{c}_{eH}^{(6)}\tilde{c}_{H\Box}^{(6)} + 1.1\frac{\hat{v}}{\bar{m}_\mu}\tilde{c}_{eH}^{(6)}\tilde{c}_{HD}^{(6)}$$

$$+0.0068\tilde{c}_{H\Box}^{(6)}\frac{\delta G_F}{\hat{G}_F} - 0.0017\tilde{c}_{HD}^{(6)}\frac{\delta G_F}{\hat{G}_F} - 0.011\frac{\hat{v}}{\bar{m}_\mu}\tilde{c}_{eH}^{(6)}\frac{\delta G_F}{\hat{G}_F} + 1.7\cdot 10^{-8}\frac{\delta G_F}{\hat{G}_F}$$

$$-0.91\tilde{c}_{H\Box}^{(6)}|\tilde{c}_{HWB}^{(6)}| + 0.23\tilde{c}_{HD}^{(6)}|\tilde{c}_{HWB}^{(6)}| + 0.96\frac{\hat{v}}{\bar{m}_\mu}\tilde{c}_{eH}^{(6)}|\tilde{c}_{HWB}^{(6)}|$$

$$+10^4\left(+4.[\tilde{c}_{HB}^{(6)}]^2 + 0.56\tilde{c}_{HB}^{(6)}\tilde{c}_{HW}^{(6)} + 0.76[\tilde{c}_{HW}^{(6)}]^2 - 2.7\tilde{c}_{HB}^{(6)}\tilde{c}_{HWB}^{(6)} - 1.6\tilde{c}_{HW}^{(6)}\tilde{c}_{HWB}^{(6)} + 1.1[\tilde{c}_{HWB}^{(6)}]^2\right)$$

$$+10^{12}\left(+1.5[\tilde{c}_{eB}^{(6)}]^2 - 1.6\tilde{c}_{eB}^{(6)}\tilde{c}_{eW}^{(6)} + 0.43[\tilde{c}_{eW}^{(6)}]^2\right)$$

$$-43.\left(\tilde{c}_{HB}^{(6)} + \tilde{c}_{HB}^{(8)}\right) - 130.\left(\tilde{c}_{HW}^{(6)} + \tilde{c}_{HW}^{(8)} + \tilde{c}_{HW,2}^{(8)}\right) + 130.\left(\tilde{c}_{HWB}^{(6)} + \tilde{c}_{HWB}^{(8)}\right)$$

$$-86.\left[\tilde{c}_{HB}^{(6)}\right]^2 - 260.\left[\tilde{c}_{HW}^{(6)}\right]^2 - 44.\left[\tilde{c}_{HWB}^{(6)}\right]^2 + 370.\tilde{c}_{HB}^{(6)}\tilde{c}_{HWB}^{(6)} + 190.\tilde{c}_{HW}^{(6)}\tilde{c}_{HWB}^{(6)}$$

$$-43.\tilde{c}_{HB}^{(6)}\tilde{c}_{H\Box}^{(6)} - 110.\tilde{c}_{HB}^{(6)}\tilde{c}_{HD}^{(6)} - 130.\tilde{c}_{HW}^{(6)}\tilde{c}_{H\Box}^{(6)} + 180.\tilde{c}_{HW}^{(6)}\tilde{c}_{HD}^{(6)} + 130.\tilde{c}_{HWB}^{(6)}\tilde{c}_{H\Box}^{(6)} - 15.\tilde{c}_{HWB}^{(6)}\tilde{c}_{HD}^{(6)}$$

$$-140.\tilde{c}_{HB}^{(6)}\tilde{c}_{He}^{(6)} + 57.\tilde{c}_{HB}^{(6)}\left(\tilde{c}_{Hl}^{(6),1} + \tilde{c}_{Hl}^{(6),3}\right) + 140.\tilde{c}_{HW}^{(6)}\tilde{c}_{He}^{(6)} - 57.\tilde{c}_{HW}^{(6)}\left(\tilde{c}_{Hl}^{(6),1} + \tilde{c}_{Hl}^{(6),3}\right) - 91.\tilde{c}_{HWB}^{(6)}\tilde{c}_{He}^{(6)}$$

$$+38.\tilde{c}_{HWB}^{(6)}\left(\tilde{c}_{Hl}^{(6),1} + \tilde{c}_{Hl}^{(6),3}\right)$$

$$+4.8\cdot 10^{-7}\tilde{c}_{HB}^{(6)}\frac{\delta G_F}{\hat{G}_F} + 1.5\cdot 10^{-6}\tilde{c}_{HW}^{(6)}\frac{\delta G_F}{\hat{G}_F} - 1.5\cdot 10^{-6}\tilde{c}_{HWB}^{(6)}\frac{\delta G_F}{\hat{G}_F}$$

$$+4.5\tilde{c}_{BH^4D^2}^{(8),1} - 1.2\tilde{c}_{WH^4D^2}^{(8),1}$$

$$-10.\tilde{c}_{e^2BH^2D}^{(8),1} - 1.8\left(\tilde{c}_{e^2H^2D^3}^{(8),1} + \tilde{c}_{e^2H^2D^3}^{(8),2}\right) + 5.5\tilde{c}_{e^2WH^2D}^{(8),1} + 3.3\left(\tilde{c}_{L^2BH^2D}^{(8),1} + \tilde{c}_{L^2BH^2D}^{(8),5}\right)$$

$$+0.55\left(\tilde{c}_{l^2H^2D^3}^{(8),1} + \tilde{c}_{l^2H^2D^3}^{(8),2}\right) + 0.28\left(\tilde{c}_{l^2H^2D^3}^{(8),3} + \tilde{c}_{l^2H^2D^3}^{(8),4}\right) - 1.7\left(\tilde{c}_{l^2WH^2D}^{(8),1} + \tilde{c}_{l^2H^2D^3}^{(8),5}\right)$$

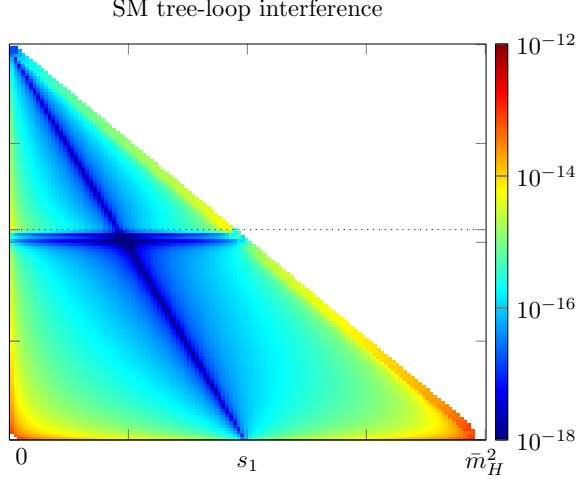

Figure 8: Dalitz plot showing the interference between the one-loop and tree-level SM amplitudes for the case $m_\ell = m_\tau$.

# E    Discussion on the tree-loop interference in the SM

Comparing the size of the tree level contributions discussed in Sec 3.1 and the loop contributions of Sec 3.2 one may be concerned, particularly in the case of the tau, that the tree-loop interference could be large or even dominant.

To achieve a more precise understanding of the size of this contribution we have estimated it by considering the interference between the one-loop triangle diagrams contributing to $h \to \gamma\gamma$ and $h \to Z\gamma$, i.e. those of Figure 1 or equivalently Figure 3a, and the tree level diagrams. This is convenient as we are able to obtain the full $m_\ell$ dependence of this interference as the loops contain no $m_\ell$ dependence. It comes with the caveat that these diagrams are not gauge-invariant on their own. We have performed the calculation in the Feynman Gauge, $\xi \to 1$, where we have tested that the triangle diagrams make a larger contribution than the box diagrams (this is also noted in e.g. [5]). Table 5 shows the size of the contribution from diagrams with only top and EW contributions, only diagrams containing internal lepton lines, and the full contribution for the squared-loop contribution to the partial width of the Higgs boson to two taus and a photon. The reason for this simplification is that evaluation of the Passarino Veltman functions corresponding to diagrams with internal fermions and their corresponding mass dependence is well beyond the scope of this work.

Figure 8 shows the Dalitz plot for the tree-loop interference described above for $m_\ell = m_\tau$.

|  | $\xi = 1$ | $\xi = 2$ | $\xi = 3$ |
|---|---|---|---|
| top+EW: | $2.84 \cdot 10^{-7}$ | $2.83 \cdot 10^{-7}$ | $2.83 \cdot 10^{-7}$ |
| leptonic internal lines: | $6.08 \cdot 10^{-10}$ | $2.17 \cdot 10^{-10}$ | $1.17 \cdot 10^{-10}$ |
| full result: | $2.84 \cdot 10^{-7}$ | $2.84 \cdot 10^{-7}$ | $2.84 \cdot 10^{-7}$ |

Table 5: Comparison between the purely top+electroweak contributions, strictly diagrams with internal leptonic lines, and the full calculation of the one-loop squared contribution to $H \to \tau\tau\gamma$. Errors are approximately per mil.

We note the distinct lines where the amplitude squared drops off, as with the diagrams of Figure 6, these lines correspond to changes in overall sign. They roughly correspond to the following two lines,

$$s_2 = \bar{m}_Z^2 \,, \tag{E.1}$$
$$s_2 = \bar{m}_H^2 - 2s_1 \,. \tag{E.2}$$

Note that above, $s_2 = \bar{m}_Z^2$ actually corresponds to two lines close together just below the $Z$-threshold. We define four regions of integration as follows:

$$
\begin{aligned}
R_1 &= \{s_2 < \bar{M}_Z^2 \quad \& \quad s_2 < \bar{m}_H^2 - 2s_1\} \,, \\
R_2 &= \{s_2 < \bar{M}_Z^2 \quad \& \quad s_2 > \bar{m}_H^2 - 2s_1\} \,, \\
R_3 &= \{s_2 > \bar{M}_Z^2 \quad \& \quad s_2 < \bar{m}_H^2 - 2s_1\} \,, \\
R_4 &= \{s_2 > \bar{M}_Z^2 \quad \& \quad s_2 > \bar{m}_H^2 - 2s_1\} \,.
\end{aligned}
\tag{E.3}
$$

Integrating over the phase space for each of these regions we find:

$$
\begin{aligned}
\Gamma(R_1) &= -1.439 \cdot 10^{-7} \,, \\
\Gamma(R_2) &= +1.439 \cdot 10^{-7} \,, \\
\Gamma(R_3) &= -1.383 \cdot 10^{-8} \,, \\
\Gamma(R_4) &= +1.384 \cdot 10^{-8} \,.
\end{aligned}
\tag{E.4}
$$

The error in these integrations is per mil. We see that, within errors, Regions 1 and 2 cancel as do 3 and 4, giving an overall vanishing contribution from the tree-loop interference. This cancellation is the reason for the need to separate the integration into regions – the Vegas algorithm was unable to converge integrating over the full phase space. Given the seemingly exact cancellation between these well defined regions it seems likely there is a kinematic or symmetry argument for the cancellation. We have not, however, explored this possibility in detail.

The above indicates the leading contributions to the tree-loop interference comes from the box diagrams. As concerns about the tree-loop interference being important hinges on the overall size of the one-loop squared contribution, for which the diagrams with fermions in the loop are subdominant, we conclude the tree-loop interference is negligible. Alternatively we can argue that, because the size of the $\Gamma(R_i)$ given above are at least two orders of magnitude smaller than the tree-level contribution for the taus the tree-loop interference is negligible. Note that in the discussions of Sec. 4.2 flavor is invoked only as a means of testing if the impact of the Class 3 operators (corresponding to SM tree-like kinematics) can be elevated above the other contributions. If there is a kinematic or symmetry argument for the cancellation this will also carry partially to the contributions from the box diagrams corresponding to the chiral structures of Eq. 16. The box diagrams, however, contain lepton mass insertions and therefore will generate the chiral structures:

$$
\begin{aligned}
&\bar{u}_{k2} P_{\pm} v_{k3} \,, \\
&\bar{u}_{k2} \sigma_{\mu\nu} v_{k3} \,.
\end{aligned}
\tag{E.5}
$$

These structures will correspond to tree-loop interference terms suppressed by $\frac{m_\ell^4}{16\pi^2}$ and are expected to be negligible.

*We reiterate that the above argument is a gauge dependent statement as gauge-independence is only obtained after inclusion of the diagrams with leptons in the loops.* Our results indicate that for gauge choices $\xi = 1, 2, 3$ that the diagrams with internal lepton lines contribute at a level nearly 3 orders of magnitude smaller than the pure electroweak and top contributions.

## F  Dipole operators

Here we discuss the impact of the dipole operators on our analysis. This is to get a better understanding of the potential size of including operators which induce chiral flips has on our analysis. We note that, in the case of dimension-eight operators, they must contribute proportional to $m_\ell$ as their chiral structure can only interfere with the SM tree-level amplitude (see discussion in App. E for more details on the interference of the SM-tree and loop contributions).

The dimension-six dipole operators relevant to our analysis are given by:

$$
\begin{array}{rcl}
Q_{eW} & = & (\bar{l}\sigma^{\mu\nu}e)\tau^I H W_{\mu\nu}^I \\
Q_{eB} & = & (\bar{l}\sigma^{\mu\nu}e)H B_{\mu\nu}
\end{array}
\tag{F.1}
$$

Note these operators are not hermitian, and therefore their Wilson coefficients are in general complex. We treat them as real, invoking stringent constraints from low energy CP measurements as a reason for treating the imaginary parts as negligible [54]. These operators can contribute to the $H \to \ell\ell\gamma$ process at tree level in many ways. They generate a four-point contact interaction and a dipole interaction of the leptons with the photon. The latter allows for both dipole-like corrections to the SM-like topology as well as to the SMEFT coupling $HV\gamma$ where $V = \{\gamma, Z\}$. The amplitudes resulting from these contributions are given by:

$$
\mathcal{M}_{\text{contact}} \;=\; c_{DP}\bar{u}_{k_2}\sigma^{\mu\nu}v_{k_3}k_1^\nu \epsilon_\mu^*(k_1)\,,
\tag{F.2}
$$

$$
\mathcal{M}_{\text{Yukawa}} \;=\; \frac{\hat{m}_\ell}{v}(1 + \Delta_{H\bar{\ell}\ell})c'_{DP}\left[\frac{\bar{u}_{k_2}\sigma^{\mu\nu}(\slashed{k}_1 + \slashed{k}_2 + \hat{m}_\ell)v_{k_3}}{(k_1 + k_2)^2 - \hat{m}_\ell^2} + \frac{\bar{u}_{k_2}(-\slashed{k}_1 - \slashed{k}_3 + \hat{m}_\ell)\sigma^{\mu\nu}v_{k_3}}{(k_1 + k_3)^2 - \hat{m}_\ell^2}\right]\epsilon_\mu^*(k_1)k_1^\nu\,,
\tag{F.3}
$$

$$
\mathcal{M}_{VV} \;=\; g_{HVV}vc'_{DP}\Pi^{\mu\nu}\bar{u}_{k_2}\sigma^{\mu\nu}v_{k_3}\frac{(k_2 + k_3)^\nu}{(k_2 + k_3)^2 - \bar{m}_V^2}\epsilon_\mu^*(k_1)\,.
\tag{F.4}
$$

Note we have retained fermion masses in the propagators as these interactions can induce chiral flips and we wish to estimate their relevant to the decay into tau leptons.

$|\mathcal{M}_{\text{contact}}|^2$ gives a mass independent contribution and so is relevant for all lepton flavors. Table 6 lists the possible contributions from dipole operators to the process $H \to \gamma\ell\ell$, their leading order in the expansion in $m_\ell$, and the size of this contributions. Excluded combinations occur only at order $1/\Lambda^6$ or higher.

Table 6 demonstrates that the dominant contributions of the dipole operators come from the tree level interference of two dimension-six two amplitudes. Subleading is the contribution from the interference of the SM with the contact interactions $HVV$ and the dipole operator in the same amplitude. The terms occurring at $\frac{1}{\Lambda^2}$ are strongly suppressed relative to these contributions. In simplifying our analysis we use this observation to drop the contribution of all dimension-eight operators with a chiral flip, as they can only interfere with the SM amplitude. There are 7 additional dimension-eight operators which induce chiral flips, some

|  | $\mathcal{O}(1/\Lambda^i)$ | $\mathcal{O}(m^i)$ | PS Integral $\tau$ | PS Integral $\mu$ |
|---|---|---|---|---|
| $c_{DP} \otimes c_{DP}$ | $\frac{1}{\Lambda^4}$ | $1$ | $6.4 \cdot 10^5$ | $6.4 \cdot 10^5$ |
| $Y\bar{e} \otimes g_{HAA}c'_{DP}$ | $\frac{1}{\Lambda^4}$ | $\hat{m}_\ell$ | $2.3 \cdot 10^4$ | $1300$ |
| $Y\bar{e} \otimes g_{HAZ}c'_{DP}$ | $\frac{1}{\Lambda^4}$ | $\hat{m}_\ell$ | $-1600$ | $-100$ |
| $Y\bar{e} \otimes c_{DP}$ | $\frac{1}{\Lambda^2}$ | $\hat{m}_\ell$ | $1.1$ | $6.6 \cdot 10^{-2}$ |
| $Yc'_{DP} \otimes Yc'_{DP}$ | $\frac{1}{\Lambda^4}$ | $\hat{m}_\ell^2$ | $1000$ | $3.6$ |
| $Y\bar{e} \otimes Yc'_{DP}$ | $\frac{1}{\Lambda^2}$ | 0 | | |
| $Yc'_{DP} \otimes c_{DP}$ | $\frac{1}{\Lambda^4}$ | 0 | | |

Table 6: Combinations of the dipole couplings and other couplings contributing to the partial width of the Higgs boson to two leptons and a photon, their leading contribution in the $m_\ell$ expansion, and the size of the corresponding phase-space integral for the case of the tau and muon. The phase space integrations include the full mass dependence and are only expressed to two significant digits. $Y$ is the Yukawa couplings proportional to the mass of the lepton, this lepton mass dependence is included in the phase space integral as is the $\bar{e}$ dependence. Entries for mass expansion of "0" indicate the amplitude squared is identically 0 (the two contributing amplitudes do not interfere).

examples of these include:

$$Q^{(2)}_{leWH^3} = (\bar{l}\sigma^{\mu\nu}e_R)H(H^\dagger\tau^I H)W^I_{\mu\nu}, \tag{F.5}$$

$$Q^{(1)}_{leWHD^2} = (\bar{l}\sigma^{\mu\nu}D^\rho e_R)\tau^I(D_\nu H)W^I_{\rho\mu}. \tag{F.6}$$

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
