# Peer review of "Higgs decays to two leptons and a photon beyond leading order in the SMEFT"

_SciPost Physics_

## Round 2 · Referee Report · Anonymous · 2022-1-24

Report
The paper considers the $H\to \ell\bar\ell\gamma$ decay in the SMEFT, and considers the contribution to the decay rate from interference of tree-level SMEFT amplitudes (up to dimension 8). In addition, the interference of the tree-level SMEFT amplitude with the one-loop SM amplitude (in the limit of $m_{\ell\to0}$ is considered.
The paper contains a combination of technical advancements (in computing SMEFT amplitudes up to dimension 8, and the corresponding implementation) and applies the machinery to the case of $H\to \ell\bar\ell\gamma$ decays (at the high-lump LHC). While the paper contains new information which is relevant and valuable for the field, I believe the presentation of the results lacks clarity and that the validity of certain assumptions made in the calculation to be poor or unjustified.
I would recommend the paper for publication after these major criticisms are addressed.
To assist the authors, I have made a list of comments below, which I into “technical comments on the presented calculation” as well as ”comments on the phenomenology”. I hope the comments are received with the aim, that they are constructive.
$\textbf{Technical comments on the presented calculation}$
Section 3.1
Large corrections induced in the presence of exact mass effects (e.g. +26% quoted for tau-leptons). Can the authors comment upon this? Naively one would expect that exact mass effects introduce power corrections of the form $m_\ell^2/Q^2$. For an inclusive decay rate, the scale Q would be the Higgs mass, meaning these corrections would be small. Instead, the large correction may indicate strong power-correction dependence on the energy cut of the photon, which is used to regulate the soft and collinear divergence. Probably the source of correction is not logarithmic, since it is small for the electron.
Section 3.2
Do I understand the statement of Eq. (14) that corrections involving both the yukawa coupling ($H\ell\bar\ell$) as well corrections from propagators and phase-space are ignored? In other words, the decay rate is considered by squaring those one-loop SM amplitudes which vanish in the strict limit that the lepton mass is set to zero (including the yukawa). This should be made clear if it is the case, by defining exactly the loop order of contributions to the decay rate which are considered [in fact only later at the start of Section 4 is that made somewhat clearer, which is not useful for a reader]
Before equation (17) it is stated that the one-loop process “does not contain IR divergences”. Inspecting those diagrams in figure (3), it is not clear why the photon cannot be emitted in either a soft and/or collinear configuration. Can the authors comment on this?
The results for Eq (17) indicate that the (one-loop)-squared pieces are numerically similar for muons (factor of three large for [one-loop]-squared pieces) and substantially smaller than the squared tree-level contribution for tau-leptons. How justified is it then to ignore those terms which are chirally suppressed but not loop suppressed (i.e. $SM^{0} * SM^{1}$? It seems not well justified for muons, and simply a bad approximation for taus.
While these comments/questions are for the SM part, which is not the main focus of the paper, as far as I can tell these assumptions are also carried across to the study of SMEFT contributions and should therefore be clarified.
Section 4.2
I find the notation $|M_{full}|^2$, i.e. “Full” to be a little misleading as a vast number of terms have been excluded.
As I understand, the expression given in Eq. (33) is an expansion in $1/\Lambda^2$ and in loop-order. But all amplitudes $M_{C}$ (which refer to specific cases of SMEFT insertions, not operator classes) are tree-level. Is this correct?
Can the authors comment further on the potential impact of M_{C} when considered beyond leading-order? In particular, it is not clear to me that the interference of two one-loop SMEFT amplitudes cannot give new kinematic structure. Or is there a reason why that is not the case.
As muon final states are considered, and the tree-level SM contribution is of the same order as the one-loop SM parts in the $m_{\ell}\to0$ limit, interference of $M_{C}$ amplitudes with the tree-level SM is surely necessary? Or have I missed an important point.
$\textbf{Comments on the phenomenology }$
With regards to the event rates in Table 3, expected with 3/ab integrated luminosity. How are these event rates estimated? This seems like a critical point for the paper, as it indicates whether the kinematic regions in the Dalitz plot will be feasibly accessed (and hence provide potentially useful information on the structure of new physics) in an experimental setup.
In particular, the muon final state is considered. This is presumably because the feasibility (i.e. efficiency, and background rejection) is higher for this channel. What efficiency (and justification) are assumed. If dressed leptons are considered (i.e. those reconstructed as a leptonic jet avoiding IRC unsafe configurations), could it also be necessary to consider $H\to \ell\bar\ell \gamma\gamma$ SMEFT contributions?

---

## Round 2 · Referee Report · Anonymous · 2022-3-2

Report
Sorry for the delay. (FWIW, I originally tried to submit it over a week ago.)
This paper presents a groundbreaking calculation of $h \to Z \gamma$ in the Standard Model Effective Field Theory (SMEFT). It is the first work to my knowledge to interfere a full one-loop SM amplitude with a claim full tree level, dimension-8 SMEFT amplitudes. My main concern is it appears not all the dimension-8 operators were included.
In particular, the so-called "Class 8" operators of Ref. [35] also contribute to this process. Specifically, the operators $Q_{WH^4D^2}^{(1)}$ and $Q_{BH^4D^2}^{(1)}$ generate an amplitude proportional to $(Z^\mu \partial^\nu h -Z^\nu \partial^\mu h) F_{\mu\nu}$. Similarly, the operators generate $Q_{WH^4D^2}^{(2)}$ and $Q_{BH^4D^2}^{(2)}$ generate an analogous amplitude but with a dual electromagnetic field strength.
These operators should be discussed in Section 4. Additionally, they should either also be included in the analysis of Sections 4.1 and 4.2 or a reason should be given as to why don't contribute.
There is a silver lining here. The inclusion of the Class 8 operators would provide additional motivation for studying the process $h \to Z \gamma$ as they do not contribute to $h \to \gamma gamma$ or $g g \to h$.

---

## Round 3 · Referee Report · Anonymous (Referee 3) · 2022-5-12

Report

The authors have adequately addressed the point I raised. I now recommend this manuscript for publication.

---

## Round 3 · Referee Report · Anonymous (Referee 1) · 2022-6-8

Report

I thank the authors for taking care in checking the results based on my initial comments. Firstly, I apologise for the delay in preparing my report. Given the numerous changes, I wanted to take care to read the new version.

Overall, the authors have responded to each of my comments and provided explanation, changes, and also referred to known results in the literature where relevant. This is all welcome.

I had not noticed when preparing my initial report, but is there an angular separation requirement placed between the charged leptons and the photon? I understand that the cut on the photon energy removes the region of phase-space contributing to the soft divergence, but wondered if the collinear emission is still a possible issue. This could be avoided by requiring an isolation requirement on the charged lepton (e.g. deltaR[lepton,photon] > dR_min). I believe this issue could explain why ‘we retain the fermion mass dependence in the denominators as this leads to quicker numerical convergence of the integral’.

Is it possible that this is necessary in the numerical integration to avoid the collinear singularity (effectively replacing it with a collinear logarithm of the fermion mass)? If so, I would advise the authors to include such a deltaR separation. I believe then the results will be stable (and will change again).

---

## Round 3 · Author Response

We thank the referees for their constructive review of our article. We have attempted to implement nearly all changes requested. One-loop dimension-six calculations are indeed of the same order as those calculated here, but are beyond the scope of the work. We have changed the language in the article to more clearly indicate this and have reframed the discussion in the context of this.

---

## Round 3 · List of Changes

General notes for all referees: 1) Please note that in the need to address the size of the correction to the tree level results from including the full mass dependence relative to the leading mass dependence we found that the initial sampling of the phase space was inadequate and lead to incorrect results for the muon and electron. This has been remedied. As the muon tree level result is in the denominator of all the ratios (\Delta's) used in the paper we had to rerun all these results so they have changed modestly from the previously presented results. We have rechecked all PS integrals and found they are robust over 3 orders of magnitude of initial PS samplings using the Vegas algorithm. 2) We have, for the sake of cleanliness of presentation, dropped the inclusion of the full results of the ratios in the main text in favor of the more concise ones presented under the scaling/loop assumptions. We hope the referees agree this improves the presentation. 3) As a result of the many corrections needed to the text, please be aware that many equation/table/figure numbers may have changed. There is also a new 'case' corresponding to the dipole operators.

Response to Report 3 1) We thank the referee for pointing out our oversight of the 'class 8' operators. We have now included them. Please note that throughout this work we have neglected the CP odd operators, and so we have also neglected the CP odd "class 8" operators.

Response to Report 2 1) As noted in the general comments we found that our initial sampling of the PS for muons and electrons was inadequate and so has been rerun. The large corrections remain, but have changed. These large corrections can be understood from the E_gamma distribution of the decay as the referee guessed. We have cited Figure 4 of [11] to explain this behavior. 2) We have clarified the inclusion/exclusion of mass dependence in the loop PS integrations below Eq17. 3) Regarding IR divergences and the loops, we have cited [11] where they include an explanation below Eq.2.8 4) Regarding the interference of the tree- and loop- SM amplitudes we have included Appendix E. Appendix E shows that in the case of the triangle diagrams the interference is 0, and that for the box diagrams it is expected to be small, allowing us to neglect it our analysis both in the SM case and that of the SMEFT. 5) We have replaced M_full -> M to avoid confusion. 6) The D6 Wilson coefficients are not included in the one loop amplitudes. We have added a discussion of this below Eq 26. We expect based on power counting that our calculations of the (D6 tree)x(D6-loop) interference should be of the same order of magnitude as our 1/Lambda^4 results, and therefore our qualitative results should hold. We have emphasized that the quantitative results are partial results which should be improved in future studies leading up to the HL-LHC. These calculations are beyond the scope of this work which took advantage of the fact the SM loops are UV finite (the loops are not when D6 operators are included). 7) While one-loop SMEFT can give new kinematic structures, they won't yield new chiral structures within the context of the operators discussed. 8) For the muon tree-loop interference please see response 4 and appendix E. 9) The event rates are very simply estimated from the total Higgs cross section, the SM BRs, and the integrated luminosity. This is now included in the discussion at the bottom of p14. 10) No efficiencies are assumed, for muons this will result in a modest drop in event rate. It will likely be much more severe for taus. We have not explored this as it is beyond the scope of this work. The ratios defined in this work and the lack of study of this aspect is consistent with other SMEFT works. We agree that this should be improved for the future precision studies as we move toward the HL-LHC.

Response to Report 1 1) We agree corrections to \bar g_i have to cancel as the referee states, however we included these relations for the sake of clarifying notation for those unfamiliar with its usage in the SMEFT. If we had not included it another referee would likely have asked for clarification. 2) Input parameter dependence is derived from the Appendix of [34]. We have added a comment to this effect below Eq40. 3) We now mention the implicit dependence below B2 and have fixed the missing dependence in the full expressions. 4) The LO corrections were wrongfully neglected and have been fixed (they enter through the tree level diagram as mentioned in the referees previous comment). We have defined c_ll below the definition of dGF in Eq43. 5) We have replaced all instances of mbar->mhat as the fermion mass is always the input parameter. This is now mentioned in the caption of Table1. For general input parameter dependence please see response to point 2. 6) The sizable corrections come from expanding the amplitude squared in powers of m_\ell. This is now explained as due to the E_\gamma dependence and the cut on E_\gamma per another referee's question. 7) We now refer to the square vertex in place of the vertex denoted by a box to help distinguish the two contributions. 8) The difference comes from the phase space region, which is now referenced at the end of the paragraph below Eq.17. 9) We added a short discussion at the end of the introduction to Section 4 (i.e. the beginning of section 4 which is not labeled by a \subsection) indicating we have simply taken the couplings to be diagonal and that we make no assumptions about universality which is possible as our discussion is channel by channel. 10) We have added the contribution of the squares of dipole operators. Because of comments regarding the interference of the chiral flipping amplitude with the SM loop/SMEFT tree contributions we have further elaborated on the potential impact of dipole beyond the squared D6 term in Appendix F. We found in the case of the muon the D6^2 term is by far dominant, but for taus others may become relevant. 11) The numbering was changed, this was a mistake. This statement has been removed as we have now discussed the interference of the chiral flipping amplitudes and the others at the referees request. 12) They do not, they were originally included until it was realized they can only contribute at 1/Lambda^6 and higher order due to the assumptions in the text. We have clarified this where they are mentioned in the appendix. We have kept them in the appendix for completeness. 13) The comment on p19 was poorly phrased and has been removed. The neglect of the D6 loops is purely a simplifying assumption and we have clarified this in the discussion below (new) Eq26 (former Eq.24). Here we have mellowed the language to indicate that our loop-tree interference is an estimate of dependence, and that therefore the numerical results are inexact, but the qualitative discussion forming the core of the main text should still hold. 14) We have added a comment to this effect. 15) We previously checked against [8] and found agreement. Noting the large changes to the PS integrals for the muon mentioned in the general notes, we re-performed the checks and again found agreement. This supports the referees point that these cuts could be important in future analyses. We have added a sentence to the bottom of the discussion below (new) Eq42 to include the referees critique and emphasize the need for more detailed studies in the future. 16) We have added a statement below Eq 42 emphasizing that in the determination of the Deltas the full dependence is used, i.e. that the normalizations are included. If the referee's point was that in determining the regions of Table~3 we should include the normalizations as well, we agree this would allow for the best determination of ideal regions, but is beyond the scope of this work as can be seen from the complexity of the resulting 'Deltas' of appendix D. 17) We have done our best to address these typos. With the large amount of rewriting that took place in responding to all the referees reports we did our best to avoid any new typos.

---

## Editorial Decision

resubmitted